# Structural basis for triacylglyceride extraction from mycobacterial inner membrane by MFS transporter Rv1410

Sille Remm [1], Dario De Vecchis[2], Jendrik Schöppe [3,7], Cedric A. J. Hutter[1,8], Imre Gonda[1], Michael Hohl [1,9], Simon Newstead [4,5], Lars V. Schäfer [2] ✉ & Markus A. Seeger [1,6] ✉

*Mycobacterium tuberculosis* is protected from antibiotic therapy by a multi-layered hydrophobic cell envelope. Major facilitator superfamily (MFS) transporter Rv1410 and the periplasmic lipoprotein LprG are involved in transport of triacylglycerides (TAGs) that seal the mycomembrane. Here, we report a 2.7 Å structure of a mycobacterial Rv1410 homologue, which adopts an outward-facing conformation and exhibits unusual transmembrane helix 11 and 12 extensions that protrude ~20 Å into the periplasm. A small, very hydrophobic cavity suitable for lipid transport is constricted by a functionally important ion-lock likely involved in proton coupling. Combining mutational analyses and MD simulations, we propose that TAGs are extracted from the core of the inner membrane into the central cavity via lateral clefts present in the inward-facing conformation. The functional role of the periplasmic helix extensions is to channel the extracted TAG into the lipid binding pocket of LprG.

*M. tuberculosis* (Mtb) is the notorious pathogen causing tuberculosis, a disease that claims ~1.3 million lives annually world-wide[1]. The success of the pathogen depends on its hydrophobic multi-layered cell envelope, which is both a formidable barrier offering protection from the host environment and an interface mediating host-pathogen interactions during infection. The mycobacterial cell wall is reminiscent of the cell envelope of Gram-negative bacteria in that Mtb has a periplasmic space that connects the inner membrane to an outer membrane[2]. However, the compositions of these compartments differ greatly[3,4]. In the ~20–30 nm wide periplasmic space[2,5], an arabinogalactan layer serves as a connector between the peptidoglycan layer and the outer membrane as it is covalently attached to both. The outer membrane, also called mycomembrane, is composed of two asymmetrical leaflets. The inner leaflet mainly consists of mycolic acids[6], which are linked to the arabinogalactan via ester bonds, while the outer

leaflet is formed from various non-covalently bound lipid species[3,4]. At present, the exact composition and amount of the latter is a subject of debate[3,4].

Rv1410 (P55) is a major facilitator superfamily (MFS) transporter from Mtb. It is encoded in an operon together with lipoprotein LprG (Rv1411), which is embedded in the outer leaflet of the inner membrane via its lipid anchor[7]. The operon is highly conserved among mycobacterial species and both proteins are functionally contributing to intrinsic drug tolerance in mycobacteria[8]. Rv1410 has been initially described as a proton-driven drug efflux pump[9–11]. However, more recent work suggests that Rv1410 indirectly contributes to drug resistance, likely by sealing the mycomembrane[8] with triacylglycerides (TAGs)[12].

Using lipidomics, the loss of the *lprG/rv1410c* operon or the *rv1410c* gene in Mtb was shown to result in intracellular accumulation

[1]Institute of Medical Microbiology, University of Zurich, Zürich, Switzerland. [2]Center for Theoretical Chemistry, Ruhr University Bochum, Bochum, Germany. [3]Institute of Biochemistry, University of Zurich, Zürich, Switzerland. [4]Department of Biochemistry, University of Oxford, Oxford, United Kingdom. [5] Kavli Institute for Nanoscience Discovery, University of Oxford, Oxford, United Kingdom. [6]National Center for Mycobacteria, Zurich, Switzerland. [7]Present address: Global Research Technologies, Novo Nordisk A/S, Måløv, Denmark. [8]Present address: Linkster Therapeutics, Zürich, Switzerland. [9]Present address: Department of Infectious Disease, Imperial College London, London, United Kingdom. ✉e-mail: lars.schaefer@ruhr-uni-bochum.de; m.seeger@imm.uzh.ch

of TAGs. Conversely, operon overexpression led to elevated secretion of TAGs into the culture medium, hence offering additional evidence that Rv1410 is a TAG transporter[12]. LprG has also been associated with the surface display of lipoarabinomannans, which can be tetraacylated or, akin to TAG, triacylated[13–15]. Crystal structures of LprG revealed a hydrophobic pocket that is able to accommodate lipids with two or three alkyl chains[12,13] and in vitro experiments with purified non-acylated LprG that lacks its lipid anchor show that it is capable of transferring TAGs between lipid vesicles[12].

In mycobacteria, TAGs can be synthesized by several diacylglycerol acyltransferases[16] of which many have been predicted to localize to the inner membrane[17]. There, TAGs form lipid bodies and/or are transported to the outer membrane[18]. The molecular mechanism by which Rv1410 and LprG synergize to transport TAGs across the inner membrane and finally towards the mycomembrane is unclear. An LprG cross-linking study in live *Mycobacterium smegmatis* cells did not demonstrate a reproducible interaction between LprG and Rv1410[19] and attempts to show complex formation in vitro with purified proteins were fruitless[8]. Nevertheless, it is evident from several studies that for optimal functionality, both proteins working in concert are needed[8,20,21].

In this work, we determine the crystal structure of MHAS2168, an Rv1410 homologue from *Mycobacterium hassiacum*, in complex with an alpaca-raised nanobody. Based on the structure, we conduct molecular dynamics (MD) simulations as well as an extensive mutational analysis. Thereby, we reveal structural features essential for lipid transport, enabling us to propose a model describing the mechanism of TAG transport by Rv1410, including the role of LprG.

## Results

### Determination of MHAS2168 structure

To understand the lipid transport mechanism of Rv1410, we aimed to solve its structure. We finally succeeded to crystallize its close homologue MHAS2168 (sequence identity of 62%) from the thermophilic mycobacterial species *M. hassiacum*, purified from *M. smegmatis*, in

complex with an alpaca-derived nanobody Nb_H2 using crystallization in lipidic cubic phase (LCP). Native datasets diffracting up to 2.7 Å were obtained (Supplementary Table 1), but initial attempts to solve the structure by molecular replacement failed. Therefore, we used the megabody (Mb) approach[22] to enlarge Nb_H2 and determined a cryo-EM structure of the MHAS2168-Mb_H2 complex, resulting in a density map with an average resolution of 4 Å (Fig. 1a; Supplementary Fig. 1b; Supplementary Table 2; EMDB accession code: EMD-17787). In parallel, we analysed Rv1410 in complex with another megabody (Mb_F7) by cryo-EM, yielding a map of 7.5 Å (Supplementary Fig. 1a; Supplementary Table 2). The MHAS2168-Mb_H2 map enabled us to build an initial model with assigned side chains for the bulk of the transporter. We detected a non-proteinaceous density, most likely representing a lipid, at the transporter's side wall (Fig. 1a). Using this model, we could phase the native 2.7 Å data obtained by LCP crystallization by molecular replacement. The asymmetric unit comprises two transporter/nanobody complexes (Supplementary Fig. 1d). Complete models for the transporter and the nanobody were built with good refinement statistics and geometry (Supplementary Fig. 1c; Supplementary Table 1; PDB ID: 8PNL). In the following analysis, chains A (MHAS2168) and B (Nb_H2) are used due to better map quality. In fact, the electron densities for the backside of Nb_H2 in chain D were blurry and explain the comparatively poor RSRZ scores of the model (Supplementary Table 1).

### MHAS2168 architecture

Both cryo-EM and crystal structures show MHAS2168 in its outward-facing (OF) conformation (Fig. 1). MHAS2168 adopts a canonical MFS transporter fold, featuring an N- and a C-domain each composed of six transmembrane helices (TMs). Since Rv1410 and MHAS2168 belong to the drug:H⁺ antiporter-2 (DHA2) subclass of MFS transporters (http://www.tcdb.org), they possess additional transmembrane linker helices A and B (TMA and TMB) between TM6 and TM7. The linker helices form a hairpin (Fig. 1b, c), such as seen in the structure of NorC[23], as opposed to most other 14-helix

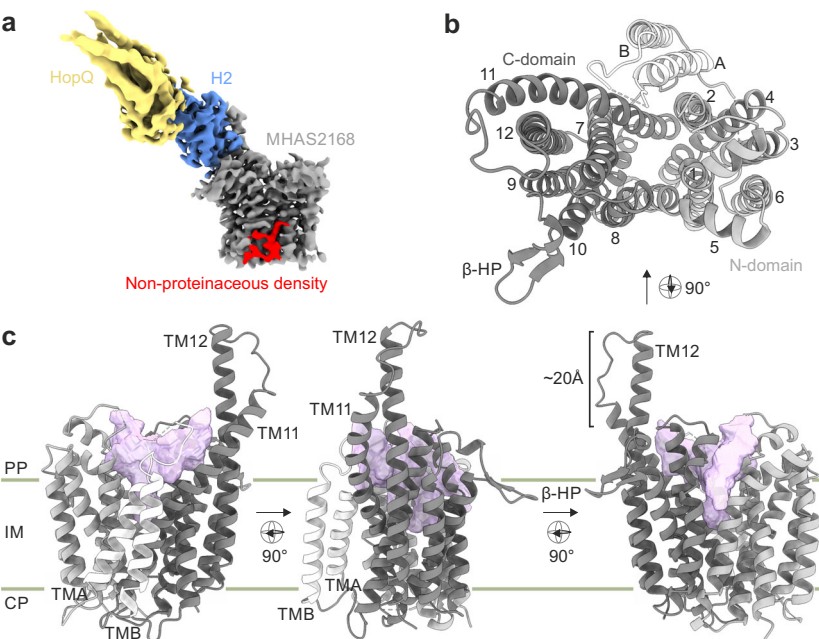

**Fig. 1 | Architecture of MHAS2168. a** 4 Å cryo-EM map of MHAS2168-Mb_H2 complex (EMDB ID: EMD-17787). MHAS2168 (gray), nanobody H2 (blue), megabody HopQ domain (yellow), nonproteinaceous density that likely corresponds to a lipid (red). **b** Periplasmic top view of MHAS2168 2.7 Å crystal structure (nanobody not depicted; PDB ID: 8PNL). N-domain, transmembrane helices 1–6 - light gray. Linker helices A and B - white. C-domain, transmembrane helices 7–12 - dark gray. **c** Side views of MHAS2168 2.7 Å crystal structure. Color scheme same as in (**b**). Central cavity volume - light purple. β-HP – β-hairpin; IM – inner membrane; PP – periplasm; CP – cytoplasm; TM – transmembrane helix.

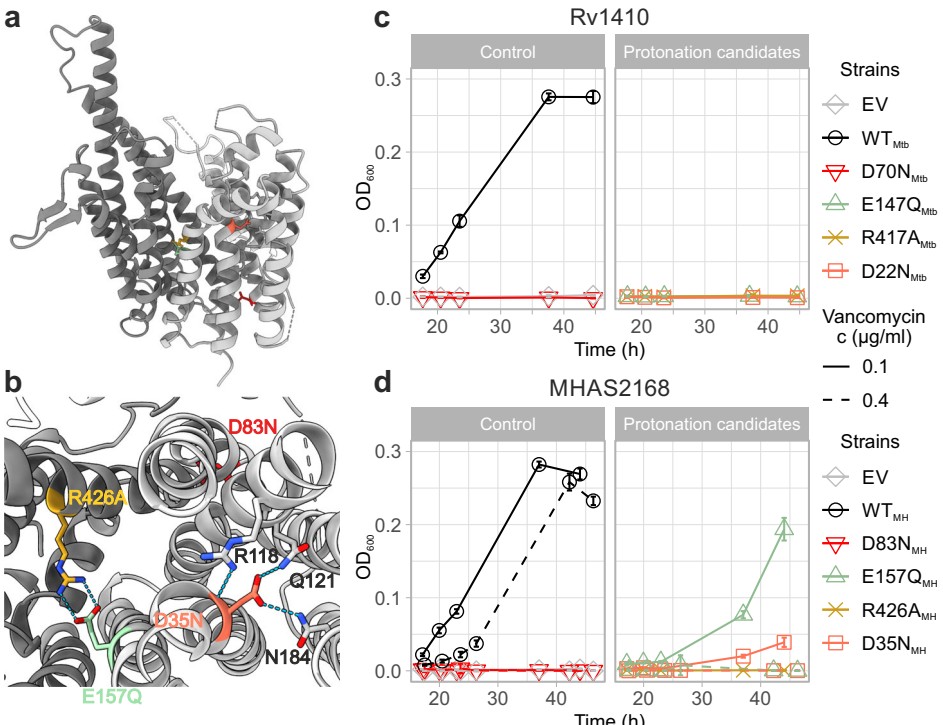

**Fig. 2 | Two candidate loci for proton translocation. a** Side view of MHAS2168. MHAS2168 color scheme same as in Fig. 1b. Point mutation sites are shown as colored sticks. **b** Enlarged periplasmic top view of MHAS2168 with mutated residues shown as colored sticks and interacting residues mentioned in the text as grey sticks, both with heteroatom depiction. Hydrogen bonds are shown as light blue dashed lines. **c** Vancomycin sensitivity assays in *M. smegmatis* dKO cells, complemented with empty vector control (EV), wild-type LprG/Rv1410 operon (WT$_{Mtb}$), or mutant operons containing unaltered LprG (Rv1411) and mutated transporter Rv1410 as indicated. **d** Analogous analysis as in (**c**), with *M. smegmatis* dKO cells expressing instead wild type MHAS2167/68 operon (WT$_{MH}$) or mutant operons containing unaltered LprG (MHAS2167) and mutated transporter MHAS2168 as indicated. The growth curves in (**c**) and (**d**) are representative of three biological replicates and data are presented as mean values +/- SD of four technical replicates.

MFS transporters whose linker helices are commonly arranged in a broader A-shape[24–27]. A conspicuous element of MHAS2168 is a ~20 Å long extension of TM11 and TM12 into the periplasm where we hypothesized the presence of a functionally important periplasmic loop in our previous work[8]. TM12 extends into the periplasm by 4 α-helical turns and is connected to the 2.5 α-helical turns-extended TM11 via a linker loop. Extended TM11 and TM12 seem to be a common feature of Rv1410 and its homologues, according to ColabFold[28] structure predictions (Supplementary Fig. 2) and the 7.5 Å cryo-EM map of Rv1410-Mb_F7 complex (Supplementary Fig. 1a). A similar extracellular element between TM11 and TM12 has been recently described in the structure of multi-drug efflux pump QacA from *Staphylococcus aureus*[29]. Finally, MHAS2168 features a small periplasmic β-hairpin between TM9 and TM10 (Fig. 1b, c), which extends along the membrane plane.

The ~2300 Å³ outward-facing central cavity of MHAS2168 is well-equipped to accommodate lipids, in that the cavity walls of both domains are very hydrophobic, even compared to other MFS lipid transporters LtaA[30,31] and MFSD2A[32–34] (Supplementary Fig. 3). Curiously, the cavity does not extend as deeply into the transporter as in the case of other MFS lipid or drug transporters (Supplementary Fig. 3) due to a constricting salt bridge (comprised of E157 on TM5 and R426 on TM11) and its neighbouring residues in the cavity (Fig. 1c; Fig. 2a, b). There is a continuity between the central cavity and the membrane space, via narrow lateral openings between the N- and C-domains (Fig. 1b, c). The narrow lateral crevice between TM5 and TM8 (TM5-TM8$^{OUT}$) is throughout its entire length 5-6 Å wide, while the would-be lateral cleft between TM2 and TM11 widens from 4 Å at the bottom to 12 Å at the top. However, the TM2-TM11$^{OUT}$ cleft is shielded from the

membrane by TMA and TMB. Therefore, it seems to be effectively blocked (Fig. 1b, c).

## Experimental system to assess functionality of Rv1410 and MHAS2168 mutants in *M. smegmatis*

Deletion of *lprG/rv1410c* homologous operon *MSM3070/69* in *M. smegmatis* results in increased vancomycin susceptibility[8,10,35,36], a phenotype we exploited here to study the transport activity of Rv1410 and MHAS2168 mutants (Supplementary Note 1). According to more recently published mechanistic models[8,12], deletion of *lprG/rv1410c* results in diminished TAG transport to the mycomembrane, which in turn becomes more permeable to vancomycin to reach its periplasmic target, namely peptidoglycan precursors, and results in cell death. Of note, since the physiological manifestations of *lprG/rv1410c* deletions are complex[10], other potential mechanisms not directly associated with TAG transport might also contribute to the vancomycin sensitivity phenotype. Complementation of the *M. smegmatis* deletion strain with *lprG/rv1410c* from Mtb or *MHAS2167/68* from *M. hassiacum* fully restores vancomycin resistance (Fig. 2c, d). Mutations in the transporter gene that cause loss of viability are therefore indicative of reduced transporter activity and TAG transport. Since complementation is carried out with the integrative pFLAG vector[37] where the transporter is fused to a FLAG tag, protein production was confirmed by Western blotting for every mutant assessed in this work (Supplementary Fig. 4a, b). As negative controls, empty vector or motif A aspartate mutants of the MFS transporters which destabilize the OF conformation[38,39] (D70N$_{Mtb}$ and D83N$_{MH}$) were used for complementation (Fig. 2c, d). Several Rv1410 mutants displaying negative phenotypes in the vancomycin sensitivity assays were in addition

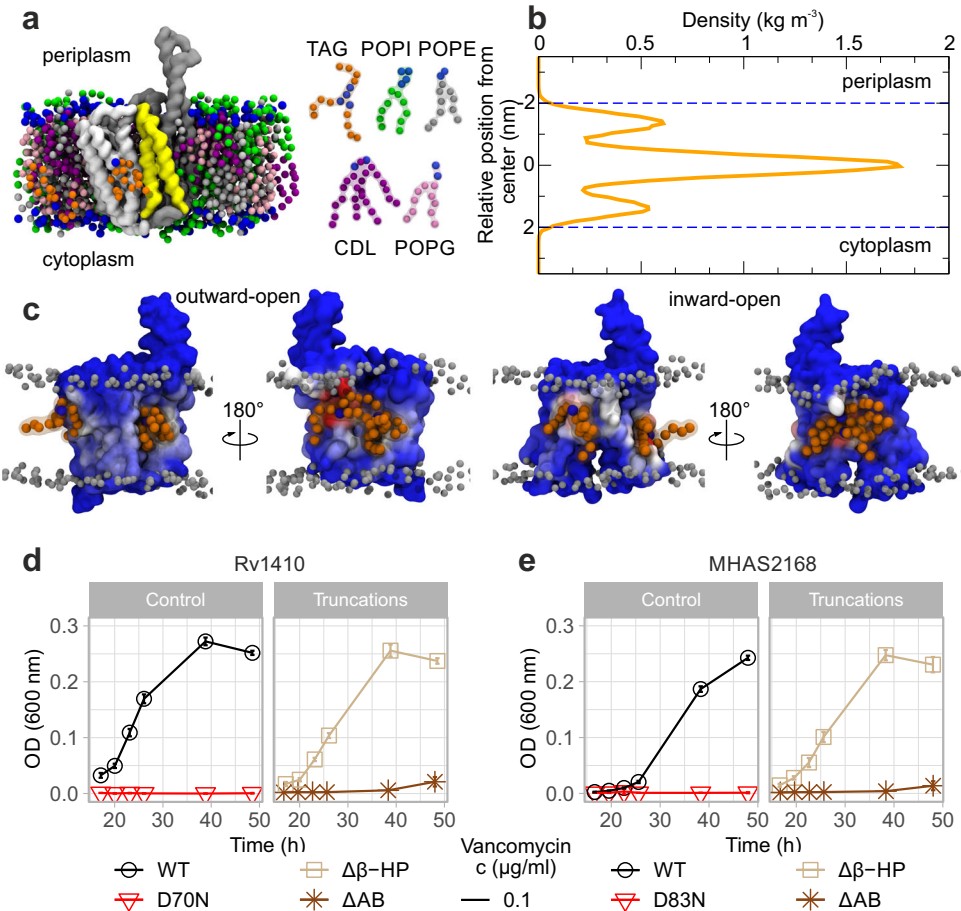

**Fig. 3 | Molecular dynamics simulations-guided mutational analysis of MHAS2168. a** The coarse-grained molecular dynamics simulation system of MHAS2168 in the outward-facing conformation. The mycobacterial plasma membrane lipid components are shown on the right, with headgroups colored blue. Solvent is not shown for clarity. **b** Density profile of TAGs along the membrane normal, centered with respect to the membrane midplane. The blue dotted lines are the average position of the phosphate headgroups. **c** The average number of TAG contacts from the MHAS2168$^{OUT}$ (left) and MHAS2168$^{IN}$ (right) simulations, respectively, projected on the surface of the protein and colored from blue (no contacts) to red (large number of contacts). **d** Vancomycin sensitivity assays in *M. smegmatis* dKO cells, complemented with wild type LprG/Rv1410 operon (WT$_{Mtb}$), or mutant operons containing unaltered LprG (Rv1411) and mutated transporter Rv1410 as indicated. **e** Analogous analysis as in (**d**), with *M. smegmatis* dKO cells expressing instead wild type MHAS2167/68 operon (WT$_{MH}$) or mutant operons containing unaltered LprG (MHAS2167) and mutated transporter MHAS2168 as indicated. Δβ-HP – β-hairpin truncation; ΔAB - truncation of linker helices TMA and TMB. The growth curves in (**d**) and (**e**) are representative of three biological replicates and data are presented as mean values +/- SD of four technical replicates.

purified to confirm their proper folding and membrane insertion (Supplementary Fig. 4c, d).

**Two candidate loci for proton translocation**

Being classified to the DHA2 subclass of MFS, Rv1410 is likely a TAG:H$^+$ antiporter. The only charged residues within the otherwise hydrophobic cavity of MHAS2168 are E157$_{MH}$ and R426$_{MH}$, which form a salt bridge (Fig. 2a, b) and constrict the bottom of the central cavity. Curiously, this ion pair is conserved in 17 mycobacterial homologues of Rv1410 (Supplementary Fig. 5) but is absent from 46 previously characterized MFS transporters (Supplementary Fig. 6). The configuration of the glutamate as a possible proton acceptor/donor, the positively charged arginine to control pKa changes of the glutamate, and their location in the solvent-accessible central cavity suggest a role of this ion pair in proton translocation[40].

To test whether the ion lock residues are important for transporter function, E157$_{MH}$ from MHAS2168 and the corresponding E147$_{Mtb}$ from Rv1410 were mutated to glutamine. Similarly, R426$_{MH}$ and R417$_{Mtb}$ were mutated to alanine. Both E147Q$_{Mtb}$ and R417A$_{Mtb}$ mutations inactivated the transporter completely in the vancomycin sensitivity assay (Fig. 2c). Similarly, the R426A$_{MH}$ mutant was unable

to grow, but the E157Q$_{MH}$ mutant showed completely abolished growth only under increased vancomycin stress (0.4 μg/ml) (Fig. 2d).

Another conserved carboxylate (Supplementary Fig. 5), D22 in Rv1410, has been shown to be required for transport before[20]. D22$_{Mtb}$ and corresponding D35$_{MH}$ are located in the middle of TM1 where their side chains point towards the center of the N-domain. D35$_{MH}$ forms hydrogen bonds with Q121$_{MH}$ (TM4, 1.8 Å), N184$_{MH}$ (TM6, 2.4 Å), and R118$_{MH}$ (TM4, 3.4 Å) (Fig. 2b). D35$_{MH}$ and the neighbouring R118$_{MH}$ are potentially implicated in proton translocation due to analogously located aspartate-arginine salt bridges, which were identified as the protonation/deprotonation sites via which substrate transport is coupled to proton translocation in certain sugar/H$^+$ symporters[41–44]. In agreement with previously investigated mutants D22A$_{Mtb}$ and D22E$_{Mtb}$[20], mutations D22N$_{Mtb}$ and D35N$_{MH}$ compromised the functionality of the transporter (Fig. 2c, d), suggesting that protonation/deprotonation of this carboxylate is necessary for transport function. While in the outward-open conformation of MHAS2168 the D35$_{MH}$-R118$_{MH}$ pair is not solvent-accessible, tunnels providing access to the bulk solvent might be present in other conformations. In summary, our data suggest that in Rv1410, both D22 and E147 likely play a role in coupling the energy stored in the proton gradient to the efflux of TAGs.

## Molecular dynamics simulations suggest a mechanism of TAG extraction from the hydrophobic core of the membrane

To gain detailed molecular-level insights into how the transporter interacts with TAG substrate molecules, we performed coarse-grained MD simulations of MHAS2168 in both outward-open (MHAS2168$^{OUT}$; our crystal structure) and inward-open conformations (MHAS2168$^{IN}$; homology model based on PepT$_{So2}$) embedded in a phospholipid bilayer doped with TAGs, mimicking the mycobacterial plasma membrane (Fig. 3; Supplementary Table 3). The MD simulations revealed that TAGs are not embedded in the individual leaflets of the lipid bilayer, but rather segregate to the hydrophobic core of the membrane (Fig. 3b) as previously reported[45,46].

TAGs probe a range of different positions along the transporter TM helices, with preference for TM8 and TM10, between the β-hairpin and the TM5-TM8 lateral clefts (Fig. 3c; Supplementary Fig. 7). Linker helices TMA and TMB and their surroundings form another hotspot for transporter-TAG interactions, especially in MHAS2168$^{IN}$. To test the importance of these features to TAG transport, either TMA and TMB or the TM9-TM10 β-hairpin were truncated in Rv1410 and MHAS2168. Surprisingly, the loss of the β-hairpin did not affect lipid transport in neither Rv1410 nor MHAS2168. In contrast, the deletion of the linker helices resulted in inactivation of both transporters while not affecting their folding (Supplementary Fig. 4c, d), indicating that they are key to TAG transport (Fig. 3d, e).

Unlike phospholipids, TAGs were not observed to enter the central cavity of the inward-facing (IF) transporter (Supplementary Note 2; Supplementary Fig. 8). However, compared to the OF conformer, the IF conformer has more contacts with TAG (Supplementary Fig. 7) and a larger central cavity (Supplementary Fig. 9). To probe whether the central cavity can accommodate TAGs, we performed molecular docking of flexible TAG on both conformers of the transporter, thereby showing that the inward-facing main cavity is a preferred TAG binding site (Supplementary Fig. 10). Subsequently, we placed coarse-grained TAG into the inward-facing central cavity to perform another round of MD simulations (MHAS2168$^{IN-TAG}$, see Supplementary Table 3). During the simulations, TAG exits from the cavity, with residence times varying from 8 to 74 μs between the different repeats. The TAG molecule consistently escapes via the TM5-TM8$^{IN}$ lateral cleft (Supplementary Note 2; Supplementary Fig. 11; Supplementary Table 4), which could therefore also provide a possible entrance pathway.

## Mutational analysis indicates TAG entry into inward-facing cavity

Reflecting the hydrophobic nature of the TAG as substrate, the central cavity of OF Rv1410 is particularly apolar (Supplementary Fig. 3). Its polarity was increased by substituting a conserved leucine, which is situated in the middle of the hydrophobic C-domain cavity wall (Fig. 4c, d), with arginine or aspartate. Interestingly, only the L289R$_{Mtb}$ and L299R$_{MH}$ mutants were inactive; the L289D$_{Mtb}$ and L299D$_{MH}$ mutants behaved like the wild-type control strains at both lower (0.1 μg/ml) and higher (0.4 μg/ml) vancomycin concentrations (Fig. 4e, f). This discrepancy might be explained by the fact that arginines retain their charge in lipid and protein environments better than carboxylates[47], thus preventing TAG binding to the central cavity.

According to our MD simulations, TAGs accumulate in the membrane core and therefore must enter the transporter through lateral openings between N- and C-domains lined with TM5 and TM8, or TM2 and TM11 in either the IF or OF conformation, respectively (Fig. 4a, b). To test whether any of the four lateral crevices could serve as TAG entry or exit sites, residues in the middle of each lateral cleft were mutated into glutamates or aspartates to introduce polarity and block the opening (Fig. 4a, b, c; see Supplementary Note 1).

Functional assays (Fig. 4e, f) suggest that TAGs enter the cavity through both TM5-TM8$^{IN}$ and TM2-TM11$^{IN}$ lateral openings, as the

corresponding mutants G140D$_{Mtb}$, G150D$_{MH}$, A411D$_{Mtb}$, and A420D$_{MH}$ were inactive. As expected due to the obstruction by the linker helices, the TM2-TM11$^{OUT}$ lateral crevice is not involved in lipid transport as the L422E$_{Mtb}$ and L431E$_{MH}$ mutants display comparable growth to the wild-type operons at both lower (0.1 μg/ml) and higher (0.4 μg/ml) vancomycin concentrations. Likewise, mutations in the TM5-TM8$^{OUT}$ lateral cleft (namely L155E$_{Mtb}$ and I165E$_{MH}$) were well tolerated at 0.1 μg/ml vancomycin concentration, suggesting that this cleft does not seem to play a role in TAG transport despite the fact that it is connecting the central cavity and membrane (Fig. 4e, f).

## Periplasmic TM11 and TM12 extensions guide TAG transfer into LprG

It is clear from the MHAS2168 structure (Fig. 1c; Fig. 5a) and structure predictions of its homologues (Supplementary Fig. 2) that the "periplasmic loop" truncations we previously investigated in Rv1410[8] manifest in fact as helix truncations (Fig. 5a), resulting in a less extended TM12. Two previously studied Rv1410 truncation mutants were therefore engineered into MHAS2168, utilizing structure predictions performed with ColabFold platform[28] (see Supplementary Note 1; Fig. 5a). Both MHAS2168 truncation mutants failed to transport lipids, while the Truncation 1$_{Mtb}$ mutant retained some of its activity and the Truncation 2$_{Mtb}$ mutant did not (Fig. 5b, c).

An in-depth analysis of the MHAS2168 structure and the structure prediction models of its mycobacterial homologues revealed that although the primary structure of the TM11-TM12 extensions varies in both length (31–38 residues) and sequence, they share common features (Supplementary Fig. 2; Supplementary Fig. 5). Firstly, the TM12 length is very conserved (extra 4 α-helical turns) while there are differences in the TM11 and TM11-TM12 loop lengths (Supplementary Fig. 2a). Secondly, hydrophobic patches are present on the side of the extensions facing the cavity (Supplementary Fig. 2b–d), and the residues on the tip of the TM12 extension (1st α-turn) are commonly hydrophobic. Thirdly, aromatic residues are prevalent on TM12 above the cavity.

To study the functional role of these molecular features on the TM12 extension, single point mutations were generated with the aim to alter the biophysical properties of this region (Fig. 5a). Therefore, the aromatic or electroneutral residues of TM12 above the cavity were mutated either to alanines to remove phenol groups that might be involved in stacking interactions (Y464A$_{Mtb}$, F468A$_{Mtb}$, T479A$_{MH}$, Y483A$_{MH}$) or to glutamates to introduce charge and bulkiness to the side chains and thus likely disrupt lipid movement (Y464E$_{Mtb}$, F468E$_{Mtb}$, T479E$_{MH}$, Y483E$_{MH}$). Similarly, the hydrophobic residues at the tip of TM12 extension were mutated into lysines (L453K$_{Mtb}$, M468K$_{MH}$) or aspartates (L453D$_{Mtb}$, M468D$_{MH}$). While all of these mutants retained wild-type activity in the vancomycin sensitivity assays under milder condition (0.1 μg/ml vancomycin; Fig. 5b, c), growth defects manifested in many of these mutant strains at higher vancomycin concentration (0.4 μg/ml; Supplementary Fig. 12c, d). In conclusion, the helical extensions cannot be removed without affecting the transporter's function. Further, disruption of conserved molecular features on TM12 results in partially defective transport activity.

We hypothesized that the TM11 and TM12 periplasmic extensions might play a role in TAG exit from the central cavity by serving as an anchor point to place LprG into a favourable position for TAG transfer. Although previous experiments failed to demonstrate physical interactions between purified Rv1410 and LprG[8], transient low-affinity interactions nevertheless may occur in the cellular context. We reasoned that if there was a specific physical interaction between the extended helices of Rv1410 and LprG, the operon partners might have co-adapted during evolution. Then, MFS transporters and lipoproteins from different mycobacterial species would not be able to act in concert. To examine this hypothesis, Rv1410 and LprG from *M.*

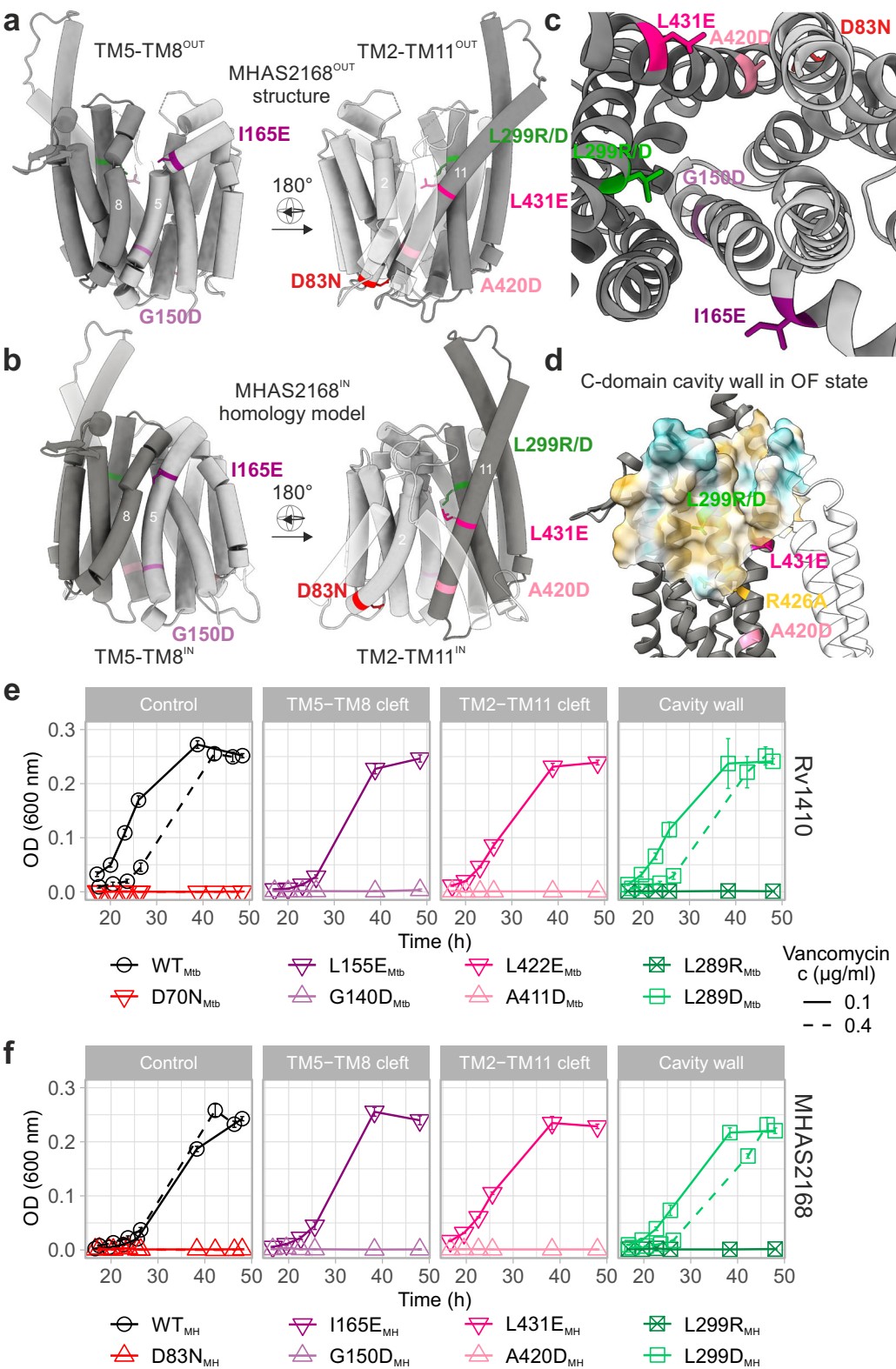

tuberculosis were shuffled with their counterpart homologues from three mycobacterial species (namely *M. smegmatis*, *M. hassiacum*, and *M. abscessus*) without affecting the overall operon structure. *M. smegmatis* dKO complemented with any tested combination of transporter and lipoprotein was able to grow as fast as with the natively paired operons while complementation with the transporters (with the exception of MAB2807 from *M. abscessus*) or lipoproteins alone resulted in substantial growth defects (Fig. 5d–h). These findings

explain the lack of a strong specific interaction between transporter and lipoprotein and rather suggest transient interactions to occur between the operon partners.

Alternatively, the importance of periplasmic helix extensions in TAG transport might lie in their interactions with the TAG during its transit from Rv1410 to LprG. Hence, we decided to study the dynamic process of TAG transfer from the transporter into LprG by MD simulations. The ColabFold platform[28] was used in multimer mode to build

**Fig. 4 | Potential TAG entry sites to the central cavity and cavity analysis. a** Side view of lateral clefts TM5-TM8$^{OUT}$ (left) and TM2-TM11$^{OUT}$ (right) in outward-facing conformation of MHAS2168 crystal structure. MHAS2168 color scheme same as in Fig. 1b. Point mutation sites are shown as colored sticks. Linker helices A and B are depicted partially transparent. **b** Side view of lateral clefts TM5-TM8$^{IN}$ (left) and TM2-TM11$^{IN}$ (right) in inward-facing conformation of MHAS2168 homology model. MHAS2168 and point mutation sites' color scheme same as in panel **a**. **c** Enlarged periplasmic top view of the central cavity and lateral clefts in outward-facing conformation of MHAS2168 crystal structure. MHAS2168 and point mutation sites' color scheme same as in panel **a**. **d** Hydrophobicity surface of the C-domain central

cavity wall of MHAS2168 outward-facing crystal structure with mutated residues shown as colored sticks. Hydrophobicity color scheme: hydrophobic – gold; hydrophilic – cyan. **e** Vancomycin sensitivity assays in *M. smegmatis* dKO cells, complemented with wild type LprG/Rv1410 operon (WT$_{Mtb}$), or mutant operons containing unaltered LprG (Rv1411) and mutated transporter Rv1410 as indicated. **f** Analogous analysis as in (**e**), with *M. smegmatis* dKO cells expressing instead wild type MHAS2167/68 operon (WT$_{MH}$) or mutant operons containing unaltered LprG (MHAS2167) and mutated transporter MHAS2168 as indicated. The growth curves in (**e**) and (**f**) are representative of three biological replicates and data are presented as mean values +/- SD of four technical replicates.

a complex of the MHAS2168$^{OUT}$ structure and an AlphaFold model of *M. hassiacum* LprG. All five obtained models consistently positioned LprG on the periplasmic side, with its hydrophobic cavity oriented towards TM11 and TM12 extending into the periplasm (Fig. 6; Supplementary Fig. 13a). A TAG molecule was inserted into the central cavity of the transporter and unbiased coarse-grained MD simulations of the MHAS2168$^{OUT}$-LprG complex were carried out. Several events of TAG transfer from the MHAS2168$^{OUT}$ central cavity into the LprG pocket were observed, at a rate of about one event per 100 µs of simulation time (Fig. 6). In addition to the transfer of TAG into LprG, also the reverse transit process was observed (Fig. 6a; Supplementary Movie 1), suggesting a rather shallow energy landscape. Interestingly, in our simulations, the TAG molecule migrated into the LprG cavity only when two of its acyl tails were pointing up towards LprG, instead of only one (Supplementary Fig. 14). The key residues of the transporter that mediate the transfer of TAG between the transporter central cavity and the LprG pocket are found in the loop connecting the linker helices TMA and TMB, the C-terminus of TM7, and particularly TM11 and TM12 (Fig. 6b, c; Supplementary Fig. 13; Supplementary Table 4). For LprG, the residues that guide TAG transfer cover its entrance mouth and form a broad hydrophobic surface that extends further into the hydrophobic cavity of LprG (Supplementary Fig. 13; Supplementary Table 4).

## Discussion

In this work, we provide structural and mechanistic insights into the mycobacterial MFS transporter Rv1410, which has been shown previously to work together with its operon partner LprG to transport triacylated lipids such as TAGs to the mycomembrane. We wish to note that that Rv1410 and LprG possibly fulfil additional, yet undiscovered functions, by transporting further lipid-like substrates that might have escaped detection in a previous lipidomics study[12].

We found that TAGs do not form an orderly part of the two leaflets of the cytoplasmic membrane, but instead accumulated close to the membrane mid-plane. From a biophysical point of view, the main energy barriers TAG faces during its journey from the core of the inner membrane to the mycomembrane are the crossing of the charged outer leaflet of the inner membrane and the polar environment of the periplasm.

Our proposed model of transport (Fig. 7) accounts for this in that Rv1410 and LprG provide a continuous series of hydrophobic cavities and surfaces to shield TAG from the bulk water and thus allow for facilitated passage from the inner membrane core to the lipid binding cavity of LprG. Our MD simulations suggest that the precise positioning of MHAS2168 relative to LprG is not important for the TAG transfer process to take place, as long as the TAG is sufficiently shielded from the aqueous phase (Fig. 5; Fig. 6; Supplementary Fig. 14; Supplementary Fig. 15).

Our functional data suggest that the TAG molecule enters the hydrophobic cavity of inward-facing Rv1410 from the cytoplasmic membrane either via the TM5-TM8$^{IN}$ or the TM2-TM11$^{IN}$ lateral opening (step 1 in Fig. 7). However, TAG was never observed to spontaneously enter into the IF cavity on the MD-simulation time scale, which could be due to limitations of the homology model of the IF conformer, in

particular concerning the precise widths of the lateral openings. In contrast, exit events were found in the MHAS2168$^{IN-TAG}$ simulations at the TM5-TM8$^{IN}$ cleft, thus mapping a potential pathway for TAG to exit (or, reversely, to enter) the inward-facing cavity (Supplementary Fig. 11).

Once TAG enters the cavity, Rv1410 transits from IF to OF conformation (step 2 in Fig. 7). Whether this transition is driven by TAG binding or protonation/deprotonation events, is not deducible from our current data. As an effect, the TAG binding cavity is remodelled such that it opens to the periplasm and the E147-R417 ion lock is formed, which is a hallmark and essential functional element of Rv1410 and its mycobacterial homologues. This constriction pushes the TAG away from the membrane core to the level of the outer leaflet, while the hydrophobic cavity shields it from the charged environment of the lipid head groups. Thus, the TAG is enclosed in the transporter and might be retained within the central cavity in the OF conformation because the TM5-TM8$^{OUT}$ lateral cleft is narrow or even closed (Supplementary Fig. 2b) and the TM2-TM11$^{OUT}$ crevice is blocked by linker helices TMA and TMB. The linker helices of Rv1410 are crucial for its function.

As the last step of our transport pathway model (step 3 in Fig. 7), TAG leaves the central cavity via the opening to the periplasm. The TAG remains concealed from the hydrophilic environment from one side by the TM11 and TM12 periplasmic extensions and from the other side by LprG which captures the lipid into its hydrophobic pocket. This final TAG transfer step is experimentally supported by the fact that TM12 truncations result in complete loss of transport activity, without disturbing transporter folding and production. A similar loss of substrate transport has been observed in the multi-drug efflux pump QacA when its extracellular element located between TM11 and TM12 was deleted[29], suggesting a more general functional relevance for structural features inserted between TM11 and TM12 in DHA2 subclass of MFS transporters.

Once TAG has left Rv1410, our model proposes that the proton gradient is exploited to revert to the IF conformation and a new transport cycle can begin. We discovered two functionally important carboxylate-arginine pairs in Rv1410, namely D22-R108 placed within the N-domain and E147-R417 placed across the N- and C-domain in the central cavity, whose likely function is to couple proton influx to the export of TAG. It should be noted that the substitutions of the key carboxylates by the respective carboxamides did not fully abrogate transport function in MHAS2168 (Fig. 2c, d). This might be explained by the fact that TAGs are transported along their concentration gradient, as they are produced within the cell and finally dilute out to the mycomembrane and, to some extent, the growth medium. Hence, the transporter might well be able to facilitate TAG export without proton coupling, as has been shown in the case of several MFS transporters and their substrates[41,48]. However, the transport rate likely increases when lipid export is coupled to the proton gradient. The fact that the R426A$_{MH}$ mutation is more deleterious than E157Q$_{MH}$, might be due to an absolute requirement for the structural interaction at the base of the OF cavity between the two residues. Q157$_{MH}$ is still able to form hydrogen bonds with R426$_{MH}$, which is not possible for A426$_{MH}$ with E157$_{MH}$.

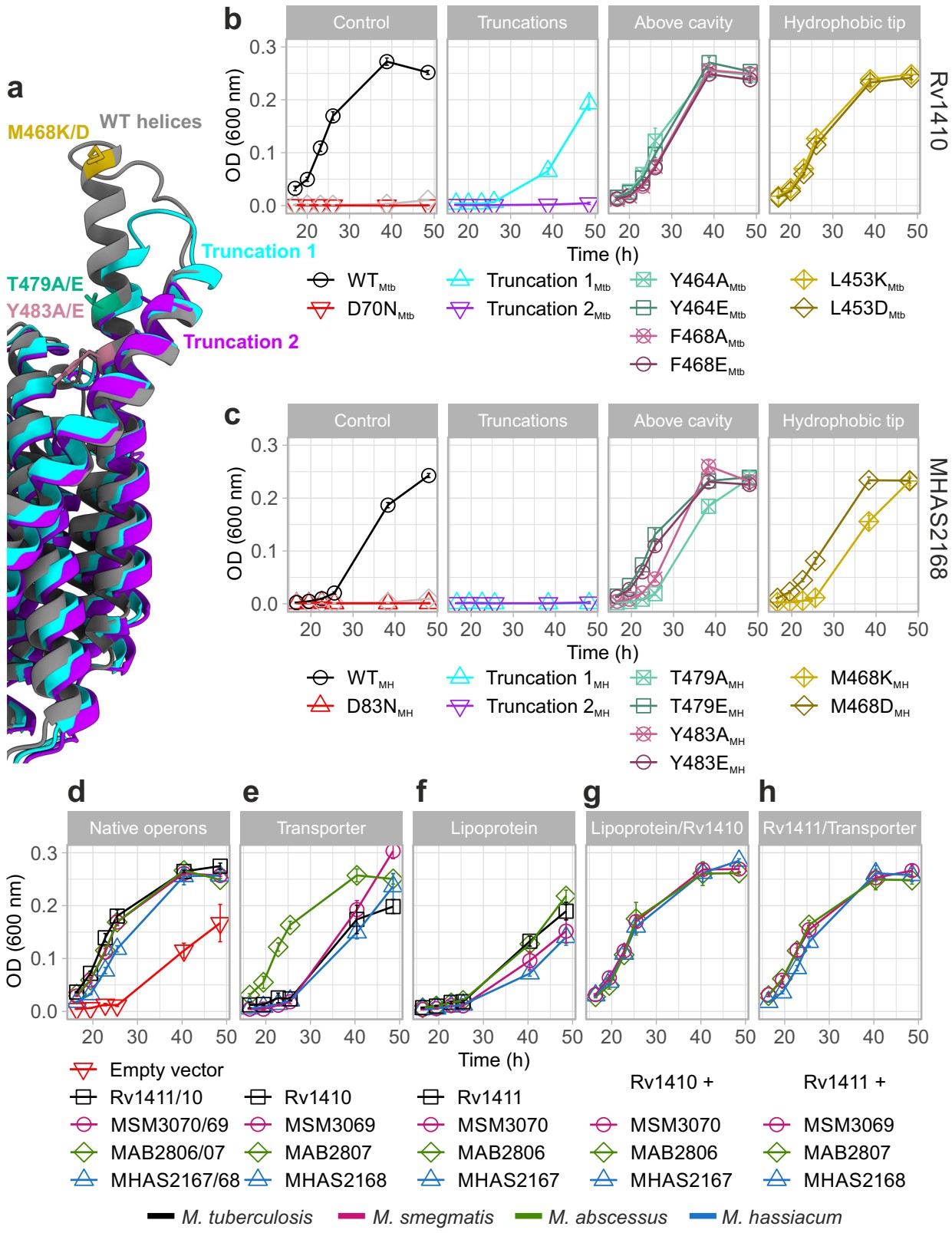

Extraction of lipopolysaccharides or lipoproteins from the outer leaflet of the inner membrane of gram-negative bacteria requires an active transport step mediated by the essential ABC transporters LptB$_2$FG and LolCDE, respectively[49–51]. These lipid extractors belong to the type VI and type VII ABC transporters[52], respectively, and utilize the energy of ATP binding and hydrolysis to pass their hydrophobic cargo to the dedicated periplasmic carrier protein. In this work, we describe a

structure of an MFS transporter capable of lipid extraction, which likely uses the proton-motive force to energize transport.

Recently, structural models of two other MFS lipid transporters have been made available: LtaA[30] and MFSD2A[32–34]. LtaA is a H⁺/lipo-teichoic acid antiporter from *S. aureus*[30] and MFSD2A is a Na⁺-dependent lysophosphatidylcholine-docosahexaenoic acid importer in the blood-brain barrier in humans[34,53]. Unlike Rv1410, LtaA and MFSD2A

**Fig. 5 | TM11 and TM12 periplasmic extensions and transporter-lipoprotein interactions. a** Side view of MHAS2168^OUT crystal structure and predicted structures of truncation mutants 1 (purple) and 2 (light blue) of TM11-TM12 periplasmic extensions. MHAS2168 color scheme same as in Fig.1b. Point mutation sites are shown as colored sticks. **b** Vancomycin sensitivity assays in *M. smegmatis* dKO cells, complemented with wild-type LprG/Rv1410 operon (WT_Mtb), or mutant operons containing unaltered LprG (Rv1411) and mutated transporter Rv1410 as indicated. **c** Analogous analysis as in (**b**), with *M. smegmatis* dKO cells expressing instead wild type MHAS2167/68 operon (WT_MH) or mutant operons containing unaltered LprG (MHAS2167) and mutated transporter MHAS2168 as indicated. **d–h** Vancomycin sensitivity assays in *M. smegmatis* dKO cells, complemented with different combinations of the transporter and/or lipoprotein from four mycobacterial species

(*M. tuberculosis*; *M. smegmatis*; *M. abscessus*; *M. hassiacum*). **d** Complementation with native operons or empty vector. **e** Complementation with only the transporter. **f** Complementation with only the lipoprotein. **g** Complementation with a shuffled operon in which the transporter from *M. tuberculosis* (Rv1410) is accompanied by lipoprotein from the other three mycobacterial species. **h** Complementation with a shuffled operon in which the lipoprotein from *M. tuberculosis* (Rv1411) is accompanied by transporter from the other three mycobacterial species. Vancomycin sensitivity assays were carried out at 0.1 µg/ml concentration on panels **b** and **c** or at 0.08 µg/ml concentration on panels **d–h**. The growth curves in (**b–h**) are representative of three biological replicates and data are presented as mean values +/- SD of four technical replicates.

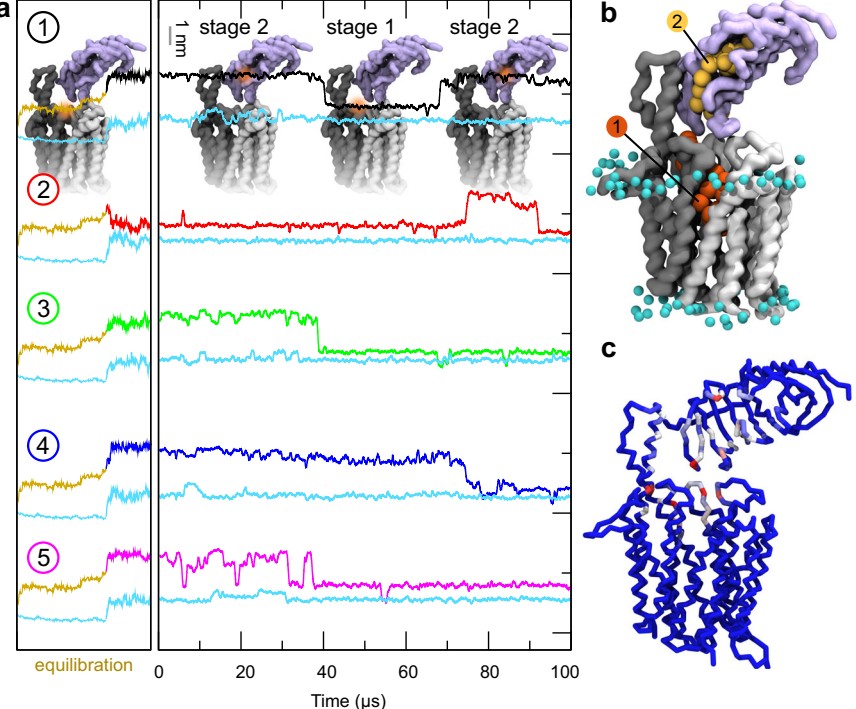

**Fig. 6 | TAG transfer into LprG in the MD simulations of the MHAS2168^OUT·LprG complex. a** For each of the five simulations (numbered 1–5), the upper line depicts the z-coordinate of the center of mass of the TAG molecule. The cyan lines below correspond to the z-coordinate of the average center of mass of the phosphate groups of the upper membrane leaflet. The gold line is the equilibration phase (see methods). The models of the MHAS2168^OUT·LprG complex shown in the top panel

indicate the position of the TAG (orange shade). **b** Visualization of TAG transfer from the transporter's main cavity (stage 1) along the TM11-TM12 extension into the LprG hydrophobic cavity (stage 2). Note that the overlay of structures shown displays the position of one single TAG molecule at two different points in time during the MD simulation. **c** TAG contacts with MHAS2168^OUT·LprG, coloured from blue (no contacts) to red (large number of contacts).

display amphipathic central cavities, reflecting the greater polarity of their substrate lipids' headgroups (Supplementary Fig. 3). Based on MD simulations and cross-linking studies[31] in which the lateral crevices are locked by disulfide bridges, it has been proposed that lipoteichoic acid enters LtaA cavity from the inner leaflet of cytoplasmic membrane through the TM5-TM8^IN lateral cleft and exits through either TM5-TM8^OUT or TM2-TM11^OUT lateral crevice to the outer leaflet, having been flipped. In MFSD2A, non-proteinaceous densities possibly corresponding to lipids have been observed in both TM5-TM8^OUT and TM2-TM11^OUT lateral clefts[33]. It is very likely that lipids enter the cavity of MFSD2A, as lipid density has been observed within the cavity in the IF state[32] and a suitable hydrophobic pocket for accommodating long aliphatic chains has been discovered in the occluded state of MFSD2A[34]. Whether analogous side pocket(s) form within Rv1410 for the concealment of TAG lipid tails remains unknown until structures of its other conformations are obtained.

In contrast to LptB_2FG and LolCDE which both directly interact with their cognate periplasmic proteins LptC[49] and LolA[54], respectively,

LprG and Rv1410 do not appear to engage in strong and specific protein-protein interactions, as shown by the functional operon shuffling experiments (Fig. 5d–h) and previously described biochemical studies[8]. The Rv1410-LprG heterodimer appears to be rather short-lived or stabilized by the transient presence of TAG at the interface of the two proteins. We therefore surmise an interplay in which LprG scans the periplasmic surface of the inner membrane while being attached via its lipid anchor until it encounters Rv1410 presenting a TAG molecule, ready to be captured into the lipid binding cavity of LprG. Whether LprG passes TAG onto further periplasmic proteins or whether it can be extracted itself from the inner membrane to channel TAG to the mycomembrane is currently unclear and requires further investigation.

## Methods

### Bacterial strains, media and plasmids

In this study, *Escherichia coli* strains DB3.1 and MC1061 were used for cloning and Rv1410, nanobody, and megabody expression, *M.*

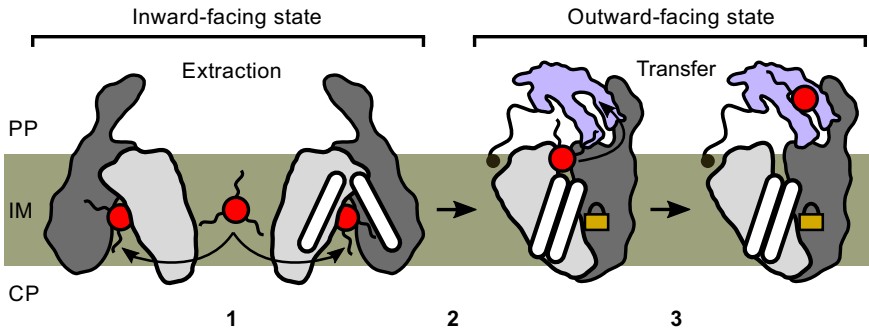

**Fig. 7 | Mechanism of TAG transport by Rv1410 and LprG.** Rv1410/MHAS2168 color scheme the same as in Fig. 1b. LprG is colored pale lilac. (1) TAG molecule (red) enters the transporter's central cavity through lateral openings between TM5-TM8$^{IN}$ and TM2-TM11$^{IN}$ in the inward-facing conformation. (2) The transporter transitions from inward-facing to outward-facing state while the TAG molecule is occluded within the central cavity. The E147-R417 ion lock (symbolized by golden lock) at the bottom of the central cavity is formed and lifts TAG toward the periplasmic leaflet. (3) TM11 and TM12 periplasmic extensions shield the TAG molecule from hydrophilic periplasm while the TAG molecule relocates from the transporter's cavity to the hydrophobic pocket of LprG.

smegmatis MC$^2$ 155 harboring pACE_C3GH_MHAS2168 was used for MHAS2168 protein expression and *M. smegmatis* MC$^2$ 155 Δ*MSMEG3069/70* (dKO) was used for complementation studies. Mycobacteria were grown at 37 °C in liquid Middlebrook 7H9 medium containing 0.05% Tween 80 supplemented with OADC or on solid Middlebrook 7H10 medium supplemented with OADC containing 4.5 ml/l glycerin. For MHAS2168 expression, 7H9 medium supplemented with 0.05% Tween 80 and 0.2% glycerol was used. *E. coli* was grown in lysogeny broth medium (LB) or Terrific broth medium (TB) at 37 °C or 25 °C respectively. Where required, the liquid medium was supplemented with the following amounts of antibiotics: 100 μg/ml of ampicillin (Amp$^{100}$) and 25 μg/ml of chloramphenicol (Cm$^{25}$) for *E. coli*, 50 μg/ml apramycin (Apr$^{50}$) for *E. coli* and *M. smegmatis*, 50 μg/ml hygromycin B (Hyg$^{50}$) for *E. coli* and *M. smegmatis*. Solid LB medium was supplemented with 120 μg/ml of ampicillin (Amp$^{120}$), 20 μg/ml of chloramphenicol (Cm$^{20}$), 50 μg/ml apramycin (Apr$^{50}$) or 100 μg/ml hygromycin B (Hyg$^{100}$). 7H10 medium was supplemented with 50 μg/ml apramycin (Apr$^{50}$) or 50 μg/ml hygromycin B (Hyg$^{50}$).

### Construction of plasmids

**Shuffled *lprG-mfs* plasmids.** The pFLAG plasmids harboring wild-type *lprG-mfs* operons and its single genes from *M. tuberculosis*, *M. smegmatis*, and *M. abscessus* originated from our previous study[8]. The same FX cloning strategy[37] was at first used for *M. hassiacum lprG-mfs* operon *MHAS2167/68* and its single genes, using primers from Supplementary Table 5 and *M. hassiacum* strain DSM 44199 as a template for colony-PCR with Q5 High-Fidelity DNA polymerase (NEB) to generate a fragment cloned into initial vector pINIT (Cm$^{25}$). However, since the *MHAS2167* gene contains a SapI cleavage site with identical overhang (AGT), it was inefficient to use FX cloning to transfer the gene products to pFLAG vector. Thus, pFLAG plasmids containing *MHAS2167* and *MHAS2167-rv1410c* operon were generated using CPEC[55] (primers in Supplementary Table 5).

Using the pINIT plasmids with wild-type *lprG-mfs* operons as templates, shuffled operons were assembled (with primers in Supplementary Table 5) using CPEC protocol[55]. The new operon sequences were confirmed by Sanger sequencing and then the shuffled operons were transferred to pFLAG vector, using FX cloning[37].

**Rv1410 and MHAS2168 mutants in pFLAG vector.** First, QuikChange site-directed mutagenesis protocol was used to introduce mutations into Rv1410 in pFLAG_Rv1410/11 or in MHAS2168 in pFLAG_MHAS2167/68, using primers (named mutation_FOR and mutation_REV) in Supplementary Table 6. However, in many cases, it was difficult to obtain the necessary PCR products, presumably due to the length of the vector (~6100–6300 bp) and GC content of the

operons (64-65%). Therefore, CPEC protocol[55] was used to assemble the constructs from two fragments. In each case, a larger "backbone" fragment was amplified using Q5 High-Fidelity DNA polymerase (NEB) with primers pFLAG_REV2 and mutation_FOR and a smaller "insert" fragment was amplified with primers pFLAG_FOR2 and mutation_REV (Supplementary Table 6). Both QuikChange products and CPEC reaction products were transformed into *E. coli* MC1061, plasmids were extracted using a QIAprep Spin Miniprep Kit (QIAGEN) and the correct sequences were confirmed by Sanger sequencing.

**Expression plasmids.** The *MHAS2168* gene was transferred from pINIT_*MHAS2168* to the pACE_C3GH[37] and *rv1410c* gene from pINIT_*rv1410c* to the pBXC3GH[56] using FX cloning, resulting in pACE_C3GH_*MHAS2168* and pBXC3GH_*rv1410c* in which the *mfs* transporters are C-terminally fused to a 3C protease cleavage site, GFP and a His$_{10}$-tag. Genes of Rv1410 mutants D70N, ΔAB, E147Q, D22N, A411D, and L289R were amplified with primers Rv1410c_for and Rv1410c_rev (Supplementary Table 6) using corresponding pFLAG_Rv1410_mutant vectors as templates and then transferred consecutively into pINIT (in which the mutation was confirmed by Sanger sequencing) and pBXC3GH vectors by using FX cloning.

The gene encoding nanobody H2 (Nb_H2) was transferred from pSb_init_Nb_H2 to pBXNPHM3[57], resulting in pBXNPHM3_Nb_H2 in which the nanobody's N-terminus is fused to the PelB signal peptide, His$_{10}$-tag, maltose binding protein and a 3C protease cleavage site. To turn nanobody Nb_H2 into a megabody MB_H2 and Nb_F7 into Mb_F7, the genes encoding Nb_H2 and Nb_F7 and lacking the first thirteen N-terminal residues were transferred to pBXMBQ vector (a kind gift from Eric Geertsma and Benedikt Kuhn), using FX cloning. The resulting constructs pBXMBQ_MB_H2 and pBXMBQ_MB_F7 encode a fusion protein that consists of an N-terminal DsbA signal peptide, the first thirteen N-terminal residues that form the β-strand A of a nanobody, a scaffold protein (HopQ adhesin domain)[22] and the rest of the nanobody residues (containing all three CDRs), fused C-terminally to a 3C protease cleavage site, His$_{10}$-tag, and Myc-tag.

### Vancomycin sensitivity assays in complemented *M. smegmatis* dKO cells

High-throughput cellular growth assays to assess the functionality of the transporter and LprG were conducted in principle as described in our previous publication[8]. In short, each tested strain was grown into stationary phase and diluted to OD$_{600}$ = 0.4 in 7H9 medium. 10 μl of these diluted cultures were transferred to wells containing 1 ml 7H9 medium complemented with vancomycin (concentrations as described in main text and figures) in 4 technical replicates and the cultures were incubated at 37 °C, 300 rpm in a 96-well plate. At indicated time-

points, 50 µl of culture were removed from the growth plate, transferred to a microtiter plate and $OD_{600}$ was measured in a PowerWave XS Microplate Reader (BioTek). The growth curves in the figures are representative of at least three biological replicates and error bars denote the standard deviation of four technical replicates. All biological replicates of Rv1410 and MHAS2168 mutants' growth curves are shown on Supplementary Fig. 12.

The vancomycin concentrations used in different assays were calibrated according to the experimental aims. 0.08 µg/ml vancomycin enables the growth of empty vector control and single protein (transporter or lipoprotein) complementation strains within the experimental time frame (~50 h). 0.1 µg/ml vancomycin concentration prevents the growth of the empty vector control and inactivated mutant, but allows separation of partially active mutants, such as previously characterized LprG V91W[12,13] and Rv1410 Truncation 1(= Long loop)[8]. 0.4 µg/ml vancomycin was used in cases where we aimed at distinguishing Rv1410 mutants that display such small growth disadvantages compared to the wild type strain that no growth differences can be observed at milder 0.1 µg/ml vancomycin condition.

### Western blotting
Western blotting for FLAG-tag detection was carried out exactly as described previously[8].

### Multiple sequence alignments
CLC Main Workbench was used to produce a multiple sequence alignment of Rv1410 homologues from 17 *Mycobacterium* species belonging to three phylogenetically separate clades: *M. abscessus, M. aurum, M. avium, M. chelonae, M. fortuitum, M. haemophilum, M. hassiacum, M. intracellulare, M. kansasii, M. leprae, M. marinum, M. phlei, M. saopaulense, M. smegmatis, M. tuberculosis, M. thermoresistibile, M. vaccae*. The alignment was visualized using JalView[58]. To prepare a multiple sequence alignment of different MFS transporters to test lack of conservation of Rv1410/MHAS2168 features, MUSCLE algorithm[59] was used for alignment of protein sequences acquired from PDB database and JalView was used for visualization of the multiple sequence alignment. The alignment was validated using a subset of the MFS transporters and their structure models by superimposition in Chimera.

### Structure predictions of Rv1410c and its homologues
ColabFold software[28] employing AlphaFold2[60] and MMseqs2[61] was used in batch mode with default settings to predict the structures of Rv1410 homologues from the following mycobacterial species: *M. abscessus, M. aurum, M. avium, M. fortuitum, M. hassiacum, M. marinum, M. phlei, M. smegmatis, M. tuberculosis, M. thermoresistibile*. ColabFold was similarly used to predict the structures of Rv1410 and MHAS2168 helix truncation mutants. Best models in outward-open conformation were chosen, superimposed on each other, and hydrophobicity analysis was performed in UCSF Chimera[62] and UCSF ChimeraX[63].

### Nanobody selections
For the selection of Rv1410 or MHAS2168 specific nanobodies, an alpaca was immunized with subcutaneous injections four times in two-week intervals, each time with 200 µg purified Rv1410 or MHAS2168 in 20 mM Tris-HCl pH 7.5, 150 mM NaCl, and 0.03% (w/v) *n*-dodecyl-*β*-D-maltopyranoside (β-DDM). Immunizations of alpacas were approved by the Cantonal Veterinary Office in Zurich, Switzerland (animal experiment licence nr. 172/2014). Blood was collected two weeks after the last injection for the preparation of the lymphocyte RNA, which was then used to generate cDNA by RT-PCR to amplify the VHH/nanobody repertoire. Phage libraries were generated and two rounds of phage display were performed against transporters solubilized in β-DDM. After the final phage display selection round, 1023-fold

enrichment was determined by qPCR using AcrB as background for MHAS2168 and 652-fold enrichment for Rv1410. The enriched nanobody libraries were subcloned into pSb_init[57] by FX cloning and 95 single clones were analyzed per transporter by ELISA. In case of MHAS2168, out of 88 positive ELISA hits, 22 were Sanger sequenced and 14 nanobodies were chosen for purification and further analysis. In case of Rv1410, out of 44 positive ELISA hits, 24 were Sanger sequenced and 7 nanobodies were discovered.

### Expression and purification of *M. tuberculosis* Rv1410 and its mutants
Rv1410 was produced in and purified from *E. coli* MC1061 following the same protocol as described previously[8]. Rv1410 mutants D70N, ΔAB, E147Q, D22N, A411D, and L289R were purified according to the same protocol by expressing them from the pBXC3GH_Rv1410_mutant vectors in *E. coli* MC1061.

### Expression and purification of *M. hassiacum* MHAS2168
*M. smegmatis* MC$^2$ 155 preculture harboring pACE_C3GH_MHAS2168 was inoculated from glycerol stocks into 7H9, HygB[50] and grown at 37 °C for 4 nights. The preculture was diluted 1:25 (v/v) into fresh expression medium (7H9, 0.2% Glc, HygB[15]) and grown at 37 °C until the culture $OD_{600}$ reached 0.8-1.0 before induction of protein expression with 0.016% acetamide overnight. Cells were harvested for 20 min at 6,000 rpm in a F9-6×1000 LEX centrifuge rotor (Thermo-Scientific) at 4 °C and resuspended in Resuspension Buffer (20 mM Tris/HCl pH 8.0, 200 mM NaCl) containing 3 mM $MgSO_4$ and traces of DNaseI. Cells were snap-frozen in liquid $N_2$ and stored at −80 °C until membrane preparation. Membranes were prepared by homogenizing the cell suspensions with a pestle in a Dounce homogenizer to remove larger cell clumps and subsequently disrupting the cells with a Microfluidizer (Microfluidics) at 30 kpsi on ice. Unbroken cells and cell debris were removed by centrifugation for 30 min at 8000 rpm in a Sorvall SLA-1500 rotor at 4 °C. Membranes were collected in a Beckman Coulter ultracentrifuge using a Beckman Ti45 rotor at 38,000 rpm for 1 h at 4 °C and resuspended in TBS (pH 7.5) containing 10% glycerol. Membranes were snap-frozen in liquid $N_2$ and stored at −80 °C until protein purification. Then, membranes were solubilized for 2 h using 1% β-DDM (w/v) and insolubilized material was removed by ultra-centrifugation. The supernatant was loaded on $Ni^{2+}$-NTA columns after addition of 15 mM imidazole, washed with Wash Buffer I (50 mM imidazole (pH 7.5), 200 mM NaCl, 10% glycerol, 0.03% (w/v) β-DDM) and eluted with Elution Buffer II (200 mM imidazole (pH 7.5), 200 mM NaCl, 10% glycerol, 0.03% (w/v) β-DDM). In order to remove the C-terminally attached GFP/His$_{10}$-tag, the buffer of the protein preparation was first exchanged to SEC Buffer (20 mM Tris/HCl pH 7.4, 150 mM NaCl, 0.03% (w/v) β-DDM) via a PD-10 desalting column. In a second step, 3C protease cleavage was performed overnight. Finally, cleaved MHAS2168 was again loaded on a $Ni^{2+}$-NTA column and washed out with SEC buffer to remove GFP/His$_{10}$-tag and the His-tagged 3C protease. Then, it was either run on size exclusion chromatography (SEC) on a Superose 6 Increase 10/300 GL column in SEC Buffer before Mb_H2 complex formation or added to nanobody H2 to form a complex.

### Expression and purification of Nb_H2
*E. coli* MC1061 preculture harboring nanobody H2 expression vector pBXNPHM3_MHAS_H2 was directly inoculated from glycerol stock into LB, Amp$^{100}$ and grown at 37 °C overnight. The preculture was diluted 1:40 (v/v) into fresh expression medium (TB, Amp$^{100}$) and grown for 2 h at 37 °C and an additional hour at 25 °C before induction of protein expression with 0.02% L-arabinose overnight. Cells were harvested for 20 min at 6000 rpm in a F9-6 × 1000 LEX centrifuge rotor (Thermo-Scientific) at 4 °C and resuspended in Resuspension Buffer containing 3 mM $MgSO_4$ and traces of DNaseI. Cells were disrupted with a

Microfluidizer (Microfluidics) at 30 kpsi on ice and unbroken cells and cell debris were removed as described above. Imidazole, to a final concentration of 20 mM, was added to the supernatant which was loaded on Ni$^{2+}$-NTA columns. The columns were washed with Wash Buffer II (1x TBS buffer (pH 7.5), 50 mM imidazole) and the bound nanobody was eluted with Elution Buffer II (1x TBS (pH 7.5), 300 mM imidazole). The eluted protein was dialyzed against SEC buffer overnight at 4 °C to remove excess imidazole and simultaneously cleaved with 3C protease to remove the N-terminally fused maltose binding protein and His-tag. Finally, nanobody H2 was again loaded on a Ni$^{2+}$-NTA column and washed out with 1x TBS (pH 7.5) containing 40 mM imidazole to remove MBP and His10-tag and the His-tagged 3C protease. The protein was then snap-frozen in liquid N$_2$ and stored at −80 °C until it was run on a SEC column (Sepax SRT-10C-300) in SEC buffer.

### Expression and purification of MB_H2 and MB_F7

*E. coli* MC1061 preculture harboring megabody expression vector pBXMBQ_MHAS_H2 or pBXMBQ_Rv_F7 was directly inoculated from glycerol stock into LB, Amp$^{100}$ and grown at 37 °C overnight. Megabody F7 expression and purification were conducted as described for nanobody H2 with the exception of the last SEC step when it was run on a Superdex 200 10/300 GL column. Megabody H2 expression, cell harvest, cell disruption and first purification steps were carried out as described for nanobody H2. After first elution in Elution Buffer II, the megabody sample was contaminated with DNA, therefore the sample was dialyzed against 1x TBS (pH 7.5) overnight, then 3 mM MgSO$_4$ was added and DNase treatment was performed for 2 h at 4 °C. Again, the sample was loaded on Ni$^{2+}$-NTA columns which were washed with Wash Buffer II and the bound megabody was eluted with Elution Buffer II. To remove the C-terminally attached His$_{10}$-tag, the buffer was exchanged to 1x TBS (pH 7.5) via a PD-10 desalting column and then 3C protease cleavage was performed overnight. Then, megabody H2 was again loaded on a Ni$^{2+}$-NTA column and washed with 1x TBS (pH 7.5) containing 30 mM imidazole to remove the His10-tag and the His-tagged 3C protease. Finally, it was separated from soluble aggregates on a Superose 6 Increase 10/300 GL column in SEC Buffer.

### Crystallization of MHAS2168 & Nb_H2 complex

Rv1410 did not yield any crystals after extensive vapour diffusion crystallization screening. Therefore, we purified its homologues from thermophilic mycobacterial species *M. thermoresistibile* and *M. hassiacum* and attempted to crystallize them. MHAS2168, the *M. hassiacum* Rv1410 homologue, produced crystals diffracting up to 7 Å. Subsequently, we generated 14 nanobodies against MHAS2168 and with three of these nanobodies, we obtained crystals diffracting up to 4 Å. Finally, systematic crystallization screening of the three MHAS2168-nanobody complexes in lipidic cubic phase (LCP) produced several crystals of the MHAS2168-Nb_H2 complex diffracting up to 2.7 Å, resulting in two native datasets (Supplementary Table 1). To obtain the MHAS2168-Nb_H2 complex in LCP, Nb_H2 was separated from soluble aggregates on a Sepax SRT-10C-300 column and mixed with MHAS2168 in a molar ratio of 1:1.5 (transporter:nanobody). After incubation on ice (10 min), the complex was run on a Superdex 200 10/300 GL column. The monodisperse peak of the complex was collected and concentrated with a 50 kDa cut-off concentrator (Vivaspin 2, Sartorius) and subsequently used for crystallization in LCP.

The concentrated transporter-nanobody complex (35 mg/ml) was mixed with molten 1-Oleoyl-rac-glycerol (monoolein, Sigma-Aldrich) at a protein:lipid ratio of 2:3 (v/v) using coupled syringe devices. 37 nl LCP boli were dispensed with a Crystal Gryphon LCP (Art Robbins Instruments) onto 96-well glass bases with a 120 µm spacer (SWISSCI), overlaid with 800 nl precipitant solution and sealed with a cover glass. The crystals were grown at 20 °C and reached full size by day 12. Two native datasets were obtained from 3 crystals (I, II, III) grown in

different reservoir solutions: I – 360 mM (NH$_4$)H$_2$PO$_4$, 0.1 M sodium citrate (pH 6.3), 31% (v/v) PEG400; II – 380 mM NaH$_2$PO$_4$, 0.1 M sodium citrate (pH 5.7), 28% (v/v) PEG400, 2.4% (v/v) 1,4-butanediole; III – 420 mM NaH$_2$PO$_4$, 0.1 M sodium citrate (pH 5.8), 28% (v/v) PEG400, 2.4% (v/v) 1,4-butanediole. X-ray diffraction data were collected at the X06SA beamline (Swiss Light Source, Paul Scherrer Institute, Switzerland) on an EIGER 16 M detector (Dectris) with an exposure setting of 0.05 s and 0.1˚ of oscillation over 120˚. Diffraction data was processed with the XDS program package[64] and datasets from crystals II and III were merged with xscale from the XDS program package[64]. Crystal I produced a complete dataset with no need for merging. The data-processing statistics are summarized in Supplementary Table 1. Both native datasets showed diffraction to 2.7 Å.

### Cryo-EM analysis of MHAS2168 & Mb_H2 complex

After separation of both the transporter and megabody alone on a Superose 6 Increase 10/300 GL column in SEC Buffer, monodisperse peaks of both proteins were gathered and mixed in molar ratio of 1:1.2 and incubated on ice for 10 minutes. After concentration with a 100 kDa cut-off concentrator (Amicon Ultra-0.5 Centrifugal Filter Unit) to remove empty β-DDM micelles, the MHAS2168-Mb_H2 complex was run again on a Superose 6 Increase 10/300 GL column in SEC Buffer, and a monodisperse peak was collected and concentrated for cryo-EM analysis.

4 µl of MHAS2168-Mb_H2 complex (9.4 mg/ml and 6 mg/ml) were applied to glow-discharged (45 s) holey carbon grids (Quantifoil R1.2/1.3 Au 200 mesh) and a Grid Plunger GP2 (Leica) was used to remove excess sample by blotting to filter paper (2.5-3.5 s, 90–95% humidity, 10 °C) and to plunge-freeze the grid rapidly in liquid ethane-propane. The grids were stored in liquid N$_2$ for data collection. The samples were imaged on a Titan Krios G3i (300 kV, 100 mm objective aperture), using a Gatan BioQuantum Energy Filter with a K3 direct electron detection camera (6k x 4k pixels) in super-resolution mode. 11,713 micrographs were recorded with a defocus range of −1 to −2.5 µm in an automated mode using EPU 2.7. The dataset was acquired at a nominal magnification of 130,000x, corresponding to a pixel size of 0.325 Å per pixel in super-resolution mode, with the total accumulated exposure of 66.54 e$^-$/Å$^2$ fractionated into 37 frames.

The data was processed in cryoSPARC v3.2[65]. First, the micrographs were subjected to patch motion correction and Fourier cropping, resulting in a pixel size of 0.65 Å per pixel. After subsequent patch CTF estimation, 11,542 good-quality micrographs were selected, based on the estimated resolution of CTF fits, relative ice thickness, and total full-frame motion. Template picking was used to pick particles from the micrographs; the templates were produced from an earlier lower-resolution map of MHAS2168-Mb_H2 complex from a smaller screening dataset obtained similarly. Particles were extracted with a box size of 600 pixels and Fourier-cropped to 300 pixels. After 6 rounds of 2D classification, 733,891 particles were subjected to 3-class ab initio reconstruction (default parameters) and 546,068 particles from the best two classes, showing megabody binding to the transporter, were used as input for heterogeneous refinement with the best-resolved and worse-resolved classes used as references and other parameters set to default. 402,229 particles from the best-resolved class (FSC resolution 7.19 Å) were directed into non-uniform refinement[66], which produced a map of the complex resolved to 4.2 Å, according to the 0.143 cut-off criterion[67]. To further improve the resolution, the particle set was extracted again from the micrographs with a 450-pixel box size without Fourier cropping and subjected to non-uniform refinement (default parameters). This resulted in a 4.0 Å cryo-EM map.

### Cryo-EM analysis of Rv1410 & Mb_F7 complex

Rv1410-Mb_F7 complex was prepared similarly to MHAS2168-Mb_H2 complex, but the sample concentration was 4.4 mg/ml and it was

applied to holey carbon grids with copper mesh (Quantifoil R1.2/1.3 Cu 200 mesh).

The data was acquired and processed similarly to MHAS2168-Mb_H2 complex, with the exception of recording 7984 micrographs with electron dose of 65.0 e⁻/Å² fractionated into 48 frames or 55.0 e⁻/Å² into 38 or 41 frames. After template picking of particles from 7631 good quality micrographs and 5 rounds of 2D classification, 427,971 particles were subjected to 3-class ab initio reconstruction (default parameters) and 300,246 particles from the best two classes, showing megabody binding to the transporter, were used as input for heterogeneous refinement with the best-resolved and worse-resolved classes used as references and other parameters set to default. 127,196 particles from the best-resolved class (FSC resolution 8.64 Å) were directed into non-uniform refinement[66] which produced a map of the Rv1410-Mb_F7 complex resolved to 7.51 Å, according to the 0.143 cut-off criterion[67].

### Structure determination

SWISS-MODEL[68] was used to generate homology models of MHAS2168 (based on PDB structure 6GS4) and Nb_H2 (based on PDB structure 5F7L) which were trimmed into polyalanine models with the CHAINSAW[69] program from CCP4 suite[70] and fitted into the 4.0 Å cryo-EM map in *Coot*[71]. ColabFold[28] software, combining MMseqs2 with AlphaFold2[60], was used to predict the structure of Rv1410 and several of its mycobacterial homologues. In *Coot*[71], registry was established and side-chains built manually into well-resolved helices 1–12 in the model fitted into the 4.0 Å cryo-EM map, while comparing the map to ColabFold structure predictions. The model was subjected to one cycle of real-space refinement in Phenix[72] before using it as a search model in molecular replacement with Phaser[73] to phase the native 2.7 Å single-crystal dataset. After iterative cycles of refinement and modelling with phenix.refine[74] and ISOLDE[75], a model was produced whose R-factors ($R_{work}$=0.2708 and $R_{free}$ = 0.3277) could not be improved with further refinement. The better-resolved chains A (MHAS2168) and B (Nb_H2) from this model were used as search model for molecular replacement with Phaser MR to phase the native 2.7 Å merged dataset. Again, refinement and modelling was performed with phenix.refine[74] and ISOLDE[75] to reach the final R-factors ($R_{work}$=0.2450 and $R_{free}$ = 0.2915) as indicated in Supplementary Table 1. The structure was validated with Molprobity[76].

### Molecular dynamics simulations

Coarse-grained MD simulations of the MHAS2168 transporter from *Mycobacteria hassiacum* in outward-open conformation were carried out with GROMACS version 2021.1[77] with the Martini 2.2 force field[78]. The sizes and compositions of all the simulated systems are listed in Supplementary Table 3. From the X-ray crystal structure, missing residues 55–57, 203–208 and 234–236 were added to the chain A using MODELLER[79], the best model was selected out of 100 structures. The model was briefly refined using ISOLDE[75] and further oriented along the z-axis using the PPM web server[80]. The obtained MHAS2168 atomistic model described above was converted to the coarse-grained resolution using the martinize.py script. To maintain the structural integrity of the protein, an elastic network with a cutoff distance of 7 Å for the MHAS2168^OUT, 9 Å for the MHAS2168^IN, 9 Å for the MHAS2168^IN-TAG and 9 Å for the MHAS2168^OUT-LprG (see below) was used with force constants of 1000 kJ × mol⁻¹ × nm⁻². For the MHAS2168^OUT-LprG complex, the refined MHAS2168 model described above was used to initiate the MD simulations of the complex after structural superposition over the model obtained by ColabFold. In other words, in the simulations of the MHAS2168^OUT-LprG complex, the refined X-ray coordinates of the MHAS2168 were used. In order to avoid a too rigid protein-protein interface, 112 intramolecular (7 within MHAS2168 and 105 within LprG) and 22 intermolecular (between MHAS2168 and LprG) harmonic potentials (i.e., the Martini elastic network) were removed. A

total of 8 elastic network bonds were kept between MHAS2168 and LprG. A set of ten independent repeat simulations of 30 μs each and without any intermolecular elastic network bonds were also run as a control (see Supplementary Table 3). These control simulations confirm the proposed TAG transfer mechanism (see Supplementary Fig. 6).

The Martini Maker tool available in the CHARMM-GUI web server[81] was used to insert the protein into a symmetric bilayer resembling the mycobacterial plasma membrane composition[3,4,82] (see Supplementary Table 3): 34% 1-palmitoyl-2-oleyl-phosphtidylethanolamine (POPE), 29% 1-palmitoyl-2-oleyl-phosphtidylglycerol (POPG), 15% cardiolipin (CDL), 21% 1-palmitoyl-2-oleyl-phosphtidylinositol (POPI) and 1% triacylglycerol (TAG)[83]. The Martini model for the cardiolipin, namely CDL2, has -2 charge (both phosphatidyl groups are charged). The Martini model for TAG represent a glycerol with three C18:1 oleoyl tails[78]. The MHAS2168^OUT-LprG complex was embedded in a bilayer of equal composition but using the insane tool[84], which was also used to build the Myco^mem system (see Supplementary Table 3). Prior to the MD simulations, to avoid a possible bias, one POPE and one POPG molecule were removed from the lower leaflet of the MHAS2168^IN system because they entered the main transmembrane cavity during the equilibration phase. For the MHAS2168^IN-TAG system, 1 TAG molecule from the bilayer was initially positioned within the central cavity of the transporter. The TAG molecule within the MHAS2168^OUT-LprG complex was modeled with 2 hydrophobic tails pointing upwards (towards LprG) and, as a control simulation, only 1 tail pointing upwards and 2 tails downwards (towards MHAS2168). All the systems were neutralized with a 150 mM concentration of NaCl and subsequently energy-minimized with the steepest descent until machine precision. The minimized systems were equilibrated in two consecutive steps of 50 ns and 200 ns of MD simulation, first with all protein beads and second with the protein backbone beads restrained by harmonic potentials (force constants of 1000 kJ × mol⁻¹ × nm⁻²). The time step for integrating the equations of motion in the coarse-grained simulations was 20 fs. The "new-RF" simulation parameters were used, as suggested by de Jong et al.[85]. Since the *M. hassiacum* is a thermophile[86], the equilibrations and production simulations were performed at 330 K with protein, each lipid species and solvent separately coupled to an external bath using the v-rescale thermostat[87] with coupling time constant $\tau_T$ = 1.0 ps. The pressure was maintained at 1 bar using the stochastic cell rescaling (c-rescale) barostat[88] with semi-isotropic conditions (coupling time constant $\tau_P$ = 12.0 ps and compressibility $3.0 \times 10^{-4}$ bar⁻¹). For each system, five independent production simulations (each of length 100 microseconds) were generated using different random seeds for the initial velocities. For the Myco^mem system, the production run was 20 microseconds long. For the control simulations of the MHAS2168^OUT-LprG complex, after the initial 100 microseconds of production run, the simulation was further extended by 100 microseconds, and 5 additional simulations of 100 microseconds were carried out with different random seeds for the initial velocities. Coordinates were saved to the disk every 400 ps. All analyses were carried out on the trajectories from the production runs. The GROMACS analysis tool gmx select in combination with an in-house script was used to calculate the protein-lipid contacts. A contact between a coarse-grained lipid headgroup bead and a protein bead was defined within a distance of 0.6 nm between the two. The tool gmx trajectory was used to extract the z-coordinate of the TAG center of mass and gmx density was used to calculate the density profile of the membrane components. The volume of the main transmembrane cavity was calculated using trj_cavity (https://sourceforge.net/projects/trjcavity/)[89]. For these calculations, we have used index files consisting of selections of protein residues encompassing the cavity: Residues 32–44, 60–75, 114–129, 153–164, 223–234, 288–303, 318–332, 388–403 and 420–438 were selected for the MHAS2168^OUT conformation. Residues 64–81, 122–137, 144–160, 288–300, 325–336,

395–408 and 414–431 were selected for the MHAS2168$^{IN}$ conformation. Before the calculations, the five trajectories where concatenated and the proteins were superimposed over the backbone atoms of the residues listed above. Molecular graphics were generated with VMD 1.9.4 (http://www.ks.uiuc.edu/Research/vmd/)[90]. Data were plotted using Grace (http://plasma-gate.weiz-mann.ac.il/Grace/).

## Modelling the MHAS2168 in inward-open conformation

To build an inward open conformation of MHAS2168, MODELLER[79] was used. Initially, a web frontend (http://www.ebi.ac.uk/Tools/msa/clustalo) to CLUSTAL Omega[91] was used to align the target protein sequence of MHAS2168 to the proton-dependent oligopeptide transporter PepTSo2 from *Shewanella oneidensis*[92] (sequence identity is 23%). The obtained alignment was manually curated to avoid fragmentation mostly in the region of the linker helices. The PepTSo2 X-ray structure PDB 4LEP (resolution 3.20 Å) was used as template in combination with an initial mock-model of MHAS2168 in inward-open conformation. To obtain this mock-model, the N- (residues 1–183) and C-terminal regions (residues 263–495) of MHAS2168 were independently superposed to the homologous parts of PepTSo2. During the modeling procedure the linker helices were restrained to assume a canonical α-helix conformation, however, their overall position differed from the template 4LEP. Therefore, to correctly position the target linker-helices, the best structural model out of 100 generated ones was further used for a second round of MODELLER together with both linker-helices modeled in the initial step and in turn superposed to the 4LEP linker-helices.

## Modelling the MHAS2168$^{OUT}$-LprG complex

The LprG protein sequence from *M. hassiacum* was downloaded from Uniprot (id: K5BJY3) and the platform ColabFold[28] was used to build the MHAS2168$^{OUT}$-LprG heterodimeric complex. At variance of all the models obtained, the first best model features LprG quite far away from the extended helices TM11 and TM12, and consequently, the second best model was selected. Further, the refined MHAS2168 X-ray structure, described above, was superposed over the TM11 and TM12 of the ColabFold MHAS2168 model and the first 30 N-terminal unstructured residues of LprG were removed.

## Molecular docking

The docking was performed using AutoDock-Vina v1.2.3[93,94] using a molecule of TAG with three C18:1 oleoyl tails as ligand. The TAG molecule was docked onto the MHAS2168 in both outward-facing and inward-facing conformations. The protein was kept rigid, while the TAG molecule could change its conformation and position in the selected search space, that was $7 \times 6.5 \times 8.8$ nm and $7.5 \times 6.5 \times 8.8$ nm, for the outward-facing and inward-facing conformation, respectively. These search spaces (400.4 nm$^3$ and 429 nm$^3$, respectively) were large enough to include not only the central cavity but also the periphery of the transporter, hence in principle allowing for a TAG to dock also from the outside. The *exhaustiveness* parameter was set to 50. A further molecular docking run was performed (using the same search space) in which, in addition to TAG, also MHAS2168 residues Glu157 and Arg426 were considered to be conformationally flexible.

## Reporting summary

Further information on research design is available in the Nature Portfolio Reporting Summary linked to this article.

## Data availability

The crystal structure of MHAS2168 in complex with Nb_H2 has been deposited in RCSB Protein Data Bank (PDB) with the accession code 8PNL. The cryo-EM map of MHAS2168 in complex with Mb_H2 has been deposited in Electron Microscopy Data Bank (EMDB) with the accession code EMD-17787. Source data are provided with this paper.

Plasmids and other data that support the findings of this study are available from the corresponding authors upon request. Source data are provided with this paper.

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

## Acknowledgements

Dr. Simona Sorrentino of the Center for Microscopy and Image Analysis, University of Zurich, is acknowledged for help with cryo-EM grid preparation and cryo-EM data collection. Beat Blattmann and Caroline Müller of the Protein Crystallization Center, University of Zurich, are acknowledged for their help with setting up the crystallization screens. We thank Saša Štefanić for conducting alpaca immunizations and the staff of the SLS beamlines X06SA and X06DA for their support during data collection. We thank Jennifer C. Earp for help with cryo-EM grid preparation and Dr. Alisa Garaeva for her advice on cryo-EM data processing. We are very grateful to Dr. Eric Geertsma and Dr. Benedikt Kuhn for sharing the pBXMBQ vector with us. All members of the Seeger lab are acknowledged for project discussion. Work in the lab of MAS was supported by a SNSF Professorship of the Swiss National Science Foundation (PP00P3_144823), the European Research Council (ERC) (consolidator grant n° 772190) and a grant of the Novartis Foundation for Medical-Biological Research (to MAS). SR was supported by a Candoc fellowship of the University of Zurich (grant nr. FK-17-035). Work in the lab of LVS was supported by the Deutsche Forschungsgemeinschaft (DFG) under Germany's Excellence Strategy – EXC 2033 – 390677874 – RESOLV and through grant SCHA1574/6-1.

## Author contributions

S.R. and M.A.S. conceived the project. M.H. and S.R. cloned all genes into the respective complementation and expression vectors for *E. coli* and mycobacteria. M.H. screened Rv1410 in vapour diffusion experiments and initiated MHAS2168 screening. M.H. and S.R. purified protein for alpaca immunization. SR conducted alpaca nanobody selections. S.R. purified all proteins and protein complexes thereafter. S.R. and J.S. crystallized MHAS2168-Nb_H2 complex in LCP. S.R., C.A.J.H., I.G., J.S., and M.A.S. collected crystal diffraction data. S.R. and C.A.J.H. processed the data and built and refined the model of MHAS2168-Nb_H2 complex. S.R. and I.G. prepared cryo-EM grids and performed cryo-EM data analysis. S.R. built and refined the MHAS2168 model into MHAS2168-Mb_H2 complex cryo-EM map. S.R. and M.A.S. analyzed the structures and S.R. produced the multiple sequence alignments and structure predictions. S.R. designed and carried out the mutagenesis of Rv1410 and MHAS2168 and performed Western blotting, as well as mutant protein purifications. S.R. conducted vancomycin sensitivity assays. D.D. and L.V.S. designed

and performed the molecular dynamics simulations. D.D. generated the MHAS2168 inward-facing homology model and the MHAS2168-LprG complex. S.N. instructed S.R. in LCP methodology and contributed to crystallization strategy development. S.R. and D.D. prepared the figures and the tables. S.R. and M.A.S. wrote the first draft of the manuscript. D.D. and L.V.S. wrote the MD simulation sections. All authors edited the manuscript.

## Competing interests

The authors declare no competing interests.
