## [Peer Review File · Nature Communications]

Structural basis for triacylglyceride extraction from mycobacterial inner membrane by MFS transporter Rv1410REVIEWER COMMENTS

Reviewer #1 (Remarks to the Author):

Overall, this is an impressive tour de force of structural methods: membrane protein purification, conventional detergent-assisted crystallography, LCP, single particle cryo EM and a good level of detail in the methods section. However, there is a strong reliance on MD simulations and these would need to be reviewed by an expert. many of the conclusions are drawn from the MD work and they would need to be verified. In addition, the MD work is oddly dependent on alpha fold predictions rather than actual structures which raises concerns about their reliability.

Despite that, overall this is an outstanding piece of work.

Line 73: Do they address how LprG interacts with Rv1410? No, not experimentally, and they seem to show that LprG does not specifically or directly interact with Rv1410. They perform molecular dynamics simulations using a rough alphafold multimer model of LprG with MHAS2168. I (or ideally an MD expert!) would need to carefully review these MD simulation videos but I am skeptical of MD even when you are starting with a real experimental protein structure as opposed to what they have here, where the starting complex model is a prediction, and a prediction generated in a vacuum (not the membrane) and it doesn't seem like this complex has been demonstrated experimentally. They aren't really able to verify the MD other than the point mutation analysis which for me only shows that you can break the transporter with obvious substitutions that affect the core structure.

Their experimental structures are outward-facing and the cryo-EM structure MAY contain a TAG molecule in the sidewall of the transporter (but no lipids found in the crystal structures?). Sidewall TAG binding the outward-facing transporter conflicts with their model and is confusing – at the outward facing stage, shouldn't the TAG already have transited through the sidewall and up into the periplasmic facing cavity prior to release to LprG? The MD in Figure 3 seems to show similar average TAG contacts with sidewalls of the inward facing and outward facing transporter, which might suggest that there is not a lot of specificity here (hydrophobic ligand interacting with hydrophobic surfaces). Also, for the single particle cryo EM, they should have a supplemental figure showing what the different classes look like and demonstrate that the class they chose to focus on is representative.

Line 85: Have the lipid components of non-model and nontuberculous mycobacteria been well-studied, ie does Hassiacum have TAGs to a similar degree? I would think thermophiles might have different lipid composition?

Line 87: Why did phasing fail with solved MFS family members?

Line 89: Nice single particle EM resolution for a small, dynamic channel!

Line 94: Where did the proposed lipid come from in this purified protein sample? Before describing the structure, the authors should describe the protein expression and purification process briefly. The membrane protein purification from smeg is pretty impressive and supports their hypothesis that the lipid species in the EM map might have come from smeg and therefore could be a TAG but the “nonprotein density” in these maps could literally be anything pulled down during the purification or during preparation and doesn’t seem to be present in the higher res crystal structure even though they both seem to use the same protein source. They could perform mass spec on their protein sample and see if they find any lipids?

Line 104: why did the phasing fail then?

Line 117: Extracellular suggests outside of the borders of the membrane, shouldn’t this say periplasmic?

Line 334: OK, the western blotting may show that you can produce the transporters containing various mutations but have they shown that these mutations do not alter folding/3D conformation.... I think not. Also there’s a ladder missing from the first figure showing the western blots.

Line 347-360: The hypothesis that TAG transport is driven by proton translocation is plausible and I like their model (fig 7). I don’t know enough about this family of transporters, is this an established mechanism for the family? If not, the mutagenesis data in this paper are not enough to convince me. They could simply be “breaking the protein” with the various mutations that disrupt vanc resistance.

Why do their growth curves look so strange? Smeg should be able to grow to a much higher density than OD600 = 0.3. Is the scale weird because of plate reader absorbance readout?

Figure 5a is cutoff?

Figure 6 -- lines depicting center of mass in the MD simulations are quite variable across the 5 simulations -- these are identical simulations and are supposed to look similar? Not sure you can really

take anything from this analysis anyway given the underlying fundamental problems I mentioned above (simulating structural predictions is like simulating a simulation).

Reviewer #2 (Remarks to the Author):

Please see attached.

What are the noteworthy results? Will the work be of significance to the field and related fields?

Remm et al., have determined the structure of Rv1410, an integral membrane protein from *Mycobacterium tuberculosis* that is conserved across mycobacteria. This result is noteworthy because this is the first structure of a multifacilitator superfamily (MFS) protein from mycobacteria and one of the very few of an MFS transporter that has a lipid substrate. Notably, the authors resolve unique extended helices that distinguish Rv1410 and its homologues from other MFS transporters and suggest a distinct mechanism or function. This study also reports mutational experiments in mycobacteria and molecular dynamics simulations that provide additional, inferential support for the pre-existing model that Rv1410 is a lipid and, specifically, a triacylglyceride transporter.

How does it compare to the established literature?

The study provides structural detail that corroborates their earlier work on Rv1410 mutations and their impact on complementation of phenotypes in a knockout strain. There is also some moderate evidence, such as non-protein electron density and additional mutational analysis based on the structure, that supports other studies implicating Rv1410 as lipid transporter. However, we note that like previous studies, the evidence is largely by inference since the primary functional assay uses an indirect bacterial phenotype (see further below in the next section).

The more obvious and more strongly supported impact of the study is the structure in comparison to other published structures of MFS transporters. The authors previously noted an insertion between helices 11 and 12 that distinguish Rv1410 from other MFS transporters and here they resolve that loop into helical extensions that are predicted to extend into the periplasm. The wealth of existing knowledge on MFS transporters would presumably inform interpretations of these distinct structural features, especially in terms of Rv1410 function and mechanism, but the manuscript currently lacks a thorough comparison of the new structure to this literature.

Does the work support the conclusions and claims, or is additional evidence needed? Are there any flaws in the data analysis, interpretation and conclusions? Do these prohibit publication or require revisions?

While the structures of Rv1410 and another mycobacterial homologue MHAS2168 are notable contributions to the field, the major criticism is that throughout the manuscript, the authors make conclusions that are not supported by the data (both previous published results and their own data). In general, we strongly encourage the authors to use careful and explicit language to describe what they conclude based directly on the data, versus what they propose as a model or speculation.

Therefore, we propose that the authors

- Edit the entire manuscript thoroughly to either remove or appropriately interpret data. In this case, we suggest that the manuscript focus on the Rv1410 structure with a deeper discussion of comparisons to other MFS transporters.
- Alternatively, perform experiments to support the more mechanistic claims, particularly with respect to the contributions of individual residues to Rv1410 function. Interpretation currently relies entirely on the vancomycin assay, which reports only on antibiotic sensitivity and not on lipid transport or Rv1410 function (e.g., lines 241-242: “Both MHAS2168 truncation mutants failed to transport lipids...” This conclusion cannot be made from the vancomycin assay, since the phenotype may not be due to TAG transport by Rv1410. See more details below.)

Fundamentally, the evidence for Rv1410 as a TAG transporter is by inference from lipidomic profiling of null strains. We note that even these data on TAG from Martinot et al. are limited to certain isoforms of TAGs. There is in fact currently no direct evidence that TAG is a substrate for Rv1410. We strongly recommend that the authors include this caveat and accordingly note that they make this assumption for this study, especially since none of the data, either here or in their previous work Hohl et al., address TAG transport explicitly.

The authors implement a vancomycin sensitivity assay because LprG and Rv1410 provide intrinsic antibiotic resistance to certain antibiotics such as vancomycin. Indeed, vancomycin and other antibiotic sensitivity phenotypes have been identified in other studies (e.g., Ramon-Garcia et al. 2009, 10.1128/AAC.00550-09; Li et al. 2022, 10.1038/s41564-022-01130-y; Xu et al., 2017, 10.1128/AAC.01334-17); the authors should cite these in addition to their own work in the introduction. The sensitivity assay is valid when mutations in Rv1410 have no effect; such results support the conclusion that changes to these residues are unimportant for function. However, the failure of certain mutants to restore vancomycin resistance in the knockout strain could be due to many factors other than those stated, including failure to fold/insert into the membrane. (The authors provide protein expression data in Figure S1, but (1) these data are not quantified and there are no loading controls and (2) these results cannot confirm that Rv1410 is properly inserted and folded in the membrane.) For such mutants, the data do not support conclusions about Rv1410 function or mechanism. We provide some detailed examples below, but these are not by any means exhaustive and we urge the authors to carefully reconsider all their conclusions based on the vancomycin assay or perform additional experiments using an independent assay, although we are not aware of any established assay that would report on lipid transport or use of the proton gradient by Rv1410.

Line 23: “transport of TAGs that seal the mycomembrane”

Line 31: “crucial for lifting TAGs away from the membrane plane”

Line 133: “Rv1410 and LprG transport TAGs which secure the impermeability”

Line 48: “LprG, which is embedded in the outer leaflet”

Line 63: “show to localize to the inner membrane”

These are examples of statements that are not directly supported by experimental evidence, but more accurately are models or speculation based on inference (first 3 examples) or assumption (last 2 examples). Please reword.

Line 62: “In mycobacteria, TAGs can be synthesized by several diacylglycerol acyltransferases”

Reference 17 cited here showed that purified antigen 85A can perform the DGAT reaction in vitro. However, there is no evidence that this activity is physiological and occurs in mycobacteria. Please reword.

Line 110: “A striking element”

Line 114: “a distinct feature”

Examples of missing context. Why is this a striking or distinct feature? Presumably in comparison to other MFS transporters, but this is not clear. Please provide additional information.

Section starting at line 131: The arguments for performing complementation experiments in *M. smegmatis* are confusing and do not follow a clear argument. In particular, the connection between the first and second sentences (lines 133-135) do not follow logically. Please clarify.

Line 143, Extended data Fig. 4: These WB data lack a loading control, explanation of potential non-specific cross-reacting bands, or quantification. This makes interpretation difficult, especially as both load and amount of signal from the band attributed to Rv1410 vary across samples. Please provide additional data and/or well-supported interpretation of these data to support the assumption that changes in protein production do not underlie observed phenotypes. Importantly, these data address production, but not native localization or folding of Rv1410, which is necessary for accurate interpretation of later mutant experiments. We suggest that the authors make note of these limitation or otherwise, as noted already above, and accordingly revise their later conclusions.

Line 155: “To test whether the ion lock residues contribute to energy coupling”

Line 163: “D22 in Rv1410 has been shown to be required for transport before”

Lines 175-176: “Our data suggest that in Rv1410, both D22 and E147 are implicated in proton translocation”

The assay used did not test whether the targeted residues are required for proton translocation, but whether they disturb Rv1410 in such a way as to affect vancomycin sensitivity. Please reword.

The authors' MD simulations are an intriguing application of their structure, especially with the inclusion of surrounding lipid membrane. However, more context and justification are needed to ensure the appropriateness of the conclusions. Especially crucial are justifications for the membrane composition and for the placement of triacylglyceride with the central cavity. On what evidence is the composition of the simulated membrane based? A review by Morita et al. (10.5772/52781) on the mycobacterial plasma membrane indicates that PG is a minor component, but in the simulated membrane in this study it is the second most common component.

As elsewhere in the manuscript, many conclusions are not clearly supported by the MD results. Again, the following examples are not exhaustive and all results should be reviewed again for interpretation:

Line 190: "MHAS2168 strongly interactions with cardiolipin and phosphatidylinositol" and extended data Fig. 7. What is the justification for a cutoff of 0.6 for "relevant" contacts? And why is the cutoff 0.4 for TAG? What is the evidence that interactions with CDL and PI are stronger than for others? PE and PG also (d and e) also show many contacts above the 0.6 threshold and for many helices such that the contacts would also be considered annular.

Line 195: "Linker helices A and B... form another hotspot for transporter-TAG interactions" What data in extended data Fig. 7 support this? No contacts above the threshold are present in TMA and TMB for the OUT conformation and contacts similar to TMA and TMB are also present in TM4 and TM8 in the IN conformation.

Line 197: "To test the importance of these features to TAG transport, either TMA and TMAB or TM9-TM10 b-hairpin were truncated" Extended Fig. 7 does not clearly support the truncation of the TM9-TM10 hairpin, as there do not appear to be any relevant contacts (as defined by the authors) in this region by MD. Please clarify.

Line 200: "deletion of linker helices resulted in inactivation" Given that TMA and TMB appear to be integral features in Rv1410 (at least as can be inferred from Fig. 1, although actual contacts made with the rest of Rv1410 are not discussed), it is not unreasonable to expect that their deletion would significantly perturb folding/localization of Rv1410. Please reword and consider alternate explanations.

Lines 203-204: "the central cavity.. is particularly apolar" Compare to what? Please provide context.

Lines 210-212: "This discrepancy might be explained by the fact that arginines retain their charge.... suggests that TAGs enter the central cavities" The connection between the charge state of arginines and the likelihood that TAGs enter the central cavity of Rv1410 is not clear. Please clarify, especially as this argument has important implications for the MD simulations starting at line 288. What is the justification for placing TAG into the central cavity given that this was not observed in the apo-Rv1410 MD simulations or the structures?

Line 236: "the "periplasmic loop" truncations... manifest in fact as helix truncations" A general question about this structural feature: Given that the nanobody targets this extended sequence unique to Rv1410, could the helical structure be an artifact of nanobody binding? Given that it was a predicted loop, is it possible that the helices are structured only in the presence of a binding partner? Please consider including (if not here, then earlier in section starting Line 102) a discussion of how the nanobody binding might impose structure and implications for structure/function of this extended sequence.

Line 242: "mutants failed to transport lipids" The vancomycin assay does not report on lipid transport. Please reword.

Given the above, many points made in the Discussion section are unsupported, or should be clearly stated as speculation (e.g., line 313-314: the unqualified statement: "Rv1410 acts in concert... to export TAGs..."). Other examples (not a complete list):

Line 337: "In fact, the discovery that linker helices of Rv1410 are crucial for its function is the first of its kind..." Based on which data?

Line 344: “TM12 truncations result in complete loss of transport activity, without disturbing folding and production.” As noted earlier, the WB data in the extended data do not address protein folding.

Lines 347-360: The arguments presented in this paragraph to explain the differences observed from the same mutants in different Rv1410 homologues does not follow a clear logic. Such differences could be due to diverse factors, and this paragraph is so speculative that we recommend it be removed.

Line 366: “first structure of an MFS transporter capable of lipid extraction” The evidence for lipid extraction is limited, given that it is based on rare events in an MD simulation that starts from assumptions about the substrate and where it can bind, and the experimental data do not report on transport. Please reword.

Is the methodology sound? Does the work meet the expected standards in your field? Is there enough detail provided in the methods for the work to be reproduced?

Methods as provided are acceptable.

The MIC assay is useful for determining residues that may be important for Rv1410 function, however it is not a direct read out of TAG transport function and is only a proxy of this. **It is important to note that the loss of viability phenotype may not be due to loss of TAG transport.** It is possible loss of Rv1410 function may lead to loss of viability in the presence of vancomycin because Rv1410 may have other functions (like in cell stress response). In addition, the authors conclude certain residues are essential for Rv1410 TAG transport activities through mutational analyses in this assay, but this phenotype could instead be due to non-functional mutants due to mutation. The authors show western blots in the extended figures to indicate that the Rv1410 mutations they make are viable, but this fails to take protein folding and location into account. Therefore, due to the many assumptions and limitations of the mutational analysis and the vancomycin MIC assay, the authors cannot make *direct* claims about TAG transport and instead need to carefully consider what conclusions can be drawn from this data.

Overall, our peer review is not complete, but we have just listed some major examples of where the authors have overinterpreted their data. We believe that the structural information contained in this paper is of great value to this field of research and think that this paper could be reframed and published once major revisions are completed.

Is the methodology sound? Does the work meet the expected standards in your field? Is there enough detail provided in the methods for the work to be reproduced?

The methodology provided is sound and thorough.

The authors provide a nice comparison between this MFS transporter with other known structures in the supplement. This MFS transporter interestingly has a periplasmic extension that appears to be necessary for its function through their MIC assay.

The authors also conduct extensive mutational analyses to try to understand how Rv1410 mediates lipid transport. However, they do not speculate about whether Rv1410 could have similar mechanisms of lipid transport as other MFS transporters (such as alternating access model). It appears that Rv1410 is quite different from the other known MFS structures. Perhaps the authors could elaborate on these differences to hypothesize how Rv1410 can transport TAG.

Reviewer #3 (Remarks to the Author):

The manuscript by Remm et al. shows structural, in-vivo functional, and in-silico characterizations of an important transport system for triacylglycerides in mycobacteria, which is required to seal the so-called mycomembrane. Remm et al. present the first structure of a transporter homolog (MHAS2168) of mycobacterium tuberculosis from a thermophilic mycobacterium in complex with a nanobody, and solved by X-ray crystallography at an impressive 2.7-angstrom resolution using lipidic cubic phase. Interestingly, the diffraction data was phased with a lower resolution structure obtained by cryo-EM using a megabody scaffold, providing an elegant example of the powerful synergy between those orthogonal structural techniques. The structural data are complemented with functional experiments in vivo, as well as molecular dynamics simulations, and a novel and exciting transport model is put forward.

The manuscript is elegantly written, concise and clear. It really is a page-turner. The figures have been carefully made and nicely represent the data and transport model. I am convinced that the work by Remm et al. will be of great interest for microbiologists, as well as for the membrane transport field.

I have several comments that aim to improve clarity, and some other aspects of the manuscript, and that the authors might like to consider during resubmission.

-Structure determination:

It is unclear why the crystallographic data could not be phased by molecular replacement. In the absence of cryoEM structures of the MHAS2168-Nb complex, I wonder if alpha-fold models could have helped with the molecular replacement. Or rather, the presence of the Nb and/or the unknown number of molecules in the asymmetric unit were the problem.

Regarding cryo-EM processing, I am a bit surprised by the medium- and low-resolutions of the maps obtained with the MHAS2168-Mb and RV1410-Mb complexes, respectively. In particular, the MHAS2168-Mb dataset is quite large and this might enable to uncover conformational heterogeneity in the sample, and possibly higher resolution maps. I am particularly concerned about the flexibility between the Nb and the HopQ scaffold that could have limited the resolutions obtained. In our experience, *ab initio* and heterogeneous classifications with default parameters, as stated in the methods, yield poor results with small membrane proteins. I would recommend to iteratively explore both types of classifications using progressively higher initial resolution values. Another important parameter to explore is the number of classes. Moreover, I would definitely run local refinement jobs masking out the HopQ part of the Mb to potentially get better alignments and resolution. Finally, cryosparc has different ways to uncover conformational heterogeneity like 3D variability and 3D classification jobs, and in more recent versions 3D flexible refinement. It might be worth exploring these options to find more homogeneous sets of particles and/or different conformations of the complex. Please, make a supplementary figure with details on the cryoEM pipeline processing.

A comparison between the cryoEM and X-ray structures of MHAS2168 is missing. Were the structures identical within the resolution limits?

What is the point to include in the manuscript a 7.5-angstrom cryoEM map of the RV1410-MbF7 complex?

-Report in bioRxiv:

There is a relevant report in the bioRxiv from the Penmatsa Lab (<https://www.biorxiv.org/content/10.1101/2022.07.09.499445v1.full.pdf>), where the authors present a cryoEM structure of an unrelated bacterial efflux pump (QacA) displaying a similar helical extracellular domain than the one observed in MHAS2168, and reaching out some 20 angstroms outside the membrane plane. Although the proposed role of the extracellular domain in QacA differs from the one in MHAS2168, I think a reference to this work and some discussion on the possible different roles of the unusual extracellular domain in MFS transporters are called upon.

-Transport mechanism:

Recent structural and functional analyses of MFS lipid transporters, particularly the bacterial LtaA, as well as chicken and human MSFD2A have shown the key role of hydrophobic cavities in the C-domain of the transporters to occlude and translocate the fatty-acid moieties of their lipid substrates. Although those works are cited in the manuscript, a discussion/comparison between MFS-mediated transport mechanisms of TGAs vs other-lipids is missing. For instance, the central cavities of LtaA and MSFD2 are, as opposed to that in MHAS2168, fairly polar because they accommodate the polar head of the lipid substrates. Do the authors observe or envision a role of C-domain cavities in TGAs transport? Are there structural features that could explain the MHAS2168/ RV1410 selectivity for TGAs?

-Protein production:

Please clarify is the FLAG-based protein production assay reports on functional protein targeted to the membrane, or it could also detect misfolded material in inclusion bodies, or else. This is important to interpret the functional results using mutants and deletions.

-MHAS2168-LprG complex:

What is the confidence of the heterodimer alpha-fold model? And what is its value since it is not validated functionally, i.e. the complementation assay using different mycobacteria species points towards no meaningful protein-protein interaction? In this regard, the structural depiction of conformational states in Fig. 7 seems a bit hyper-realistic and might be better to use a more cartoonish style.

Line 572: "Nb_H2 was separated on a Sepax SRT-10C-300 column" please clarify what separated means.

Line 617: “the templates were produced from an earlier lower-resolution map of MHAS2168-Mb_H2 complex.” Please, specify how the earlier low-resolution map was obtained.

Reviewer #4 (Remarks to the Author):

Remm et al. present a very interesting and timely study on the mechanism of triacylglyceride extraction and transport from the mycobacterial inner membrane. By combining structural biology, an impressive array of mutations and large-scale MD simulations, the authors propose a novel description of such extraction, and pinpoint critical residues involved in this process, providing mechanistic insights. The paper contains an impressive amount of data, and in general the conclusions are sound and supported by the data. I have however several comments about the presentation and interpretation of MD simulations, that, in my opinion, should be amended in the revised version.

The author performed a large amount of MD simulations to essentially study two aspects of TAG transport by Rv1410/MHAS2168: i) extracting of TAG from the membrane (transporter in the inward facing orientation), and ii) loading/channeling of a TAG molecule into LprG (transporter in the outward facing orientation). Overall, the simulations seem to be well planned, prepared and executed, however some details are missing. Only after reading the details of MD simulations in the SI, more questions arise. It does leave an impression that the authors picked the observations that supported their hypothesis in the main text, and do not discuss possible alternative scenarios, that seem to be plausible based on the MD report available in the SI. The results and description of the MD simulations are scattered across the main text, methods, extended data Figures and SI report. In my opinion it makes the manuscript hard to follow.

Comments relating to the i) set of simulations: extracting of TAG from the membrane (transporter in the inward facing orientation).

- The authors do not record any spontaneous insertions of TAGs to the transporter main cavity, even though it is a critical part of the proposed transport mechanism. In my opinion this should be investigated further. The authors say it is because of low TAG concentration they used (only 2 molecules) - why not try some simulations with a larger concentration? (even if it's larger than physiological one, for this purpose it would be justified). Moreover, for a separate simulation set (MHAS2168IN-TAG), where the authors placed a TAG molecule in the central cavity (if I understood correctly), the authors are listing the residence times, that vary 8-74 μ s. That implies that the TAG molecule escaped the central cavity? This is unfortunately not reported at all. If TAG escaped the cavity to the membrane, that would provide a pathway for TAG to enter/exit the cavity? If TAG escaped to the water phase, that could imply that it binds to the cavity from there? Please do explain. All this should be

also discussed in the context of using a homology model for the inward facing state, that might lead to inaccuracies and, for example, a possibility of alternative scenarios allowing TAG to bind to the central cavity, such as larger conformational changes of involved helices.

^[1]_{SEP} TAG molecules (or similar ones) had been previously found before to aggregate in the membrane core (Khandelia et al., PLOS ONE 2010). If the same occurs in the mycobacterium membrane, then several scenarios of TAGs entrance to the transporter are possible. Do single molecules enter or an aggregate is necessary? Do the aggregates leave the membrane and then TAGs bind the transporter from the aqueous phase (see previous comment?) I don't expect the authors to try all these hypotheses, but some alternatives should be at least mentioned in the revised manuscript.

- The authors see the entering of other lipids to the central cavity, but then rather ignore it or even delete the molecules that did that. What about the possibility that it might be functionally relevant? The cavity seems to be large enough to accommodate more than one lipid (is that true?). Perhaps constant occupation of the cavity by a lipid(s) increases the hydrophobicity of it, allowing for easier TAG binding?

Comments relating to the ii) set of simulations: loading/channeling of a TAG molecule into LprG (transporter in the outward facing orientation)

- The second critical part of the mechanism proposed in the current manuscript, is the formation the E147-R417 ion lock at the bottom of the central cavity, that "lifts TAG toward the periplasmic leaflet". This is a quite plausible scenario. I see here however a missed chance for MD simulations: to really prove this hypothesis, simulations with mutations in residues forming this lock would be particularly useful.^[1]_{SEP}

The authors say in the introduction that "An LprG cross-linking study in live *Mycobacterium smegmatis* cells did not demonstrate a reproducible interaction between LprG and Rv141020 and attempts to show complex formation in vitro with purified proteins were fruitless." and then further argue that "transient low affinity interactions might occur in the cellular context." The authors then decided to prepare a model of the LprG and Rv141020 complex using ColabFold to study TAG loading into LprG. It is not clear from the Methods section whether the two proteins (LprG and MHAS2168) were made to interact with each through harmonic potentials or not; this sentence is not clear: "For the MHAS2168OUT-LprG complex, 134 (out of a total of 3698) intra- and inter-molecular elastic network bonds were removed to avoid a too rigid protein-protein interface in the MHAS2168-LprG complex." If there is no additional forces to keep LprG bound to MHAS2168 in MD simulations, how the authors reconcile the fact that the complex is seemingly stable for at least 100us, with "transient low affinity interactions"?

Minor comments:

Extended Data Figure 7 should be ideally replaced by the 2D density plots for each lipid around the protein, especially if, as the authors argue, an annular lipid belt of specific lipids is created around the transporter.

In the SI the authors talk about “snorkelling events” for TAG molecules and “entry/exit events” for other lipids, referring to the SI Figure 2. I cannot easily see how one distinguishes between these two types of events from such plots. In the caption, the same definition is given for both types of events. Surprisingly, there are no events at all in the conformation, at which TAGs are expected to bind (inward facing, see main comments).

there are multiple models of cardiolipin available in Martini 2.2, please specify which was used

it is not clear what was the exact chemical structure of the TAG molecule used in simulations, and which Martini model was selected to describe it. Please specify.

Throughout the text and in the caption of Figure 1, the authors use the expression “non-proteinaceous density”, whereas the Figure 1 says “Lipid” in red. That seems too unambiguous.

The wording in the last two paragraphs on page 7 is somewhat confusing. First the authors argue that “Functional assays (Fig. 4e,f) suggest that TAGs enter the cavity through both TM5-TM8 and TM2-TM11 lateral openings in the inward-facing state” to later say “Even though the narrow TM5-TM8 lateral cleft in the OF state is connecting the central cavity and membrane, it does not seem to play a role in TAG transport.” I think that the authors want to say that the TM5-TM8 cleft plays a role in TAG transport only in the inward-facing state, but it is somewhat confusing in the current formulation.

Reviewer #1 (Remarks to the Author):

Overall, this is an impressive tour de force of structural methods: membrane protein purification, conventional detergent-assisted crystallography, LCP, single particle cryo EM and a good level of detail in the methods section. However, there is a strong reliance on MD simulations and these would need to be reviewed by an expert. many of the conclusions are drawn from the MD work and they would need to be verified.

Authors' reply:

We thank reviewer#1 for the overall positive assessment of our work. We agree with the reviewer that important aspects of our proposed mechanism are supported by (but not solely dependent on) MD simulations. Importantly, essential elements of the extraction mechanism are footed on structural and functional observations made as part of this study (i.e. the biophysical nature of the highly hydrophobic outward-facing lipid binding pocket containing the ion lock and the periplasmic extensions being unique to Rv1410) as well as previous works on LprG clearly showing how TAG and other triacylated lipids are bound to its hydrophobic cavity.

In addition, the MD work is oddly dependent on alpha fold predictions rather than actual structures which raises concerns about their reliability.

Authors' reply:

We wish to note that at least some parts of the MD work were performed based on the MHAS2168 outward-facing structure. Other parts of the MD indeed involved a homology model of inward-facing MHAS2168. And finally the simulations that involved the MHAS2168-LprG complex were indeed footed on a AlphaFold prediction. To initiate the MD simulations of the MHAS2168-LprG complex, the actual structure of MHAS2168 was used next to an AlphaFold prediction of *M. hassiacum* LprG. We wish to note that recent works set out to test the ability of AlphaFold to predict protein-protein interactions, highlighting potentials and limitations of the tool (Bryant et al., 2022. Nat. Commun. 13, 1-11; Burke et al. 2023. Nat. Struct. Mol. Biol. 30, 216-225; Evans et al., 2022. bioRxiv 2021.10.04.463034; Yu et al., 2023, Bioinformatics 39), and showing that AlphaFold can be used with an overall very promising performance towards the investigation of protein complexes (Bryant et al., 2022. Nat. Commun. 13, 1-14).

In the context of our work, we would like to note that the top five MHAS2168-LprG complex predictions looked very similar (Figure S9), suggesting that the prediction of the interaction appears fairly robust. Further, we wish to note that the handover of the hydrophobic TAG molecule from MHAS2168 to LprG does not depend on the fine details of the LprG interface with MHAS2168, but are rather robust. In this context, please also consult our reply to reviewer#4 below where we describe additional control MD simulations that support our initial findings.

Despite that, overall this is an outstanding piece of work.

Authors' reply:

We thank the reviewer for this very positive judgment of our work.

Line 73: Do they address how LprG interacts with Rv1410? No, not experimentally, and they seem to show that LprG does not specifically or directly interact with Rv1410.

Authors' reply:

Indeed, as we pointed out in our manuscript (and in our previous manuscript on Rv1410-LprG), we were not able to determine a complex between the two proteins. However, in these experiments, LprG was produced in its soluble form devoid of its lipid attached to the N-terminus and Rv1410 was purified in detergent. Hence, the reviewer is certainly correct in claiming that protein-protein interactions between the operon partners are weak and likely only transient, and our own experiments in mixing and mingling the operon partners from different mycobacterial species support this notion (Fig 5d-h). However, it should be taken into consideration that in the *in vivo* context, Rv1410 sits in its native lipid bilayer and LprG is tethered to the outer leaflet of the inner membrane via a N-terminal lipid-anchor, thereby restricting its conformational freedom. Hence, the local concentrations of the operon partners might be rather high in the native context.

They perform molecular dynamics simulations using a rough alphafold multimer model of LprG with MHAS2168. I (or ideally an MD expert!) would need to carefully review these MD simulation videos but I am skeptical of MD even when you are starting with a real experimental protein structure as opposed to what they have here, where the starting complex model is a prediction, and a prediction generated in a vacuum (not the membrane) and it doesn't seem like this complex has been demonstrated experimentally.

Authors' reply:

We agree that the reviewer has a point here in saying that MD simulations have to be taken with a grain of salt, in particular if they involve a protein complex that was "only" predicted, but not validated experimentally. However, first of all it should be noted that our AlphaFold predictions of the MHAS2168-LprG complex were rather robust (Figure S9). In addition, there are several published examples of successful prediction of protein complex structures with AlphaFold (see references listed in the answer to the question above). Secondly, we did not use the AlphaFold coordinates of MHAS2168 but the actual X-ray structure for our MD simulations. Thirdly, in our case the handover of hydrophobic TAG from MHAS2168 to LprG (and backward) was rather robust and did not depend on the fine details of the relative orientation of LprG versus MHAS2168. Finally, to avoid possible confusion, we would like to point out that AlphaFold does not generate a vacuum structure. While the reviewer is correct in that neither membrane nor solvent molecules are explicitly included in AlphaFold, the effects of the

environment are taken into account via the nature of the training data from which the engine “learned” how proteins fold.

They aren’t really able to verify the MD other than the point mutation analysis which for me only shows that you can break the transporter with obvious substitutions that affect the core structure.

Authors’ reply:

We respectfully disagree with the reviewer that we have tested our hypothesis only by “breaking the transporter with obvious substitutions that affect the core structure”. Quite to the contrary, we carefully chose (see Supplementary Note 1: Rationale for mutant design) a large set of substitutions along the proposed TAG entry and exit pathway including truncations of the helical extensions and functionally analyzed them in the context of Rv1410-LprG and the MHAS2168-LprG stemming from *M. tuberculosis* and *M. hassiacum*, respectively.

Finally, it should be stressed that other colleagues in field have unambiguously established in a number of published works that both Rv1410 and LprG are needed to export triacylated lipids such as TAGs and how LprG recognizes these lipids (Martinot et al. 2016, PLoS Pathogens. 12, e1005351; Drage et al. 2010, Nat. Struct. Mol. Biol. 17, 1088-95).

The major contribution of this work presented here was to deliver the structural characterization of the transporter side of the extraction mechanism, thereby finding conspicuous structural elements such as the very hydrophobic lipid binding cavity, the characteristic helical extensions and the linker helices, all being shown here to be critical for transport activity.

Their experimental structures are outward-facing and the cryo-EM structure MAY contain a TAG molecule in the sidewall of the transporter (but no lipids found in the crystal structures?).

Authors’ reply:

We thank the reviewer for pointing this out. Indeed, in our crystal structure we do not see additional lipid density at the position in question. We suspect that the lipid may have been “lost” in the process of mixing monoolein as part of LCP crystal preparation. Hence, it is well possible that the preparation of the protein for cryo-EM analyses is less harsh and thus a higher proportion of annular lipids remain bound to the transporter. Further, we are cautious in claiming what lipid is contained in this density; we now state in the revised manuscript version that it may correspond to a lipid (see lines 90-91).

Sidewall TAG binding the outward-facing transporter conflicts with their model and is confusing – at the outward facing stage, shouldn’t the TAG already have transited through the sidewall and up into the periplasmic facing cavity prior to release to LprG?

Authors’ reply:

We agree with the reviewer that the location of this lipid species at the sidewall of outward-facing MHAS2168 is puzzling. Further, it should be noted that LprG was not present in our cryo-EM analyses.

This is why we do not make any additional claims about the potential functional role of this lipid binding site, other than mentioning its existence (which we believe should be done as we clearly observe it). One may speculate that this lipid species may represent a TAG being positioned close where the inward-facing cavity is expected to open (to achieve higher local concentration, to act like a reservoir). But we consider this interpretation way too speculative and not substantiated with experimental data to be included in the manuscript text.

The MD in Figure 3 seems to show similar average TAG contacts with sidewalls of the inward facing and outward facing transporter, which might suggest that there is not a lot of specificity here (hydrophobic ligand interacting with hydrophobic surfaces).

Authors' reply:

We agree with the reviewer that the MD simulations did not indicate a highly specific TAG binding site. However, concerning the similarity of the TAG contacts in the inward-facing and outward-facing conformations, we would like to refer to Table S4 and stress that the MHAS2168 residues can be grouped into three distinct categories. About one third of the TAG-contacting amino acids are indeed common to both conformations, as inferred by the reviewer from Figure 3. However, the remaining two thirds of the residues subdivide into two distinct groups that do not overlap but are conformer-specific. Therefore, while there is no single specific binding site (due to the hydrophobic nature of the interactions, as correctly pointed out by the reviewer), our results suggest that there is nevertheless a conformer-specific "fingerprint" in terms of the TAG interactions.

Furthermore, we invite reviewer#1 to see our replies to reviewer#4 below, where we describe additional molecular docking analyses that we performed to compare MHAS2168-TAG contacts.

Also, for the single particle cryo EM, they should have a supplemental figure showing what the different classes look like and demonstrate that the class they chose to focus on is representative.

Authors' reply:

We thank reviewer#1 for pointing this out; indeed, a figure showing the cryo-EM processing pipeline was lacking and now is included in the revised manuscript. Please see Figure S1e.

Line 85: Have the lipid components of non-model and nontuberculous mycobacteria been well-studied, ie does *Hassiacum* have TAGs to a similar degree? I would think thermophiles might have different lipid composition?

Authors' reply:

TAG is such a common metabolite that we would consider it as hard to find an organism which does not have any TAG. Of course, this does not tell about the TAG level in different organisms. We are not aware of any studies regarding the lipid composition of *M. hassiacum* membranes. However, a search for homologues with tblastn reveals at least 5 triacylglycerol synthase genes in the genome of *Mycolicibacterium hassiacum* DSM 44199 isolate. The tblastn search was separately conducted against

Rv3130c (Tgs1), Rv3734c (Tgs2), and Rv3234c (Tgs3) proteins from *Mycobacterium tuberculosis* H37Rv and each search yielded the same genes from *M. hassiacum*: MHAS_00538, MHAS_01477, MHAS_03988, MHAS_04337, and MHAS_01120. We do not exclude that search for homologues of the rest of the triacylglycerol synthase genes from *M. tuberculosis* could yield additional triacylglycerol synthase homologues in *M. hassiacum*. However, in our opinion, these results at least seem to indicate that triacylglyceride synthesis is possible in *M. hassiacum*.

Line 87: Why did phasing fail with solved MFS family members?

Authors' reply:

That's a question we asked ourselves a lot. The project was stuck for more than a year after having obtained the native 2.7 Å crystal dataset. We tried very hard to get a molecular replacement solution by employing all possible approaches (using 50 different MFS structures as search models, cutting MFS transporters into two halves as search models, trying all possible conformations, producing search models lacking the most diverging transmembrane helices based on structural comparisons of existing MFS models and finally using a large set of different nanobodies for phasing, etc). It should be mentioned that this phase of the project was pre-AlphaFold era, and we have not tried to phase with an AlphaFold model of MHAS2168. We think that the periplasmic TM11-TM12 helix extensions of Rv1410 and MHAS2168 might have played a role in the failing of molecular replacement because none of the used MFS structures displayed such a feature.

Line 89: Nice single particle EM resolution for a small, dynamic channel!

Authors' reply:

Thanks a lot; we indeed were happy about the outcome; the distinctive features of the helical extensions and the nanobody/megabody approach certainly helped. And newer versions of CryoSPARC were critical as well.

Line 94: Where did the proposed lipid come from in this purified protein sample? Before describing the structure, the authors should describe the protein expression and purification process briefly. The membrane protein purification from smeg is pretty impressive and supports their hypothesis that the lipid species in the EM map might have come from smeg and therefore could be a TAG but the "nonprotein density" in these maps could literally be anything pulled down during the purification or during preparation and doesn't seem to be present in the higher res crystal structure even though they both seem to use the same protein source. They could perform mass spec on their protein sample and see if they find any lipids?

Authors' reply:

We thank reviewer#1 for this remark and we have now included a small addition to the manuscript pointing out the fact that the protein was produced in *M. smegmatis*.

“We finally succeeded to crystallize its close homologue MHAS2168 (sequence identity of 62%) from the thermophilic mycobacterial species *M. hassiacum*, purified from *M. smegmatis*, in complex with an alpaca-derived nanobody Nb_H2 using crystallization in lipidic cubic phase (LCP).” (See lines 79-83)

The expression and purification of MHAS2168 are described in great detail in the methods section and because of the space restraints we would refrain from more detailed descriptions in the main text.

As the reviewer rightly points out, the extra-density could be any lipid present in the inner membrane of *M. smegmatis*. The fact that we see density corresponding to three acyl chains limits the options. As suggested by the reviewer, we carried out a lipid analysis of MHAS2168 purified from *M. smegmatis*. As a control, we used the ABC transporter IrtAB, as well expressed in and purified from *M. smegmatis*.

The lipid analyses were carried out in our metabolomics facility at the Functional Genomics Center Zurich, and for each sample around 15 lipid species were detected. We were able to detect mostly charged lipid species, such as phosphatidylethanolamines and cardiolipins.

To our disappointment, we did not observe TAGs to be co-purified with MHAS2168 (nor any other triacylated mycobacterial lipid). Their hydrophobic nature, large mass diversity and poor ionization poses significant challenges to detect TAGs (see also the mycobacterial lipidomics papers of Branch Moody lab, in particular PMID: 22195556, where technical details on how to optimally extract, separate and ionize mycobacterial lipids for mass spectrometry detection, but also PMID: 25815898, PMID: 24516143).

Line 104: why did the phasing fail then?

Authors' reply:

Good question. Molecular replacement appears very sensitive to small structural deviations. Even the individual lobes exhibit considerable conformational flexibility and thus are prone to fail as molecular replacement search models.

Line 117: Extracellular suggests outside of the borders of the membrane, shouldn't this say periplasmic?

Authors' reply:

We agree with the reviewer and have changed it accordingly:

“Finally, MHAS2168 features a small periplasmic β -hairpin between TM9 and TM10 (Fig. 1b,c), which extends along the membrane plane.” (Lines 115-117)

Line 334: OK, the western blotting may show that you can produce the transporters containing various mutations but have they shown that these mutations do not alter folding/3D conformation.... I think not. Also there's a ladder missing from the first figure showing the western blots.

Authors' reply:

We agree that Western blotting only shows whether a mutant is produced or not and that it does not inform about its structural integrity. To address the reviewer's justified concern, we have successfully purified selected mutants of Rv1410 (D70N, Δ AB, E147Q, D22N, A411D, L289R) that showed negative phenotypes in our vancomycin sensitivity assays (mutants with wild type-like growth are obviously folded and functional). We have chosen the *M. tuberculosis* homologue because it is generally less stable than MHAS2168, because the latter stems from the thermophilic *M. hassiacum*. Please see Figure S4c,d for the size exclusion chromatography and SDS-PAGE analysis results.

"Several Rv1410 mutants displaying negative phenotypes in the vancomycin sensitivity assays were in addition purified to confirm their proper folding and membrane insertion (Fig. S4c,d)." (See lines 147-149)

We have also added the missing ladder to Figure S4a.

Line 347-360: The hypothesis that TAG transport is driven by proton translocation is plausible and I like their model (fig 7). I don't know enough about this family of transporters, is this an established mechanism for the family? If not, the mutagenesis data in this paper are not enough to convince me. They could simply be "breaking the protein" with the various mutations that disrupt vanc resistance.

Authors' reply:

There is a large number of MFS transporters that operate as proton-substrate antiporters and this coupling mechanism is also the most plausible one for Rv1410 and MHAS2168, as they are classified to the "2.A.1.3: The Drug:H⁺ Antiporter-2 (14 Spanner) (DHA2) Family" according to sequence homology to other antiporters (<https://www.tcdb.org/>). As for "breaking the transporter", we have now purified a set of mutants to show that the mutants do not affect folding (see previous comment).

Why do their growth curves look so strange? Smeg should be able to grow to a much higher density than OD₆₀₀ = 0.3. Is the scale weird because of plate reader absorbance readout?

Authors' reply:

As we point out in the methods section, OD₆₀₀ is measured in a plate reader with an approximate path length of 0.1 cm. Hence, the cells grew perfectly fine.

Figure 5a is cutoff?

Authors' reply:

We wish to confirm that Fig. 5a is depicted as intended. Our intention was to emphasize the unique periplasmic extension composed of helices TM11 and TM12 of the transporter.

Figure 6 -- lines depicting center of mass in the MD simulations are quite variable across the 5 simulations -- these are identical simulations and are supposed to look similar? Not sure you can really take anything from this analysis anyway given the underlying fundamental problems I mentioned above (simulating structural predictions is like simulating a simulation).

Authors' reply:

The lines in Figure 6 are variable because MD trajectories are stochastic in nature. We carried out several repeats that were initialized from different random particle velocities, which were drawn from a Maxwell-Boltzmann velocity distribution that corresponds to the desired temperature. Although such repeats are (unfortunately) not always done in the literature, they are very important because they improve the statistical reliability of the MD results (in addition to the sheer length of the simulations, which in our case is quite extended, 0.1 ms for each trajectory). Concerning the comment about "simulating a simulation", please see our answers to the question above.

Reviewer #2

What are the noteworthy results? Will the work be of significance to the field and related fields?

Remm et al., have determined the structure of Rv1410, an integral membrane protein from *Mycobacterium tuberculosis* that is conserved across mycobacteria. This result is noteworthy because this is the first structure of a multifacilitator superfamily (MFS) protein from mycobacteria and one of the very few of an MFS transporter that has a lipid substrate. Notably, the authors resolve unique extended helices that distinguish Rv1410 and its homologues from other MFS transporters and suggest a distinct mechanism or function. This study also reports mutational experiments in mycobacteria and molecular dynamics simulations that provide additional, inferential support for the pre-existing model that Rv1410 is a lipid and, specifically, a triacylglyceride transporter.

Authors' reply:

We thank reviewer #2 for recognizing the importance of our work.

How does it compare to the established literature?

The study provides structural detail that corroborates their earlier work on Rv1410 mutations and their impact on complementation of phenotypes in a knockout strain. There is also some moderate evidence, such as non-protein electron density and additional mutational analysis based on the structure, the supports other studies implicating Rv1410 as lipid transporter. However, we note that like previous

studies, the evidence is largely by inference since the primary functional assay uses an indirect bacterial phenotype (see further below in the next section).

Authors' reply:

We consider the evidence in favor of Rv1410 and its homologues to be TAG transporters as solid, because it foots on convincing lipidomics datasets clearly establishing a link between the absence/presence of Rv1410-LprG and accumulation/secretion of TAGs (see Martinot et al, PloS pathogens, 2016). And it is evident from structural works that LprG binds tri-acylated lipids, including TAGs. In our previous work, we could also convincingly show that vancomycin sensitivity in the transporter knockout strain arises from a permeability change of the mycomembrane linked to TAG export (and maybe also other lipids); direct efflux of vancomycin can be excluded.

Nevertheless, we agree with the reviewer statement that our evidence is rather circumstantial and indirect. We would have wished to see TAG (or another tri-acylated lipid) to be bound to places in MHAS2168 directly relevant to the transport pathway, and we have tried hard to achieve this by crystallizing the protein in the presence of TAG. Unfortunately, we did not achieve this goal. In this context, we wish to note that it is generally difficult to find specific lipid binding sites in lipid transporter structures (see for example PMID: 35710838).

The more obvious and more strongly supported impact of the study is the structure in comparison to other published structures of MFS transporters. The authors previously noted an insertion between helices 11 and 12 that distinguish Rv1410 from other MFS transporters and here they resolve that loop into helical extensions that are predicted to extend into the periplasm. The wealth of existing knowledge on MFS transporters would presumably inform interpretations of these distinct structural features, especially in terms of Rv1410 function and mechanism, but the manuscript currently lacks a thorough comparison of the new structure to this literature.

Authors' reply:

When writing this manuscript, no structures of Major Facilitator Superfamily transporters sported similar periplasmic helix extensions; now, a manuscript on QacA structure has been brought to our attention and we have therefore edited our manuscript accordingly. Please see lines 113-115, 340-344:

“A similar extracellular element between TM11 and TM12 has been recently described in the structure of multi-drug efflux pump QacA from *Staphylococcus aureus*.”

“A similar loss of substrate transport has been observed in the multi-drug efflux pump QacA when its extracellular element located between TM11 and TM12 was deleted²⁹, suggesting a more general functional relevance for structural features inserted between TM11 and TM12 in DHA2 subclass of MFS transporters.”

Does the work support the conclusions and claims, or is additional evidence needed? Are there any flaws in the data analysis, interpretation and conclusions? Do these prohibit publication or require revisions?

While the structures of Rv1410 and another mycobacterial homologue MHAS2168 are notable contributions to the field, the major criticism is that throughout the manuscript, the authors make conclusions that are not supported by the data (both previous published results and their own data). In general, we strongly encourage the authors to use careful and explicit language to describe what they conclude based directly on the data, versus what they propose as a model or speculation. Therefore, we propose that the authors

- Edit the entire manuscript thoroughly to either remove or appropriately interpret data. In this case, we suggest that the manuscript focus on the Rv1410 structure with a deeper discussion of comparisons to other MFS transporters.

Authors' reply:

In this point, we politely disagree with Reviewer #2. First of all, the biophysical properties of the MHAS2168 structure (and with this we mean for example the size and surface chemistry of the lipid binding pocket, the unique ion-lock and the biophysical properties of the helical extensions) are hard facts, and we consider it as appropriate to interpret these structural elements. Second, we have generated more than 40 mutants of Rv1410 and MHAS2168 and studied them in our vancomycin sensitivity assay. Also these functional data, which go hand in hand with the structural analyses, we consider as a solid piece of original data, which we interpret in the context of TAG transport.

Hence, we feel that our structures bring more value to the field, when interpreted in the context of the physiological function of the transporter, instead of comparing it excessively with other MFS transporter structures. Having said this, we wish to note that we highlighted the structural features that differ from other MFS transporter structures (lines 102-107; 118-123; 154-155) and shown comparisons to other MFS lipid and drug transporter structures to illustrate the reason why Rv1410 and MHAS1268 could indeed be TAG transporters (Figure S3). We have now also added discussion about the differences and similarities in lipid transporter mechanisms between Rv1410 and other MFS lipid transporters to the Discussion (see lines 368-383):

“Recently, structural models of two other MFS lipid transporters have been made available: LtaA³⁰ and MFSD2A³²⁻³⁴. LtaA is a H⁺/lipoteichoic acid antiporter from *S. aureus*³⁰ and MFSD2A is a Na⁺-dependent lysophosphatidylcholine-docosahexaenoic acid importer in the blood-brain barrier in humans.^{34,53} Unlike Rv1410, LtaA and MFSD2A display amphipathic central cavities, reflecting the greater polarity of their substrate lipids' headgroups (Fig. S3). Based on MD simulations and cross-linking studies³¹ in which the lateral crevices are locked by disulfide bridges, it has been proposed that lipoteichoic acid enters LtaA cavity from the inner leaflet of cytoplasmic membrane through the TM5-TM8^{IN} lateral cleft and exits through either TM5-TM8^{OUT} or TM2-TM11^{OUT} lateral crevice to the outer leaflet, having been flipped. In MFSD2A, non-proteinaceous densities possibly corresponding to lipids have been observed in both TM5-TM8^{OUT} and TM2-TM11^{OUT} lateral clefts.³³ It is very likely that lipids enter the cavity of MFSD2A, as lipid

density has been observed within the cavity in the IF state³² and a suitable hydrophobic pocket for accommodating long aliphatic chains has been discovered in the occluded state of MFSD2A³⁴. Whether analogous side pocket(s) form within Rv1410 for the concealment of TAG lipid tails remains unknown until structures of its other conformations are obtained.”

- Alternatively, perform experiments to support the more mechanistic claims, particularly with respect to the contributions of individual residues to Rv1410 function. Interpretation currently relies entirely on the vancomycin assay, which reports only on antibiotic sensitivity and not on lipid transport or Rv1410 function (e.g., lines 241-242: “Both MHAS2168 truncation mutants failed to transport lipids...” This conclusion cannot be made from the vancomycin assay, since the phenotype may not be due to TAG transport by Rv1410. See more details below.)

Fundamentally, the evidence for Rv1410 as a TAG transporter is by inference from lipidomic profiling of null strains. We note that even these data on TAG from Martinot et al. are limited to certain isoforms of TAGs. There is in fact currently no direct evidence that TAG is a substrate for Rv1410. We strongly recommend that the authors include this caveat and accordingly note that they make this assumption for this study, especially since none of the data, either here or in their previous work Hohl et al., address TAG transport explicitly.

Authors’ reply:

We respectfully disagree with the following statement of reviewer#2: “Fundamentally, the evidence for Rv1410 as a TAG transporter is by inference from lipidomic profiling of null strains. We note that even these data on TAG from Martinot et al. are limited to certain isoforms of TAGs. There is in fact currently no direct evidence that TAG is a substrate for Rv1410.”

In our understanding, the lipidomic data shown by Martinot et al. (2016) in fact is direct evidence for the notion that TAGs are transported by Rv1410: in the knockout, TAG accumulates within the cell and if the operon is overexpressed, more TAGs are present in the culture filtrate. And in our view, the argument that not all TAG species have been observed does not hold, because not all TAG species are equally abundant and some of them might simply fly below the detection limit of the mass spectrometry analysis. What is more, Martinot et al. demonstrated the ability of LprG to bind TAG by co-crystallization and showed that LprG is able to transfer TAG molecules between lipid vesicles.

Together with the data from our previous paper (Hohl et al. 2019) which demonstrated that the drug sensitivity of LprG-Rv1410 null strain in *M. smegmatis* stems from increased permeability of the cell envelope and biophysical changes in it, and the studies which have detected TAGs in the outer membrane (Bansal-Mutalik & Nikaido, 2014; Chiaradia et al. 2017), we have no reason to presume that the TAG distribution differences in the lipidomics study by Martinot et al. (2016) are caused by an indirect effect, and not TAG transport. Furthermore, our current study on Rv1410 (and MHAS2168) shows how the transporter is structurally suited for TAG transport.

To make it clear that we are interpreting our data under the assumption that Rv1410 is a TAG transporter, we have included the following sentence in the Results section besides explaining the rationale in the Introduction:

“Rv1410 and LprG have been shown previously to extract TAGs from the inner membrane so that TAGs could reach the mycomembrane where they secure its impermeability to certain drugs.^{8,12”} (See lines 132-134)

The authors implement a vancomycin sensitivity assay because LprG and Rv1410 provide intrinsic antibiotic resistance to certain antibiotics such as vancomycin. Indeed, vancomycin and other antibiotic sensitivity phenotypes have been identified in other studies (e.g., Ramon-Garcia et al. 2009, 10.1128/AAC.00550-09; Li et al. 2022, 10.1038/s41564-022-01130-y; Xu et al., 2017, 10.1128/AAC.01334-17); the authors should cite these in addition to their own work in the introduction.

Authors’ reply:

We have included these citations in the Results section “Experimental system to assess functionality of Rv1410 and MHAS2168 mutants in *M. smegmatis*”, please see lines 136-138:

“The deletion of *lprG/rv1410c* homologous operon *MSM3070/69* in *M. smegmatis* results in increased vancomycin influx to the periplasm as in other mycobacteria.^{8,10,35,36”}

The sensitivity assay is valid when mutations in Rv1410 have no effect; such results support the conclusion that changes to these residues are unimportant for function. However, the failure of certain mutants to restore vancomycin resistance in the knockout strain could be due to many factors other than those stated, including failure to fold/insert into the membrane. (The authors provide protein expression data in Figure S1, but (1) these data are not quantified and there are no loading controls and (2) these results cannot confirm that Rv1410 is properly inserted and folded in the membrane.) For such mutants, the data do not support conclusions about Rv1410 function or mechanism

Authors’ reply:

We agree with the reviewer that non-functional mutants might simply misfold. We have now addressed this issue by purifying Rv1410 mutants that exhibit negative phenotypes in our growth assays to show that they are successfully folded and inserted to the membranes (see answer to reviewer #1, Figure S4c,d and lines 147-149).

We provide some detailed examples below, but these are not by any means exhaustive and we urge the authors to carefully reconsider all their conclusions based on the vancomycin assay or perform additional experiments using an independent assay, although we are not aware of any established assay that would report on lipid transport or use of the proton gradient by Rv1410.

Authors' reply:

As we have outlined above, we consider our indirect way of measuring transporter activity by the vancomycin sensitivity assay as a good surrogate to quantify the amount of TAG being present in the outer mycobacterial membrane. As we have shown in a previous paper (Hohl, Remm et al., Mol Micro, 2019), deletion of the *M. smegmatis* *rv1410-lprG* operon results in increased permeability of the outer mycobacterial membrane, as we have shown based on BCECF-AM fluorescence measurements conducted in the presence and absence of the proton gradient uncoupler CCCP.

As the reviewer rightly notes, there is no high-throughput assay available to directly measure TAG transport for the mutants (next to the lipidomics assay published in Martinot et al. 2016, which is, however, clearly not high-throughput and it would not be justified to study all our mutants in such a complex lipidomics assay).

Line 23: "transport of TAGs that seal the mycomembrane"

Authors' reply:

We consider this statement as correct based on previous works.

Line 31: "crucial for lifting TAGs away from the membrane plane"

Authors' reply:

We have reworded the sentence in question to:

"The functional role of the periplasmic helix extensions is to channel the extracted TAG into the lipid binding pocket of LprG." (Lines 30-32)

Line 133: "Rv1410 and LprG transport TAGs which secure the impermeability"

Authors' reply:

We consider this statement as correct based on previous works.

Line 48: "LprG, which is embedded in the outer leaflet"

Author's reply:

To our knowledge, there is currently no evidence to show that LprG is transported from the inner membrane to the mycomembrane. We have included this statement to emphasize the fact that LprG is a lipoprotein and as such is localized to the outer leaflet of the inner membrane. Many papers exploring LprG's function have done so with its non-acylated form, and our goal was to stress the fact that LprG has restraints to its localization in the periplasm due to its nature of a lipoprotein.

Line 63: “show to localize to the inner membrane”

In Daniel et al. (2004) paper the Tgs activity was identified in the membrane fraction when the enzymes were expressed in *E. coli*. In Mawuenyega et al. (2005), all Tgs proteins that were identified by mass spectrometry were localized to the membrane fraction (Rv3130c, Rv3740c, Rv3371, Rv1425, Rv2484c), or cell wall fraction (Rv3130c, Rv0895). We have rephrased the sentence in the revised manuscript (lines 62-63):

“In mycobacteria, TAGs can be synthesized by several diacylglycerol acyltransferases¹⁶ of which many have been predicted to localize to the inner membrane¹⁷.”

These are examples of statements that are not directly supported by experimental evidence, but more accurately are models or speculation based on inference (first 3 examples) or assumption (last 2 examples). Please reword.

Line 62: “In mycobacteria, TAGs can be synthesized by several diacylglycerol acyltransferases”

Reference 17 cited here showed that purified antigen 85A can perform the DGAT reaction in vitro. However, there is no evidence that this activity is physiological and occurs in mycobacteria. Please reword.

Authors' reply:

We would like to bring to the attention of Reviewer #2 that the paper by Daniel et al. (2004) identified several TAG synthetases. To satisfy Reviewer #2 we have removed the citation of Ag85A.

Line 110: “A striking element”

Line 114: “a distinct feature”

Examples of missing context. Why is this a striking or distinct feature? Presumably in comparison to other MFS transporters, but this is not clear. Please provide additional information.

Authors' reply:

We agree that “striking” is not appropriate wording here: we call it “A conspicuous element” in the revised version. Further, we reworded the statement in line 114 to “Extended TM11 and TM12 seem to be a common feature of Rv1410 and its homologues,...” (lines 111-112).

Section starting at line 131: The arguments for performing complementation experiments in *M. smegmatis* are confusing and do not follow a clear argument. In particular, the connection between the first and second sentences (lines 133-135) do not follow logically. Please clarify.

Authors' reply:

We hope that the changes to this section have made it easier to understand the rationale of complementation assays.

“Rv1410 and LprG have been shown previously to extract TAGs from the inner membrane so that TAGs could reach the mycomembrane where they secure its impermeability to certain drugs.^{8,12} Thus, the functionality of Rv1410 and LprG can be probed indirectly in *M. smegmatis* cells, by exploiting the increase in cell envelope permeability if TAGs are not transported to the mycomembrane. The deletion of *lprG/rv1410c* homologous operon *MSM3070/69* in *M. smegmatis* results in increased vancomycin influx to the periplasm as in other mycobacteria.^{8,10,35,36}”

Line 143, Extended data Fig. 4: These WB data lack a loading control, explanation of potential non-specific cross-reacting bands, or quantification. This makes interpretation difficult, especially as both load and amount of signal from the band attributed to Rv1410 vary across samples. Please provide additional data and/or well-supported interpretation of these data to support the assumption that changes in protein production do not underlie observed phenotypes. Importantly, these data address production, but not native localization or folding of Rv1410, which is necessary for accurate interpretation of later mutant experiments. We suggest that the authors make note of these limitation or otherwise, as noted already above, and accordingly revise their later conclusions.

Authors' reply:

For each mutant, we used the same amount of cells (based on OD measurements) to prepare the Western blot sample. Further, we have included 2 control samples on each Western blot - the wild-type transporter (accordingly to the mutants, Rv1410 or MHAS2168) and an empty vector control in which no transporter is produced to be able to compare the mutant expression levels. Indeed, we have not included a loading control, because we did not attempt to quantify each signal, but were interested in a qualitative readout whether an individual mutant is produced similar to wild-type or not. Our Western blots show valuable insights: the mutant proteins are produced and they don't seem to be degraded or aggregated compared to the wild-type control. What is more, the empty vector control enables to interpret which bands are non-specific to the α -FLAG antibody. We have changed the figure caption of Figure S4 to include these caveats.

Importantly, we have now addressed the criticism about proper folding and membrane insertion by purifying several mutants that show negative phenotypes in our vancomycin sensitivity assays (see Figure S4c,d). With regards to the native localization of Rv1410: it is general knowledge that bacterial membrane transporters of the MFS superfamily (and any other alpha-helical bacterial membrane protein) are folded into the cytoplasmic membrane, while misfolded proteins aggregate in the cytoplasm. Hence, so long we can show a mutant of our transporter is produced and can be purified via size exclusion chromatography (as we did as part of our revision), we can be certain that the mutant is well-folded and localized to the cytoplasmic membrane.

Line 155: “To test whether the ion lock residues contribute to energy coupling”

Line 163: “D22 in Rv1410 has been shown to be required for transport before”

Lines 175-176: “Our data suggest that in Rv1410, both D22 and E147 are implicated in proton translocation”

The assay used did not test whether the targeted residues are required for proton translocation, but whether they disturb Rv1410 in such a way as to affect vancomycin sensitivity. Please reword.

Author's reply:

We agree with the reviewer that we do not provide a formal proof for Rv1410 to translocate protons. This is why we have made the following changes to the Results section "Two candidate loci for proton translocation" to make clear that Rv1410's dependence on proton motive force is a hypothesis:

"Being classified to the DHA2 subclass of MFS, Rv1410 is likely a TAG:H⁺ antiporter." (Line 151)

We also have a whole paragraph dedicated to proton translocation as a potential mechanism in the Discussion.

As to the wordings, we hope that the following changes satisfy the Reviewer #2:

"To test whether the ion lock residues are important for transporter function, E157_{MH} from MHAS2168 and the corresponding E147_{Mtb} from Rv1410 were mutated to glutamine." (Lines 159-160)

"In summary, our data suggest that in Rv1410, both D22 and E147 likely play a role in coupling the energy stored in the proton gradient to the efflux of TAGs." (Lines 177-179)

We did not change the sentence "D22 in Rv1410 has been shown to be required for transport before" as the functional data from Farrow & Rubin's 2008 paper is quite clear and now our purification of the Rv1410 D22N mutant confirms that the protein is in general folded and inserted to the membranes. It seems plausible to us that the D22A and D22E mutations would not affect the folding of the transporter in a much more extreme manner.

Reviewer #3 (Remarks to the Author):

The manuscript by Remm et al. shows structural, in-vivo functional, and in-silico characterizations of an important transport system for triacylglycerides in mycobacteria, which is required to seal the so-called mycomembrane. Remm et al. present the first structure of a transporter homolog (MHAS2168) of mycobacterium tuberculosis from a thermophilic mycobacterium in complex with a nanobody, and solved by X-ray crystallography at an impressive 2.7-angstrom resolution using lipidic cubic phase. Interestingly, the diffraction data was phased with a lower resolution structure obtained by cryo-EM using a megabody scaffold, providing an elegant example of the powerful synergy between those orthogonal structural techniques. The structural data are complemented with functional experiments in vivo, as well as molecular dynamics simulations, and a novel and exciting transport model is put forward.

The manuscript is elegantly written, concise and clear. It really is a page-turner. The figures have been

careful made and nicely represent the data and transport model. I am convinced that the work by Remm et al. will be of great interest for microbiologists, as well as for the membrane transport field.

Authors' reply:

We would like to thank the reviewer for their encouraging comments on our work.

I have several comments that aim to improve clarity, and some other aspects of the manuscript, and that the authors might like to consider during resubmission.

-Structure determination:

It is unclear why the crystallographic data could not be phased by molecular replacement. In the absence of cryoEM structures of the MHAS2168-Nb complex, I wonder if alpha-fold models could have helped with the molecular replacement. Or rather, the presence of the Nb and/or the unknown number of molecules in the asymmetric unit were the problem.

Authors' reply:

As pointed out above in the answer to Reviewer#1, we tried very hard to solve the LCP crystal structure by molecular replacement without any success. This was done during the pre-AlphaFold era and we have not tested whether phasing would have worked with AlphaFold. Of note, also the nanobody did not help for molecular replacement and the fact that the density for the nanobody was very poor in one of the asymmetric units certainly did not help to find a molecular replacement solution.

Regarding cryo-EM processing, I am a bit surprised by the medium- and low-resolutions of the maps obtained with the MHAS2168-Mb and RV1410-Mb complexes, respectively. In particular, the MHAS2168-Mb dataset is quite large and this might enable to uncover conformational heterogeneity in the sample, and possibly higher resolution maps. I am particularly concerned about the flexibility between the Nb and the HopQ scaffold that could have limited the resolutions obtained. In our experience, ab initio and heterogeneous classifications with default parameters, as stated in the methods, yield poor results with small membrane proteins. I would recommend to iteratively explore both types of classifications using progressively higher initial resolution values. Another important parameter to explore is the number of classes. Moreover, I would definitely run local refinement jobs masking out the HopQ part of the Mb to potentially get better alignments and resolution. Finally, cryosparc has different ways to uncover conformational heterogeneity like 3D variability and 3D classification jobs, and in more recent versions 3D flexible refinement. It might be worth exploring these options to find more homogenous sets of particles and/or different conformations of the complex. Please, make a supplementary figure with details on the cryoEM pipeline processing.

Authors' reply:

We would like to stress that we did try different parameters when processing the data in cryoSPARC but included to the manuscript only information on the workflow that yielded the best-resolution map. To be more precise, we tried cryo-EM analysis of only MHAS2168-Nb_H2 complex, but the best resolution obtained (4.4 Å) was clearly beneath the MHAS2168-Mb_H2 complex resolution. We did try masking out the HopQ domain of the megabody and explored Local Refinement jobs with different fulcrum points, but this approach was never as successful as using non-uniform refinement (new). We also tested Ab Initio Reconstruction and Heterogeneous Refinement jobs with different number of classes. We played around with Non-Uniform Refinement (new) parameters and tested different box sizes for re-extraction of particles. All in all, it might have been possible to get a map with better resolution if we had invested even more time in the data processing and perhaps even tried other programs like Relion and it might be possible to get even better results now with updated cryoSPARC versions and new job types. However, we already knew that we had X-Ray data diffracting to 2.7 Å, so when we reached the 4.0 Å cryo-EM map we decided to try model building and molecular replacement. Upon the success of phasing with molecular replacement we did not feel it was necessary to dedicate any more resources to the improvement of the cryo-EM map, but rather directed our efforts toward the functional studies of mutants.

We have produced a figure depicting MHAS2168-Mb_H2 cryo-EM data analysis pipeline, please see Figure S1e.

A comparison between the cryoEM and X-ray structures of MHAS2168 is missing. Were the structures identical within the resolution limits?

Authors' reply:

Initially, we did not polish the cryo-EM structure due to its inferior resolution and the lack of side-chain information in linker helices TMA and TMB for which only very crude densities are present. Now, we have fitted the crystal structure into the cryo-EM map, polished it in ISOLDE and conducted final refinement with Phenix (main validation parameters from Phenix: MolProbity score 1.04, Clash Score 2.56, Ramachandran Outliers 0.0, Ramachandran Allowed 0.69, Ramachandran Favored 99.31). We then removed the linker helices TMA and TMB (Gln 209 – Arg 277), N-terminus (Thr16-Ser17) and C-terminus (Phe 507-510) from both the X-ray structure and the cryo-EM structure. When ChimeraX Matchmaker tool was used to align the two structures, R.M.S.D.=0.692 Å across 402 pruned C α pairs (with the cutoff value of 2.0 Å) and R.M.S.D.=0.814 Å across all 411 C α pairs was observed, confirming that both structures feature the transporter in the same conformation. The low R.M.S.D. values also exemplify why molecular replacement worked – the structures are extremely similar (see Figure 1 below). Nevertheless, we have not included the cryo-EM structure in the manuscript nor plan to publish it due to the worse resolution: the side-chain placement is less precise in the cryo-EM structure and linker helices TMA and TMB as well as many loops and the β -hairpin showed very crude densities. Therefore, we don't believe that any biologically meaningful differences could be gleaned from the cryo-EM map by an average PDB database user.

Figure 1. Comparison of the X-ray and Cryo-EM structures of MHAS2168. MHAS2168 crystal structure is depicted in grey and MHAS2168 cryo-EM structure is depicted in pink. Only C α atoms are visualized in both structures.

What is the point to include in the manuscript a 7.5-angstrom cryoEM map of the RV1410-MbF7 complex?

Authors' reply:

First of all, we consider helpful to report that Rv1410 looks very similar to MHAS2168 (i.e. also possesses periplasmic TM11-TM12 extensions, as predicted by AlphaFold), as Rv1410 is the best-studied homologue of this MFS transporter subclass. Further, we found it interesting to show where the nanobody binds in that case. For these reasons, we would like to keep this extra information as part of our manuscript, knowing that the resolution is not the greatest.

-Report in bioRxiv:

There is a relevant report in the bioRxiv from the Penmatsa Lab (<https://www.biorxiv.org/content/10.1101/2022.07.09.499445v1.full.pdf>), where the authors present a cryoEM structure of an unrelated bacterial efflux pump (QacA) displaying a similar helical extracellular domain than the one observed in MHAS2168, and reaching out some 20 angstroms outside the membrane plane. Although the proposed role of the extracellular domain in QacA differs from the one

in MHAS2168, I think a reference to this work and some discussion on the possible different roles of the unusual extracellular domain in MFS transporters are called upon.

Authors' reply:

We thank the reviewer for spotting this structural study. There are indeed some interesting commonalities between the structure of QacA and MHAS2168, which are worth being highlighted and discussed. We have added the following text to the manuscript:

“A similar extracellular element between TM11 and TM12 has been recently described in the structure of multi-drug efflux pump QacA from *Staphylococcus aureus*.” (Lines 113-115 in Results)

“A similar loss of substrate transport has been observed in the multi-drug efflux pump QacA when its extracellular element located between TM11 and TM12 was deleted³⁰, suggesting a more general functional relevance for structural features inserted between TM11 and TM12 in DHA2 subclass of MFS transporters.” (Lines 340-344 in Discussion)

-Transport mechanism:

Recent structural and functional analyses of MFS lipid transporters, particularly the bacterial LtaA, as well as chicken and human MFSD2A have shown the key role of hydrophobic cavities in the C-domain of the transporters to occlude and translocate the fatty-acid moieties of their lipid substrates. Although those works are cited in the manuscript, a discussion/comparison between MFS-mediated transport mechanisms of TGAs vs other-lipids is missing. For instance, the central cavities of LtaA and MFSD2 are, as opposed to that in MHAS2168, fairly polar because they accommodate the polar head of the lipid substrates. Do the authors observe or envision a role of C-domain cavities in TGAs transport? Are there structural features that could explain the MHAS2168/ RV1410 selectivity for TGAs?

Authors' reply:

To a large extent, this has been addressed in Figure S3 where we provide a comparison of the hydrophobicity surfaces of the lipid MFS transporters and some drug transporters. We have now also added a paragraph to the discussion to highlight the known similarities and differences in lipid transport mechanisms of MFS transporters. Please see lines 368-383:

“Recently, structural models of two other MFS lipid transporters have been made available: LtaA³⁰ and MFSD2A³²⁻³⁴. LtaA is a H⁺/lipoteichoic acid antiporter from *S. aureus*³⁰ and MFSD2A is a Na⁺-dependent lysophosphatidylcholine-docosahexaenoic acid importer in the blood-brain barrier in humans.^{34,53} Unlike Rv1410, LtaA and MFSD2A display amphipathic central cavities, reflecting the greater polarity of their substrate lipids' headgroups (Fig. S3). Based on MD simulations and cross-linking studies³¹ in which the lateral crevices are locked by disulfide bridges, it has been proposed that lipoteichoic acid enters LtaA cavity from the inner leaflet of cytoplasmic membrane through the TM5-TM8^{IN} lateral cleft and exits through either TM5-TM8^{OUT} or TM2-TM11^{OUT} lateral crevice to the outer leaflet, having been flipped. In MFSD2A, non-proteinaceous densities possibly corresponding to lipids have been observed in both TM5-

TM8^{OUT} and TM2-TM11^{OUT} lateral clefts.³³ It is very likely that lipids enter the cavity of MFSD2A, as lipid density has been observed within the cavity in the IF state³² and a suitable hydrophobic pocket for accommodating long aliphatic chains has been discovered in the occluded state of MFSD2A³⁴. Whether analogous side pocket(s) form within Rv1410 for the concealment of TAG lipid tails remains unknown until structures of its other conformations are obtained.”

-Protein production:

Please clarify is the FLAG-based protein production assay reports on functional protein targeted to the membrane, or it could also detect misfolded material in inclusion bodies, or else. This is important to interpret the functional results using mutants and deletions.

Authors' reply:

We thank the reviewer for this question that was also raised by other reviewers. As part of the revision, we have now purified selected Rv1410 mutants that display negative phenotypes in our functional assays. Please see our answer to Reviewer #1 above and Figure S4c,d.

-MHAS2168-LprG complex:

What is the confidence of the heterodimer alpha-fold model? And what is its value since it is not validated functionally, i.e. the complementation assay using different mycobacteria species points towards no meaningful protein-protein interaction? In this regard, the structural depiction of conformational states in Fig. 7 seems a bit hyper-realistic and might be better to use a more cartoonish style.

Authors' reply:

We thank the reviewer for this suggestion and concur with the statement, that Fig. 7 is a bit hyper-realistic. We have changed Fig. 7 accordingly to a more schematic representation.

Concerning the confidence of the heterodimer of the AlphaFold model and its value, we wish to note that the top 5 models generated by AlphaFold were fairly similar in terms of relative orientation of MHAS2168 versus *M. hassiacum* LprG. In addition, we would like to highlight that our mechanistic conclusions that arise from the MD simulations, in terms of the handover of the TAG molecule from MHAS2168 to LprG, do not depend on the details of the LprG interface with MHAS2168 but are rather robust, see also reply to reviewer#1 and #4 below and additional control MD simulations we performed that support our initial findings (see also below). We assign this robustness to the hydrophobic nature of the tunnel between LprG and MHAS2168: TAG can slide along this tunnel, whose atomic-level details do not seem to be that important (as long as it is *a*) in place, and *b*) “greasy”). From a mechanistic viewpoint, this insensitiveness might be beneficial as it confers some degree of “robustness” to the process.

Line 572: “Nb_H2 was separated on a Sepax SRT-10C-300 column” please clarify what separated means.

Authors' reply:

We changed this to “Nb_H2 was run on a Sepax SRT-10C-300 column”. We also made according changes elsewhere in the methods section.

Line 617: “the templates were produced from an earlier lower-resolution map of MHAS2168-Mb_H2 complex.” Please, specify how the earlier low-resolution map was obtained.

Authors' reply:

A low resolution map of the MHAS2168-Mb_H2 was obtained from a smaller screening dataset that was acquired similarly to that described in the Methods section. We have stated this in the Methods section for clarity as well:

“Template picking was used to pick particles from the micrographs; the templates were produced from an earlier lower-resolution map of MHAS2168-Mb_H2 complex from a smaller screening dataset obtained similarly.” (Lines 639-642)

Reviewer #4 (Remarks to the Author):

Remm et al. present a very interesting a timely study on the mechanism of triacylglyceride extraction and transport from the mycobacterial inner membrane. By combining structural biology, an impressive arrays of mutations and large-scale MD simulations, the authors propose a novel description of such extraction, and pinpoint critical residues involved in this process, providing mechanistic insights. The paper contains an impressive amount of data, and in general the conclusions are sound and supported by the data. I have however several comments about the presentation and interpretation of MD simulations, that, in my opinion, should be amended in the revised version.

Authors' reply:

We thank the reviewer for the positive and encouraging comments. As is explained in more detail below, spawned by the questions and suggestions of the reviewer, we performed additional MD simulations, molecular docking, and analyses, which we included in the revised version of our manuscript.

The author performed a large amount of MD simulations to essentially study two aspects of TAG transport by Rv1410/MHAS2168: i) extracting of TAG from the membrane (transporter in the inward facing orientation), and ii) loading/channeling of a TAG molecule into LprG (transporter in the outward facing orientation). Overall, the simulations seem to be well planned, prepared and executed, however some details are missing. Only after reading the details of MD simulations in the SI, more questions arise. It does leave an impression that the authors picked the observations that supported their hypothesis in the main text, and do not discuss possible alternative scenarios, that seem to be plausible

based on the MD report available in the SI. The results and description of the MD simulations are scattered across the main text, methods, extended data Figures and SI report. In my opinion it makes the manuscript hard to follow.

Authors' reply:

We decided to present the MD results together with the experiments in an “integrative” manner, instead of having two distinct sections (which then run the risk of being somewhat disconnected). We hope that the revised version of our manuscript is now easier to follow for the reviewer.

Comments relating to the i) set of simulations: extracting of TAG from the membrane (transporter in the inward facing orientation).

- The authors do not record any spontaneous insertions of TAGs to the transporter main cavity, even though it is a critical part of the proposed transport mechanism. In my opinion this should be investigated further. The authors say it is because of low TAG concentration they used (only 2 molecules)

Authors' reply:

The reviewer is correct. In our MD simulations we did not observe spontaneous full entrance events of TAG molecules within the transporter main cavity, neither for the outward-facing MHAS2168 X-ray structure nor for the inward-facing MHAS2168 homology model. Nevertheless, we cannot rule out that such event might take further simulation time. In addition, as the reviewer#4 correctly mentioned in the questions below, a higher TAG membrane concentration might be needed. An additional limitation is the usage of a coarse-grained force-field that does not enable to sample large conformational changes of the protein. Such structural rearrangements might be required to allow TAG molecules to enter within the main transporter cavity.

- why not try some simulations with a larger concentration? (even if it's larger than physiological one, for this purpose it would be justified). Moreover, for a separate simulation set (MHAS2168IN-TAG), where the authors placed a TAG molecule in the central cavity (if I understood correctly), the authors are listing the residence times, that vary 8-74 us. That implies that the TAG molecule escaped the central cavity? This is unfortunately not reported at all. If TAG escaped the cavity to the membrane, that would provide a pathway for TAG to enter/exit the cavity? If TAG escaped to the water phase, that could imply that it binds to the cavity from there? Please do explain. All this should be also discussed in the context of using a homology model for the inward facing state, that might lead to inaccuracies and, for example, a possibility of alternatives scenarios allowing TAG to bind to the central cavity, such as larger conformational changes of involved helices.

Authors' reply:

Regarding performing MD simulations with a larger TAG concentration, we now carried out additional calculations (for results of these simulations, please see below).

The reviewer is correct that the TAG entrance/exit path is an important aspect that should be discussed in the text. In our MHAS2168^{IN-TAG} system the TAG molecule initially positioned within the main cavity, indeed escaped after 20, 15, 18, 8 and 74 μ s, repeats 1 to 5, respectively. In all the MHAS2168^{IN-TAG} simulated repeats, the TAG initially positioned inside the main cavity escaped from the portal encompassed by the helices 5 and 8. In particular, the majority of the contacts involved helices 5 (N-terminal part), 8 (C-terminal part), 10 (C-terminal part) and 11 (N-terminal part) (see Table S4).

We agree with reviewer#4 that protein residues that contact the TAG molecule in its way out from the main central cavity, could also provide a pathway for TAG to enter the cavity. For this reason, in the MHAS2168^{IN-TAG} system we determined the protein residues possibly involved in TAG-exit/entry, but we did not incorporate them in our manuscript. We now added the list of protein-TAG contacts in the Table S4 (page 5, see also below). We never observed TAG escaping to the water phase.

[...] continue from Table S4						
MHAS2168 interactions with TAG in the MHAS2168^{IN-TAG} system ^h						
Leu149	Ala146	Gly150	Leu333	Gly336	Ala400	Arg323
Trp337	Thr403	Ser404	His143	Arg145	Ala147	Leu407
Arg269	His415	Ser419				
(h) The residues listed have TAG contacts above a threshold of 40% with respect to the residue that has the maximum number of TAG contacts.						

We agree with the reviewer that the usage of a homology model for the inward-facing conformation might introduce limitations/uncertainties in our interpretations, and consequently alternative scenarios should not be excluded. As mentioned by the reviewer, large conformational changes of the proposed portal helices as well as changes involving helices TMA and TMB, which are connected by extended and likely flexible unstructured regions, could provide alternative scenarios for TAG membrane extraction and or escape. Even though the coarse-grained approach did not allow us to study such large conformational changes, we believe our model is instrumental and informative to show how the substrate TAG interacts with MHAS2168/RV1410 and to start investigating possible mechanisms of extraction and transport across the membrane.

We have added to the Discussion to address this question:

“However, TAG was never observed to spontaneously enter into the IF cavity on the MD-simulation time scale, which could be due to limitations of the homology model of the IF conformer, in particular concerning the precise widths of the lateral openings. In contrast, exit events were found in the MHAS2168^{IN-TAG} simulations at the TM5-TM8^{IN} cleft, thus mapping a potential pathway for TAG to exit (or, reversely, to enter) the inward-facing cavity (Fig. S12).” (See lines 317-322)

In addition, we have performed TAG docking analysis, described in Supplementary Information:

“In order to expand our investigation and to support our approach of positioning one TAG molecule within the central cavity, we carried out a molecular docking study on both the outward-facing and the inward-facing conformations. The docking procedure is reported in the Method section. Briefly, AutoDock-Vina^{5,6} was used, with the transporter (i.e., the receptor) considered rigid. One molecule of TAG (i.e., the ligand) was docked, which was considered to be flexible in AutoDock-Vina. For the outward-facing conformation, our results show a consistent preferential hotspot where all the poses were found nearby helices 7, 11 and facing helices A and B (Fig. S13a). However, no TAG was docked inside the central cavity. For the inward-facing conformation, three distinct hotspots were found: two towards the periplasmic leaflet and on both sides of the transporter, and one within the central cavity (see Fig. S13c), thereby confirming that our inward-facing structural model can accommodate one molecule of TAG within the central main cavity. Consistent with the outward-facing conformer, TAG poses were found nearby helix 11 and facing helices A and B (Fig. S13c), suggesting that this could be a potential “reservoir” where TAGs might accumulate in between transport cycles. In addition, the docking analysis reveals hotspots (Fig. S13, dotted circles) in close proximity to residues identified in contact with TAG during the MD simulations.

To extend our analysis, we then performed a further cycle of molecular docking accounting for flexibility not only of the TAG molecule, but also of residues Glu157 and Arg426, which are located within the main cavity and hypothesized to be a candidate locus for H⁺ translocation. All the rest of the protein was considered rigid. For the inward-facing conformation, we found the same hotspots (Fig. S13d). However, for the outward-facing conformation, in addition to the hotspot nearby helices 7, 11 and TMA-TMB, one TAG molecule was found within the main central cavity (Fig. S13b). Overall, the docking results strengthen our initial guess/assumption that the MHAS2168 main central cavity can accommodate one TAG molecule.”

Figure S13. Results of the molecular docking of TAG on the MHAS2168 structure. (a) Docking results for the MHAS2168 in outward-facing conformation (TAG is flexible; protein residues are rigid). The N- and C-terminal parts of the transporter are in ribbon and colored in light gray and dim gray, respectively. A and B linker helices are in yellow. Docked TAG molecules are shown in ball and stick representations and colored rainbow from blue to red. The dotted circle indicates the docking hotspots. The C α atoms of residues found in contact with TAGs in the CG-MD simulations are colored pink, while residues found in contact with TAGs in both the outward- and inward-facing conformations are colored brown (see Table S4). (b) As in (a), but in addition to TAG, protein residues Glu157 and Arg426 were flexible during docking. (c) As in (a), but for MHAS2168 inward-facing conformation. (d) As in (a), but for MHAS2168 inward-facing conformation and, in addition to TAG, protein residues Glu157 and Arg426 were flexible during docking.

- TAG molecules (or similar ones) had been previously found before to aggregate in the membrane core (Khandelia et al., PLOS ONE 2010). If the same occurs in the mycobacterium membrane, then several scenarios of TAGs entrance to the transporter are possible. Do single molecules enter or an aggregate is necessary? Do the aggregates leave the membrane and then TAGs bind the transporter from the aqueous phase (see previous comment?) I don't expect the authors to try all these hypotheses, but some alternatives should be at least mentioned in the revised manuscript.

Authors' reply:

We thank the reviewer for their suggestions and for directing us to the work of Khandelia and coworkers, which we had indeed overlooked (it is now cited in our revised manuscript, together with more recent related work by Zoni et al. 2021). We followed the suggestion of the reviewer and carried out additional CG-MD simulations in which we included a larger number of TAGs in the membrane (7%, corresponding to 14 TAG molecules overall), previously was 1% TAG (2 molecules). Simulations were carried out under the same conditions as previously for both the outward-facing and the inward-facing conformations. For each system, five different repeat simulations of 50 μ s each were carried out, initiated with different random starting velocities.

Indeed, as reported by Khandelia et al. 2010, PLOS ONE and Zoni et al. 2021, eLife, the TAGs clustered in the bilayer core and formed a "micelle-like" aggregate that was preferentially located close to the membrane midplane, see Figure 2 below.

Figure 2. Coarse-grained MD simulations of MHAS2168 in presence of 7% concentration of TAGs. (a) Two-dimensional density maps of TAGs over five concatenated repeat simulations for the MHAS2168 in the outward-facing (left panel) and inward-facing (right panel) conformations, from blue (no density) to red (high density). (b) Snapshots from CG-MD simulations showing how TAG can diffuse in the membrane bilayer or form membrane midplane aggregates as previously reported (Khandelia et al. 2010, Zoni et al. 2021).

These additional simulations were carried out for both MHAS2168 conformers, inward-facing and outward-facing. However, despite the extensive sampling time of 0.25 ms for both the outward-facing and inward-facing conformers, we did not observe a TAG entering event into the central cavity of the transporter, although the TAG clusters did sample the periphery of the transporter (including the putative lateral gate regions). TAGs were never observed to dissolve into the water. In summary, while we cannot exclude the possibility that an even higher (but then probably rather unrealistic?) concentration of TAGs in the bilayer would change this picture, we conclude from our simulations that there is at least a substantial energy barrier for TAGs to enter into the central MHAS2168 cavity, which cannot be surmounted in the simulations within the accessible sampling time. Therefore, as the results

of these additional control simulations do not contribute much to the overall mechanistic conclusions, we would prefer not to include them in the manuscript.

- The authors see the entering of other lipids to the central cavity, but then rather ignore it or even delete the molecules that did that. What about the possibility that it might be functionally relevant?

Authors' reply:

Indeed, the POPE and POPG lipids that entered the central cavity during the equilibration phase were removed. This was done because we could not exclude that the entering of the lipids was related to the position restraints that were imposed on the protein during the equilibration. However, we do not think that this was crucial, because during the subsequent production simulations, lipids do (again) access the central cavity, see Figure S11. This lipid access is rather dynamic, but for the inward-facing conformation (Figure S11b), the cavity is indeed constantly occupied (by different lipids).

The cavity seems to be large enough to accommodate more than one lipid (is that true?). Perhaps constant occupation of the cavity by a lipid(s) increases the hydrophobicity of it, allowing for easier TAG binding?

Authors' reply:

That's an interesting hypothesis. In general, the cavity is large enough to host a single phospholipid molecule (or a single TAG molecule, see also our docking results shown in Figure S13). When a phospholipid is fully bound (both tails in), an additional one might occasionally put one of its tails also inside the cavity, which then harbors three tails in total (as in TAG), but two fully bound phospholipids were not observed due to the size of the cavity. Thus, the possibility of "recruitment" of TAG by a pre-bound phospholipid is fully included in our simulations (but TAG binding does not happen spontaneously, nevertheless). Of course, this could be linked to either the homology model of the inward-facing conformer or the restricted conformational flexibility of the protein in the CG force field (or to both these aspects together). See also answer to the question above.

Comments relating to the ii) set of simulations: loading/channeling of a TAG molecule into LprG (transporter in the outward facing orientation)

- The second critical part of the mechanism proposed in the current manuscript, is the formation the E147-R417 ion lock at the bottom of the central cavity, that "lifts TAG toward the periplasmic leaflet". This is a quite plausible scenario. I see here however a missed chance for MD simulations: to really prove this hypothesis, simulations with mutations in residues forming this lock would be particularly useful.

Authors' reply:

This would indeed be an interesting next step to look at, but would require all-atom simulations, which are very time consuming. We therefore think that this reviewer request goes beyond the scope of this study.

The authors say in the introduction that “An LprG cross-linking study in live *Mycobacterium smegmatis* cells did not demonstrate a reproducible interaction between LprG and Rv141020 and attempts to show complex formation in vitro with purified proteins were fruitless.” and then further argue that “transient low affinity interactions might occur in the cellular context.” The authors then decided to prepare a model of the LprG and Rv141020 complex using ColabFold to study TAG loading into LprG. It is not clear from the Methods section whether the two proteins (LprG and MHAS2168) were made to interact with each through harmonic potentials or not; this sentence is not clear: “For the MHAS2168OUT-LprG complex, 134 (out of a total of 3698) intra- and inter-molecular elastic network bonds were removed to avoid a too rigid protein-protein interface in the MHAS2168-LprG complex.” If there is no additional forces to keep LprG bound to MHAS2168 in MD simulations, how the authors reconcile the fact that the complex is seemingly stable for at least 100us, with “transient low affinity interactions”?

Authors' reply:

The reviewer is correct in that, in our initial set of CG-MD simulations, harmonic bonds (the “elastic network”) were used not only within the transporter and LprG, but also some bonds between these two proteins. While the use of such an elastic network is (unfortunately) required in CG-Martini simulations of proteins (and therefore considered to be the “default”), the definition of inter-protein elastic bonds might be considered to be somewhat arbitrary, especially in light of the fact that this is done on the basis of the distances between pairs of backbone beads, which for the MHAS-LprG intermolecular contacts solely result from the AlphaFold prediction. Therefore, we carried out additional control simulations in which we completely removed all inter-molecular (that is, between MHAS2168 and LprG) elastic bonds, see Figure 3 below.

Figure 3. Elastic bond network used in CG-MD simulations of the MHAS2168-LprG complex. The removed intramolecular and intermolecular network are indicated in orange and cyan, respectively. The final elastic network bonds used in our initial CG-MD simulations is indicated in green. In these simulations, a total of 8 intermolecular bonds were finally kept. In our additional control CG-MD simulations of the MHAS2168-LprG complex, we removed all intermolecular bonds, including the 8 mentioned above (the network is indicated in purple).

Reassuringly, also in these simulations, TAG loading from MHAS2168 into LprG was observed, supporting the robustness of our findings (see Figure S15 below).

Figure S15. TAG transfer into LprG in the control MD simulations of the MHAS2168^{OUT}-LprG complex without MHAS2168-LprG elastic network bonds. (a) Final snapshots of the ten control simulations (consecutively numbered) carried out for 30 μs each and with different starting velocities. Simulations started with TAG in a 2-tails-up configuration and after the equilibration phase of the MHAS2168^{OUT}-LprG system. (b) The cyan line is the z-coordinate of the average center of mass of the phosphate groups of the upper membrane leaflet. The colored lines (as in (a)), depict the z-coordinate of the center of mass of the TAG molecule. The inset models of the MHAS2168^{OUT}-LprG complex shown in the panel indicate the position of the TAG (orange shade).

The dissociation of LprG from MHAS2168 was never observed in any of the simulations (on the 0.3 ms time scale). However, this does by no means imply that there is a high-affinity interaction, but only that the complex represents a local(!) minimum on the free energy landscape. In principle, we should not exclude that the stability of the heterodimer might be rather short-lived and transiently stabilized by the presence of TAGs within the interface of the two proteins, thus particularly hard to functionally validate. We think it is fair to say that, on the experimental time scale, 0.1 ms (for the simulations with the intermolecular elastic bonds, Figure 6 in main text) and 30 μs (for the control simulations, Figure S15 above) can still be called “transient” for a protein-protein complex, so we do not see a contradiction there. In order to draw any conclusions concerning the affinity of LprG towards MHAS2168 from the simulations, extensive free-energy simulations would need to be carried out, which are beyond the scope of this work.

Finally, concerning the fact that in our initial set of CG-MD simulations of the MHAS2168-LprG complex, 134 (out of a total of 3698) intra- and inter-molecular elastic network bonds were removed, we now rephrased that part in the Method section to increase clarity.

“In order to avoid a too rigid protein-protein interface, 112 intramolecular (7 within MHAS2168 and 105 within LprG) and 22 intermolecular (between MHAS2168 and LprG) harmonic potentials (i.e., the Martini elastic network) were removed. A total of 8 elastic network bonds were kept between MHAS2168 and LprG. A set of ten independent repeat simulations of 30 μ s each and without any intermolecular elastic network bonds were also run as a control (see Table S3). These control simulations confirm the proposed TAG transfer mechanism (see Fig. S15).” (Lines 703-705)

We have added Figure S15 and the following text to the Supplementary Information:

“In our initial set of CG-MD simulations, harmonic bonds (the “elastic network”) were used not only within the transporter and LprG, but also some bonds between these two proteins. While the use of such an elastic network is by default required in CG-Martini simulations of proteins, the definition of inter-protein elastic bonds might be considered to be somewhat arbitrary, especially in light of the fact that this is done on the basis of the distances between pairs of backbone beads, which for the MHAS-LprG intermolecular contacts solely result from the AlphaFold prediction. Therefore, we carried out additional control simulations in which we completely removed all inter-molecular (that is, between MHAS2168 and LprG) elastic bonds (Fig. S15). Reassuringly, also in these simulations, TAG transfer from MHAS2168 into LprG was observed, supporting the robustness of our findings (Fig. S15).

It is noteworthy that the dissociation of LprG from MHAS2168 was never observed in any of the simulations (on the 0.3 ms time scale). However, this should not be interpreted as the existence of high-affinity interaction between MHAS2168 and LprG, but rather that the complex represents a local minimum on the free energy landscape. The heterodimer might nevertheless be rather short-lived and might be transiently stabilized by the presence of TAGs within the interface of the two proteins, which makes it hard to experimentally validate this interaction. The experimental time scale, 0.1 ms (for the simulations with the inter-molecular elastic bonds, Fig. 6) and 30 μ s (for the control simulations, Fig. S15) still accounts for a “transient” protein-protein complex, so these findings do not directly contradict previously reported lack of strong physical interactions between the proteins^{3,7} or our data from the operon shuffling experiments (Fig. 5d-h).”

Minor comments:

Extended Data Figure 7 should be ideally replaced by the 2D density plots for each lipid around the protein, especially if, as the authors argue, an annular lipid belt of specific lipids is created around the

transporter.

Authors' reply:

We thank the reviewer for their suggestion. In the main text we wanted to say that preferential phospholipid interactions happen with some lipids rather than others. We have now changed the section extensively (see lines 199-209). Regarding the Figure S7, we prefer to report the protein-lipid interaction with the usage of histograms and associated pinpoint residues listed in the Table S4. Even though 2D density plots provide a broad overview of the overall localizations of lipids, we believe in this way the data are explicit and easier to interpret for the readers.

In the SI the authors talk about “snorkelling events” for TAG molecules and “entry/exit events” for other lipids, referring to the SI Figure 2. I cannot easily see how one distinguish between these two types of events from such plots. In the caption, the same definition is given both types of events. Surprisingly there is no events at all in the conformation, at which TAGs are expected to bind (inward facing, see main comments).

Authors' reply:

The reviewer is right because the plot shown in Figure S11 does not record snorkelling events, and we apologise if the text that referred to Figure S11 was confusing. In our analysis, an entrance event is recorded when the phosphate bead of a phospholipid or the TAG backbone bead overlap with the central cavity volume shown in Fig. S10. During the MD simulations of the outward-facing conformation, such events have been recorded during our analysis (as shown in the Figure S11), but they only involved the periphery of the volume shown in Figure S10. Visual inspection of the trajectories confirmed that no entrance events occurred in the simulations of the outward-facing conformation, neither for TAG nor for the phospholipids. We now rephrased the text accordingly (see page 31, Supplementary Information):

“Furthermore, we recorded the entrance events of the different lipid species into the central cavity of MHAS2168 during MD simulations (Fig. S11). In our analysis, an entrance event is recorded when the phosphate bead of a phospholipid or the TAG backbone bead overlap with the central cavity volume shown in Fig. S10. Apart from a few snorkelling events, in which a TAG molecule probes the periphery of the cavity but does not fully enter, the MHAS2168^{OUT} cavity is little accessed by phospholipids and TAGs. Visual inspection of the trajectories confirmed that no entrance events occurred in the simulations of the outward-facing conformation, neither for TAG nor for the phospholipids.”

Conversely, in the simulations of the inward-facing conformation, we observed full entrance events of phospholipids, but we did not observe any TAG entrance events. We believe this is because the main cavity is mainly occupied by phospholipids that compete with TAG, which is indeed very hydrophobic. This was also observed in our control simulations where the TAG concentration was increased to 7%.

there are multiple models of cardiolipin available in Martini 2.2, please specify which was used

Authors' reply:

The reviewer is correct, we did not specify the cardiolipin model we used. The Martini force-field model is the cardiolipin with minus 2 charge (both the phosphatidyl groups are charged). The ID name is CDL2 (see M. Dahlberg 111:7194–7200, 2007, JPC-B and Dahlberg and Maliniak, 6:1638-1649, 2010, JCTC). This has now been specified in the Method section more precisely.

it is not clear what was the exact chemical structure of the TAG molecule used in simulations, and which Martini model was selected to describe it. Please specify.

Authors' reply:

Thank you for spotting this. The TAG model used in the present work is from Vuorela et al., 2010. More precisely, the model represents a glycerol with three C18:1 oleoyl tails. This has now been specified in the Method section more precisely.

Throughout the text and in the caption of Figure 1, the authors use the expression “non-proteinaceous density”, whereas the Figure 1 says “Lipid” in red. That seems too unambiguous.

Authors' reply:

We agree with the reviewer that our labelling of Figure 1 is too unambiguous. We call it now also in the Figure 1 “non-proteinaceous density”.

In the main text, we provide the following explanation:

“Intriguingly, we detected a non-proteinaceous density, most likely representing a lipid, at the transporter’s side wall (Fig. 1a).” (Lines 90-91)

The wording in the last two paragraphs on the page 7 is somewhat confusing. First the authors argue that “Functional assays (Fig. 4e,f) suggest that TAGs enter the cavity through both TM5-TM8 and TM2-TM11 lateral openings in the inward-facing state” to later say “Even though the narrow TM5-TM8 lateral cleft in the OF state is connecting the central cavity and membrane, it does not seem to play a role in TAG transport.” I think that the authors want to say that the TM5-TM8 cleft plays a role in TAG transport only in the inward-facing state, but it is somewhat confusing in the current formulation.

Authors' reply:

Thank you for directing our attention to this ambiguity in wording. The reviewer has grasped the concept very well, but we understand that it might be difficult to follow the argument because two different clefts are lined by the same helix pair in both IF and OF conformation. We have altered the designation of the four lateral clefts in the following manner in both the main text and the Supplementary Information to emphasize that the transporter sports four different clefts: TM5-TM8^{OUT}, TM5-TM8^{IN}, TM2-TM11^{OUT}, TM2-TM11^{IN} and keep the descriptions more consistent. We hope that this helps the readers. Please see the following larger changes in the main text:

“According to our MD simulations, TAGs accumulate in the membrane core and therefore must enter the transporter through lateral openings between N- and C-domains lined with TM5 and TM8, or TM2 and TM11 in either the IF or OF conformation, respectively (Fig. 4a,b). To test whether any of the four lateral crevices could serve as TAG entry or exit sites, residues in the middle of each lateral cleft were mutated into glutamates or aspartates to introduce polarity and block the opening (Fig. 4a,b,c; see Supplementary Note 1).” (Lines 219-224)

“Functional assays (Fig. 4e,f) suggest that TAGs enter the cavity through both TM5-TM8^{IN} and TM2-TM11^{IN} lateral openings, as the corresponding mutants G140D_{Mtb}, G150D_{MH}, A411D_{Mtb}, and A420D_{MH} were inactive. As expected due to the obstruction by the linker helices, the TM2-TM11^{OUT} lateral crevice is not involved in lipid transport as the L422E_{Mtb} and L431E_{MH} mutants display comparable growth to the wild-type operons at both lower (0.1 µg/ml) and higher (0.4 µg/ml) vancomycin concentrations. Likewise, mutations in the TM5-TM8^{OUT} lateral cleft (namely L155E_{Mtb} and I165E_{MH}) were well tolerated at 0.1 µg/ml vancomycin concentration, suggesting that this cleft does not seem to play a role in TAG transport despite the fact that it is connecting the central cavity and membrane (Fig. 4e,f).” (Lines 225-233)

And in the Supplementary Information (Supplementary Note 1, p 28):

“However, we decided to introduce mutations to each of the four lateral clefts, forming on opposite sides of the transporter in both outward-facing and inward-facing conformations, assessed in our MHAS2168^{OUT} crystal structure and MHAS2168^{IN} homology model.”

Table S7. Mutations introduced to lateral openings between N- and C-domains in Rv1410 and MHAS2168.

Lateral cleft	Mutation in Rv1410	Mutation in MHAS2168	Helices lining the lateral cleft	The lateral cleft is open in
TM5-TM8 ^{OUT}	L155E _{Mtb}	I165E _{MH}	TM5-TM8	Outward-facing state
TM5-TM8 ^{IN}	G140D _{Mtb}	G150D _{MH}	TM5-TM8	Inward-facing state
TM2-TM11 ^{OUT}	L422E _{Mtb}	L431E _{MH}	TM2-TM11	Outward-facing state
TM2-TM11 ^{IN}	A411D _{Mtb}	A420D _{MH}	TM2-TM11	Inward-facing state

REVIEWERS' COMMENTS

Reviewer #1 (Remarks to the Author):

The authors have adequately addressed all my concerns.

Reviewer #2 (Remarks to the Author):

In this revised manuscript, Remm et al. address a number of the points raised by us and other reviewers. However, a fundamental issue remains with interpretations of the literature and of data in this study. In the most general sense, there remains a concerning disconnect in the manuscript between experimental observations and conclusions. In some cases, the authors cite evidence, but then link them to conclusions that the cited studies themselves did not make and that based on the reported data, are overinterpretations (see further below).

As with the original manuscript, the solved structures and associated analyses are of fundamentally high value to the field; the mutational data also contribute to our understanding of the mycobacterial transporter Rv1410c. The concerns raised in our original review and in our discussion below would be easily addressed by the use of appropriately qualified language. The associations between mycobacterial biology and biochemistry are often indirect because appropriate assays are not available— as noted below, direct measurements of processes like lipid transport that relate directly to LprG/Rv1410c function are limited or not available, and reported bacterial phenotypes are often only consistent with, rather than direct evidence of, protein function. While it requires additional effort to incorporate such caveats, we strongly feel that this is paramount to scientific accuracy and rigor, especially for presenting this detailed study and its implications accurately to the broad audience of Nat Comm.

We discuss a few examples below, but our assessment of this revision is based on statements of this kind throughout the manuscript. (All line numbers below refer to the unmerged manuscript file; the line numbers appear to be -1 in the merged version.)

In the rebuttal, the authors state, “we consider our indirect way of measuring transporter activity by the vancomycin sensitivity assay as a good surrogate to quantify the amount of TAG being present in the outer mycobacterial membrane. As we have shown in a previous paper (Hohl, Remm et al., Mol Micro, 2019), deletion of the *M. smegmatis* rv1410-lprG operon results in increased permeability of the outer mycobacterial membrane, as we have shown based on BCECF-AM fluorescence measurements conducted in the presence and absence of the proton gradient uncoupler CCCP.”

The experimental evidence does not support this proxy. What the authors present here is correlation and not causation. First, there is no experimental evidence in the literature that vancomycin sensitivity is correlated with the amount of TAG in the mycomembrane. Further, increased uptake of the fluorescent probe is not equivalent to increased uptake of vancomycin, which also has not been directly measured (even though the authors make a statement to this effect in the manuscript, lines 138-139-- “The deletion of *lprG/rv1410c* homologous operon 138 MSM3070/69 in *M. smegmatis* results in increased vancomycin influx to the periplasm”-- the cited papers report MICs and other sensitivity data, but do not include accumulation or uptake measurements). Further, even if there were increased uptake of vancomycin, this is not necessarily equivalent with vancomycin sensitivity, which is a growth phenotype that is the complex downstream outcome of vancomycin activity.

Line 54: “more recent work has shown that Rv1410 indirectly contributes to drug tolerance by sealing the mycomembrane with triacylglycerides (TAGs).”

The first half of this sentence is supported for *M. smegmatis* by Hohl et al. as cited (although the authors are encouraged to use the word “tolerance” carefully, as this has a particular meaning in the field; they may mean instead sensitivity or resistance). However,, they connect this with a TAG accumulation phenotype in *Mtb* from the cited Martinot et al study. Importantly, this phenotype has not been confirmed in *Msm*. Neither paper shows that TAGs are responsible for changes in cell envelope integrity (since Hohl et al. examined uptake into the cytosol and did not distinguish permeability of the mycomembrane). As stated, this sentence is an example of oversimplification and overinterpretation of a correlation.

Lines 56-58, “operon overexpression led to elevated secretion of TAGs into the culture medium, hence directly demonstrating that Rv1410 is a TAG transporter.” This is an inference, not a direct demonstration. Elevated secretion of TAGs could arise from excess TAG at the mycomembrane, but this could be an indirect effect of overexpressing *lprG-rv1410c*, e.g., as a stress response to gain of function in this operon.

Line 133: “Rv1410 and *LprG* have been shown previously to extract TAGs from the inner membrane so that TAGs could reach the mycomembrane where they secure its impermeability”

We agree that the literature cited (refs 8 and 12) support an inferred model. However, the statement “shown... to extract” implies that this was experimentally observed. This is the case for *LprG*, which was tested in a model *vitro* system with soluble truncated protein, but is not true for Rv1410c, for which no measurements of TAG transport/extraction have been published. Also, as above, there is no experimental evidence that TAG itself contributes to the impermeability of the mycomembrane, only that the loss of *lprG-rv1410c* results in permeability defects and antibiotic sensitivity, which could be due to pleiotropic effects of genetic modification on cell envelope content and structure.

Reviewer #3 (Remarks to the Author):

The authors have thoroughly addressed all my concerns, and I have no further comments.

Reviewer #4 (Remarks to the Author):

The authors addressed most of my comments and performed a lot of additional simulations and analyses. I therefore have only minor comments that should help the authors to polish the manuscript. I do not need to be included in the next round of reviews, if it happens.

- The authors write "However, no TAG molecule entered the cavity in the MD simulations, likely because of the much lower TAG concentration compared to the competing phospholipid species." (pages 31-32 in the SI). The authors performed simulations at higher TAG concentration, as I suggested in the first round, but opted not to include these data in the final manuscript, saying

"Therefore, as the results of these additional control simulations do not contribute much to the overall mechanistic conclusions, we would prefer not to include them in the manuscript."

I find this situation not optimal. I'd suggest to include these results in the SI, not only because it's an interesting result on its own, but also to not potentially waste the time and resources of other researchers that might want to study systems containing higher TAG concentrations.

- The authors mention that "each lipid species and solvent separately coupled to an external bath using the v-rescale thermostat" this is somewhat unexpected, the authors might want to clarify why multiple thermostats were used.

- Figures S7, S9 and S12 both show TAG-protein contacts, in figure S7 this number is normalized but not in S9 and S12. I'd suggest to make these figures in a consistent manner to not confuse the reader.

We wish to thank reviewers #2 and #4 for their additional comments on our revised manuscript and we are pleased to see that we could address all concerns raised by reviewers #1 and #3.

To react on the remaining points raised by reviewer #2, we have toned down statements throughout the manuscript and included a cautionary note to the discussion, saying that next to TAG transport, Rv1410-LprG might also fulfil additional, yet undiscovered functions, by transporting additional lipid-like substrates that might have escaped detection in a previous lipidomics study (Martinot et al. 2016, PMID: 26751071). Further, we acknowledge that the molecular reasons underlying our indirect vancomycin sensitivity assay may be more complex than assumed, and might involve mechanisms that go beyond the lack of TAG being inserted to properly seal the mycomembrane. We have made adjustments to the manuscript that clarify this point.

Reviewer #2 (Remarks to the Author)

In this revised manuscript, Remm et al. address a number of the points raised by us and other reviewers. However, a fundamental issue remains with interpretations of the literature and of data in this study. In the most general sense, there remains a concerning disconnect in the manuscript between experimental observations and conclusions. In some cases, the authors cite evidence, but then link them to conclusions that the cited studies themselves did not make and that based on the reported data, are overinterpretations (see further below).

As with the original manuscript, the solved structures and associated analyses are of fundamentally high value to the field; the mutational data also contribute to our understanding of the mycobacterial transporter Rv1410c. The concerns raised in our original review and in our discussion below would be easily addressed by the use of appropriately qualified language. The associations between mycobacterial biology and biochemistry are often indirect because appropriate assays are not available— as noted below, direct measurements of processes like lipid transport that relate directly to LprG/Rv1410c function are limited or not available, and reported bacterial phenotypes are often only consistent with, rather than direct evidence of, protein function. While it requires additional effort to incorporate such caveats, we strongly feel that this is paramount to scientific accuracy and rigor, especially for presenting this detailed study and its implications accurately to the broad audience of Nat Comm.

We discuss a few examples below, but our assessment of this revision is based on statements of this kind throughout the manuscript. (All line numbers below refer to the unmerged manuscript file; the line numbers appear to be -1 in the merged version.)

In the rebuttal, the authors state, “we consider our indirect way of measuring transporter activity by the vancomycin sensitivity assay as a good surrogate to quantify the amount of TAG being present in the outer mycobacterial membrane. As we have shown in a previous paper (Hohl, Remm et al., Mol Micro, 2019), deletion of the *M. smegmatis* rv1410-lprG operon results in increased permeability of the outer mycobacterial membrane, as we have shown based on BCECF-AM fluorescence measurements conducted in the presence and absence of the proton gradient uncoupler CCCP.” The experimental evidence does not support this proxy. What the authors present here is correlation and not causation. First, there is no experimental evidence in the literature that vancomycin sensitivity is correlated with the amount of TAG in the mycomembrane. Further, increased uptake of the fluorescent probe is not equivalent to increased uptake of vancomycin, which also has not been directly measured (even though the authors make a statement to this effect

in the manuscript, lines 138-139– “The deletion of *lprG*/*rv1410c* homologous operon 138 MSM3070/69 in *M. smegmatis* results in increased vancomycin influx to the periplasm”-- the cited papers report MICs and other sensitivity data, but do not include accumulation or uptake measurements). Further, even if there were increased uptake of vancomycin, this is not necessarily equivalent with vancomycin sensitivity, which is a growth phenotype that is the complex downstream outcome of vancomycin activity.

The observed phenotype is undisputable: genetic deletion of *rv1410* and/or *lprG* results in a vancomycin sensitivity phenotype. Hence, we assume that the reviewer agrees that the assay by which we analyzed the Rv1410 mutants is sound *per se*. That is also evident in our assays, as we include empty vector negative controls as well as a control mutation (Dton mutation) that abrogates MFS transporter function in all assays, consistently showing this vancomycin sensitivity phenotype.

The way we understand the points raised by reviewer#2, the controversy touches on the question why the deletion of *rv1410* results in vancomycin resistance. We wish to note, that this is a dispute that has been discussed in previously published literature over the last two decades.

The initial explanation for vancomycin sensitivity upon genetic deletion of *rv1410* was that Rv1410, maybe in concert with the lipoprotein or also alone, is a drug efflux pump; hence it would directly pump out vancomycin from the cell (PMID: 11181364). In a recent study (PMID: 30742339), we provided experimental evidence showing that direct vancomycin efflux mediated by Rv1410 is very unlikely.

In recent years, there was growing evidence that Rv1410-LprG act together to export triacylated lipids, namely TAGs and LAMs to the outer mycobacterial membrane, and that the drug sensitivity patterns observed upon deletion of *rv1410* are an indirect effect, caused by alteration of the outer mycobacterial membrane.

Evidence in favour of this latter mechanism are:

- 1) LprG was shown by X-ray crystallography to bind triacylated lipids, but not drugs (PMID: 20694006).
- 2) Untargeted lipidomics analyses showed that gene deletion mutants of *rv1410* accumulate TAGs, whereas overexpression of the *rv1410-lprG* operon leads to increased TAG secretion into the culture medium (PMID: 26751071).
- 3) We have shown in a previous paper (PMID: 30742339) that the outer membrane of *M. smegmatis* lacking *rv1410* is more permeable for the dye BCECF-AM, which in our interpretation is a proxy to explain how vancomycin can reach its target structure (peptidoglycan precursor) in the periplasm more easily.
- 4) Growth defects and surface structure alterations of mycobacteria upon deletion of *rv1410* have been early recognized and have been attributed to a more general role of this transporter in the biosynthesis of the outer mycobacterial membrane by virtue of lipid transport (PMID: 19564371, PMID: 25232742).

Indeed, we think that the above mentioned papers (all published previously to this work and all cited in our manuscript) collectively support a model wherein Rv1410-LprG transport TAGs and (likely further lipid species), and that vancomycin susceptibility is an indirect consequence of these lipid transport defects.

But the reviewer has certainly a point in saying that “associations between mycobacterial biology and biochemistry are often indirect because appropriate assays are not available– as noted below,

direct measurements of processes like lipid transport that relate directly to LprG/Rv1410c function are limited or not available, and reported bacterial phenotypes are often only consistent with, rather than direct evidence of, protein function.

In other words, it is certainly correct to say that our current understanding how a deletion of *rv1410* and/or *lprG* impacts cellular physiology and vancomycin sensitivity might be not solely attributable to TAG transport deficiency, and that other still elusive mechanisms might play a substantial role as well.

To address these concerns, we have reformulated the results section where the vancomycin resistance assay is explained (lines 133 ff).

Deletion of lprG/rv1410c homologous operon MSM3070/69 in M. smegmatis results in increased vancomycin susceptibility, a phenotype we exploited here to study the transport activity of Rv1410 and MHAS2168 mutants. According to more recently published mechanistic models^{8,12}, deletion of lprG/rv1410c results in diminished TAG transport to the mycomembrane, which in turn becomes more permeable to vancomycin to reach its periplasmic target, namely peptidoglycan precursors, and results in cell death. Of note, since the physiological manifestations of lprG/rv1410c deletions are complex¹⁰, other potential mechanisms not directly associated with TAG transport might also contribute to the vancomycin sensitivity phenotype.

To acknowledge the fact that Rv1410-LprG might have functions that still remained undiscovered, we added the following statement right at the beginning of the discussion section:

In this work, we provide structural and mechanistic insights into the mycobacterial MFS transporter Rv1410, which has been shown previously to work together with its operon partner LprG to transport triacylated lipids such as TAGs to the mycomembrane. We wish to note that that Rv1410 and LprG possibly fulfil additional, yet undiscovered functions, by transporting further lipid-like substrates that might have escaped detection in a previous lipidomics study¹².

Line 54: “more recent work has shown that Rv1410 indirectly contributes to drug tolerance by sealing the mycomembrane with triacylglycerides (TAGs).”

The first half of this sentence is supported for *M. smegmatis* by Hohl et al. as cited (although the authors are encouraged to use the word “tolerance” carefully, as this has a particular meaning in the field; they may mean instead sensitivity or resistance). However,, they connect this with a TAG accumulation phenotype in *Mtb* from the cited Martinot et al study. Importantly, this phenotype has not been confirmed in *Msm*. Neither paper shows that TAGs are responsible for changes in cell envelope integrity (since Hohl et al. examined uptake into the cytosol and did not distinguish permeability of the mycomembrane). As stated, this sentence is an example of oversimplification and overinterpretation of a correlation.

We fully agree that tolerance is inappropriate wording here, and it was changed to drug resistance.

In addition, we agree that a formal proof of whether Rv1410-LprG also transports TAGs in *M. smegmatis* (and not only *M. tuberculosis*) has never been made (though given the high sequence conservation of Rv1410 and LprG, it is a highly plausible assumption).

Here our adjusted wording in the manuscript.

However, more recent work suggests that Rv1410 indirectly contributes to drug resistance, likely by sealing the mycomembrane⁸ with triacylglycerides (TAGs)¹².

We wish to note here that Hohl et al. is a previously published and peer-reviewed paper by our group. Indeed, BCECF-AM gets fluorescent only within the cytoplasm, but in order to get there it needs to pass outer and inner membrane; under the conditions we carried out the assay (i.e. in the presence of proton uncoupler CCCP), we looked solely at downhill flux. And it is well known that the outer membrane is the more elaborate/thicker membrane to pass as opposed to the much simpler inner membrane, hence we are fairly confident that BCECF-AM influx differences reflected on outer membrane permeability in that case.

Lines 56-58, “operon overexpression led to elevated secretion of TAGs into the culture medium, hence directly demonstrating that Rv1410 is a TAG transporter.” This is an inference, not a direct demonstration. Elevated secretion of TAGs could arise from excess TAG at the mycomembrane, but this could be an indirect effect of overexpressing lprG-rv1410c, e.g., as a stress response to gain of function in this operon.

Although this critics targets previous works published by colleagues in the field (Martinot et al, PMID: 26751071), the reviewer has a point in saying that elevated TAG secretion to the culture medium is not a direct demonstration of TAG transport, but might be a consequence of protein overexpression. More solid evidence for this claim could have been obtained by overexpressing a transport-defective mutant of Rv1410 and perform the analysis; but such an analysis was unfortunately not offered in Martinot et al. Hence, it is indeed an overstatement to call this published result a direct demonstration.

We changed the sentence to :

Conversely, operon overexpression led to elevated secretion of TAGs into the culture medium, hence offering additional evidence that Rv1410 is a TAG transporter¹².

Line 133: “Rv1410 and LprG have been shown previously to extract TAGs from the inner membrane so that TAGs could reach the mycomembrane where they secure its impermeability”

We agree that the literature cited (refs 8 and 12) support an inferred model. However, the statement “shown... to extract” implies that this was experimentally observed. This is the case for LprG, which was tested in a model vitro system with soluble truncated protein, but is not true for Rv1410c, for which no measurements of TAG transport/extraction have been published. Also, as above, there is no experimental evidence that TAG itself contributes to the impermeability of the mycomembrane, only that the loss of lprG-rv1410c results in permeability defects and antibiotic sensitivity, which could be due to pleiotropic effects of genetic modification on cell envelope content and structure.

We agree that this wording sounds too conclusive. We thus reformulated the section of line 133ff the following way:

Deletion of lprG/rv1410c homologous operon MSM3070/69 in M. smegmatis results in increased vancomycin susceptibility^{8,10,35,36}, a phenotype we exploited here to study the transport activity of Rv1410 and MHAS2168 mutants. According to more recently published mechanistic models^{8,12}, deletion of lprG/rv1410c results in diminished TAG transport to the mycomembrane, which in turn becomes more permeable to vancomycin to reach its periplasmic target, namely peptidoglycan precursors, and results in cell death. Of note, since the physiological manifestations of lprG/rv1410c deletions are complex¹⁰, other potential mechanisms not directly associated with TAG transport might also contribute to the vancomycin sensitivity phenotype.

Reviewer #3 (Remarks to the Author)

The authors have thoroughly addressed all my concerns, and I have no further comments.

Reviewer #4 (Remarks to the Author)

The authors addressed most of my comments and performed a lot of additional simulations and analyses. I therefore have only minor comments that should help the authors to polish the manuscript. I do not need to be included in the next round of reviews, if it happens.

- The authors write "However, no TAG molecule entered the cavity in the MD simulations, likely because of the much lower TAG concentration compared to the competing phospholipid species." (pages 31-32 in the SI). The authors performed simulations at higher TAG concentration, as I suggested in the first round, but opted not to include these data in the final manuscript, saying

"Therefore, as the results of these additional control simulations do not contribute much to the overall mechanistic conclusions, we would prefer not to include them in the manuscript."

I find this situation not optimal. I'd suggest to include these results in the SI, not only because it's an interesting result on its own, but also to not potentially waste the time and resources of other researchers that might want to study systems containing higher TAG concentrations.

We agree and followed the suggestion by reviewer #4. The following sentence was added to the Suppl. Information (p. 37, end of 1st paragraph): *"Spontaneous TAG binding was also not observed in additional control CG-MD simulations with a higher concentration of TAG in the bilayer (7%, corresponding to 14 TAG molecules), likely due to the aggregation of the TAGs in the membrane midplane."* We also updated the Supplementary Table 3 with the new MD simulations.

- The authors mention that "each lipid species and solvent separately coupled to an external bath using the v-rescale thermostat" this is somewhat unexpected, the authors might want to clarify why multiple thermostats were used.

This is the default procedure adopted in the CHARMM-GUI, which we followed.

- Figures S7, S9 and S12 both show TAG-protein contacts, in figure S7 this number is normalized but not in S9 and S12. I'd suggest to make these figures in a consistent manner to not confuse the reader.

The reviewer is correct. We now updated the Supplementary Figure 7 with the absolute number of contacts to avoid confusion.